# Lack of p38 activation in T cells increases IL-35 and protects against obesity by promoting thermogenesis

Ivana Nikolic [1,8] ✉, Irene Ruiz-Garrido[1,8], María Crespo[1], Rafael Romero-Becerra [1], Luis Leiva-Vega[1,2], Alfonso Mora [1,2], Marta León[1], Elena Rodríguez[1,2], Magdalena Leiva[1,3], Ana Belén Plata-Gómez [2], Maria Beatriz Alvarez Flores [1], Jorge L Torres [4,5], Lourdes Hernández-Cosido [6], Juan Antonio López [1,7], Jesús Vázquez [1,7], Alejo Efeyan [2], Pilar Martin [1,7], Miguel Marcos [4] & Guadalupe Sabio [1,2] ✉

## Abstract

**Obesity is characterized by low-grade inflammation, energy imbalance and impaired thermogenesis. The role of regulatory T cells (Treg) in inflammation-mediated maladaptive thermogenesis is not well established. Here, we find that the p38 pathway is a key regulator of T cell-mediated adipose tissue (AT) inflammation and browning. Mice with T cells specifically lacking the p38 activators MKK3/6 are protected against diet-induced obesity, leading to an improved metabolic profile, increased browning, and enhanced thermogenesis. We identify IL-35 as a driver of adipocyte thermogenic program through the ATF2/UCP1/FGF21 pathway. IL-35 limits CD8+ T cell infiltration and inflammation in AT. Interestingly, we find that IL-35 levels are reduced in visceral fat from obese patients. Mechanistically, we demonstrate that p38 controls the expression of IL-35 in human and mouse Treg cells through mTOR pathway activation. Our findings highlight p38 signaling as a molecular orchestrator of AT T cell accumulation and function.**

**Keywords** Adipose Tissue; Obesity; p38 Stress Kinases; Thermogenesis; T Regulatory Cells
**Subject Categories** Immunology; Metabolism; Signal Transduction

## Introduction

Obesity is a serious worldwide health epidemic associated with an increased risk of life-threatening diseases (type 2 diabetes—T2D, cardiovascular diseases, and cancer) (Nikolic et al, 2020). The hallmark of obesity is a chronic low-grade inflammation, both systemically and in adipose tissue (AT). This inflammation is one of the main promoters of insulin resistance and the development of T2D (Saltiel and Olefsky, 2017). Inflammatory myeloid cells, including macrophages and neutrophils, are key players in obesity-related comorbidities such as diabetes and steatosis (Crespo et al, 2020; Gonzalez-Teran et al, 2016b). Chronic inflammation in obese AT is also mediated by adaptive immune cells, including CD8+ and CD4+ T cells (Schipper et al, 2012), and absence of these conventional T cells in mice reduces AT inflammation (Khan et al, 2014). CD8+ T cell infiltration and accumulation in AT plays a crucial role in the recruitment of macrophages and the maintenance of inflammation (Nishimura et al, 2009). Single-cell RNA sequencing (scRNA-seq) of AT obtained from obese patients identified a potentially dysfunctional population of CD8+ T cells associated with metabolic disease (Vijay et al, 2020). Moreover, whereas infiltration of AT by conventional T cells increases during obesity, the numbers of protective T regulatory (Treg) cells are sharply decreased, and expansion of this population ameliorates AT inflammation and insulin resistance (Ilan et al, 2010; Li et al, 2021). Tregs are abundant in lean AT from mice and humans, and levels of the Treg hallmark transcription factor Foxp3 show an inverse correlation with body mass index (Feuerer et al, 2009). This suggests that preserving Treg accumulation in AT could be a novel anti-obesogenic strategy.

In addition to controlling the immune response within AT, immune cells play a significant role in regulating AT thermogenic function. The recruitment of anti-inflammatory macrophages and type 2 innate lymphoid cells actively promotes adaptive thermogenesis (Cereijo et al, 2018; Lee et al, 2015), while pro-inflammatory macrophages and CD8+ T cells are one of the main promoters of the compromised thermogenic capacity of AT (Moysidou et al, 2018; Sakamoto et al, 2016). However, the specific role of Treg cells in thermogenesis remains unclear, with some data indicating that this cell population facilitates obesity development and represses AT browning (Beppu et al, 2021), while other studies highlight the protective role of Treg cells in promoting adaptive thermogenesis and browning (Fang et al, 2020; Kalin et al, 2017). Further research in this field is necessary to gain a comprehensive understanding of

[1]Centro Nacional de Investigaciones Cardiovasculares (CNIC), Madrid 28029, Spain. [2]Programme of Molecular Oncology, Spanish National Cancer Research Center (CNIO), Madrid 28029, Spain. [3]Department of Immunology, School of Medicine, Universidad Complutense de Madrid, Madrid 28040, Spain. [4]Department of Internal Medicine, University Hospital of Salamanca-IBSAL, Department of Medicine, University of Salamanca, Salamanca 37007, Spain. [5]Complejo Asistencial de Zamora, Zamora 49022, Spain. [6]Bariatric Surgery Unit, Department of General Surgery, University Hospital of Salamanca, Department of Surgery, University of Salamanca, Salamanca 37007, Spain. [7]CIBER de Enfermedades Cardiovasculares, Madrid 28029, Spain. [8]These authors contributed equally: Ivana Nikolic, Irene Ruiz-Garrido. ✉E-mail: inikolic@cnic.es; gsabio@cnio.es

how Treg cells influence thermogenic mechanisms. Such insights could provide valuable information regarding the development of obesity and potential therapeutic strategies targeting AT metabolism.

The p38 kinases, comprising four members (p38α, p38β, p38γ, and p38δ), belong to the stress protein kinase family and are activated by the upstream MAP kinase kinases MKK3 and MKK6 (Gonzalez-Teran et al, 2013). The p38 pathway serves as a central regulator of the immune response and plays a critical role in the development of inflammatory diseases associated with obesity, becoming activated in various organs, including immune cells (Nikolic et al, 2020). For instance, the activation of p38γ/δ in neutrophils and p38α in macrophages promotes liver steatosis (Gonzalez-Teran et al, 2016b; Zhang et al, 2019). Interestingly, Treg cells present marked activation of the p38 pathway compared with conventional T cells (Huber et al, 2008), and the lack of p38α/β in T cells enhances Treg cell induction in mice (Hayakawa et al, 2017). Therefore, we wondered how p38 activation in T cells would affect AT inflammation and function during obesity. To address this question, we generated mice with specific deletion of the p38 family upstream activators MKK3 and MKK6 in conventional T cells (MKK3/6[CD4-KO] mice). In our study, we observed that the lack of p38 activators results in an increased population of Treg cells in circulation, lymph nodes, and AT in obesity. Furthermore, we described a novel function of p38 kinases in Treg cells: repressing IL-35 expression in human and mouse Treg cells through mTOR pathway inhibition. IL-35 levels, which are reduced in visceral fat from obese patients, directly correlate with reduced CD8[+] T cell infiltration and inflammation in AT. In addition, it is associated with an increased adipocyte thermogenic program characterized by activation of the ATF2 pathway, which correlates with higher UCP1 and FGF21 levels. Consequently, mice lacking the upstream activators of p38 kinases in T cells were protected against diet-induced obesity (DIO) and AT inflammation, along with an improved metabolic profile characterized by enhanced browning and reduced liver steatosis. Our findings highlight the role of p38 signaling in reducing Treg accumulation in AT and identifies this kinase family as an important regulator in T cell function during obesity. In addition, we identify IL-35 as a potential therapeutic target for immunotherapy aimed at combating obesity.

## Results

### Lack of MKK3/6 in T cells protects against HFD-induced obesity, type 2 diabetes, and liver steatosis

The significance of the p38 kinase family in innate cells, such as macrophages and neutrophils, in relation to obesity and metabolic diseases, has been extensively researched (Nikolic et al, 2020). However, the role of this stress kinase family in T cells during obesity is not fully understood. As an initial step, we took advantage of the data available from single-cell RNA sequencing of human AT from White Adipose Atlas (Emont et al, 2022) and examined the expression pattern of the p38 signaling pathway in the cluster of T cells infiltrating AT from lean individuals (BMI < 30) and patients with obesity (BMI 30–40) or severe obesity (BMI 40–50) (Fig. 1A and Dataset EV1). Patients with severe obesity (BMI 40–50) exhibited a signature of activation of p38

pathway in T cells from AT, as opposed to lean individuals (BMI 20–30). Precisely, we found higher expression of p38α (*MAPK14*), its upstream activator, MKK6 (*MAP2K6*), and its substrate *ATF2*, along with down-expression of the negative regulator the phosphatase *DUSP1* (Fig. 1B,C). All together, these data suggest that the p38 signaling pathway has a role in T cells during obesity.

To evaluate this hypothesis, we generated mice lacking the p38 upstream activators, MKK3 and MKK6, specifically in conventional T cells (MKK3/6[CD4-KO]). Both kinases were specifically deleted in CD4[+] and CD8[+] T cells, whereas expression was unaffected in other cell types and tissues (NK, liver, and AT) (Appendix Fig. S1A). Since previous studies using mice lacking p38 family members (Hayakawa et al, 2017; Risco et al, 2018) showed lymphoid dystrophy, we evaluated whether the lack of p38 activation results in lymphocyte abnormalities. Analysis of thymus, spleen and peripheral lymphoid organs' weight and cell number showed no differences in MKK3/6[CD4-KO] mice compared to control mice CD4-Cre (Appendix Fig. S1B,C). In addition, we did not observe differences in T cell development in the thymus (Appendix Fig. S1D). Furthermore, we measured real-time bioenergetic changes in naive CD4[+] T cells activated with PMA/Ionomycin and found no differences between genotypes (Appendix Fig. S1E). These data suggest that, unlike p38 deficiency, the lack of p38 activation in T cells does not induce lymphoid dystrophy or problems in T cell activation.

Interestingly, MKK3/6[CD4-KO] mice were leaner than control CD4-Cre mice and showed lower epididymal WAT (eWAT) and subcutaneous WAT (sWAT) weight (Appendix Fig. S1F,G). Metabolic cages analysis suggested a trend towards increased energy expenditure (EE) in chow-diet-fed MKK3/6[CD4-KO] mice (Fig. EV1A), without differences in food intake or locomotor activity (Fig. EV1B). In addition, chow-diet-fed MKK3/6[CD4-KO] mice presented elevated brown adipose tissue (BAT) thermogenesis, determined by higher interscapular temperature (Fig. EV1C,D), compared to both CD4-Cre mice and their littermates (MKK3/6[f/f]). These data suggest that p38 activation in T cells regulates BAT temperature and energy homeostasis. In addition, it might have an important role in the development of obesity-associated metabolic diseases.

To further elucidate the contribution of p38 activation in T cells during obesity, we fed animals a high-fat diet (HFD) for 8 weeks. MKK3/6[CD4-KO] mice were protected from HFD-induced obesity compared to CD4-Cre mice (Fig. 2A). The lower body weight was correlated with reduced body and fat mass, as measured by MRI (Fig. 2B), reduced fat-depot and liver mass (Fig. 2C), without differences in food intake (Fig. 2D). A similar phenotype was observed in MKK3/6[CD4-KO] female mice (Figs. EV2A–C). While metabolic cages analysis from these animals did not show differences in EE (Kcal/h) (Fig. 2E,F), when EE corrected by lean mass was performed, we observed an increase of EE in HFD-fed MKK3/6[CD4-KO] mice compared to those in CD4-Cre control mice (Fig. 2G,H). Since obesity is usually followed by hyperglycemia and ectopic fat accumulation in liver (also known as liver steatosis) (Nikolic et al, 2020), analysis of blood glucose and liver histology showed protection against hyperglycemia and steatosis in HFD-fed MKK3/6[CD4-KO] mice (Fig. 2I,J). We also performed glucose and insulin tolerance test (GTT and ITT), and found no differences in glucose tolerance but significantly improved insulin sensitivity in HFD-fed MKK3/6[CD4-KO] mice (Appendix Fig. S2A,B).

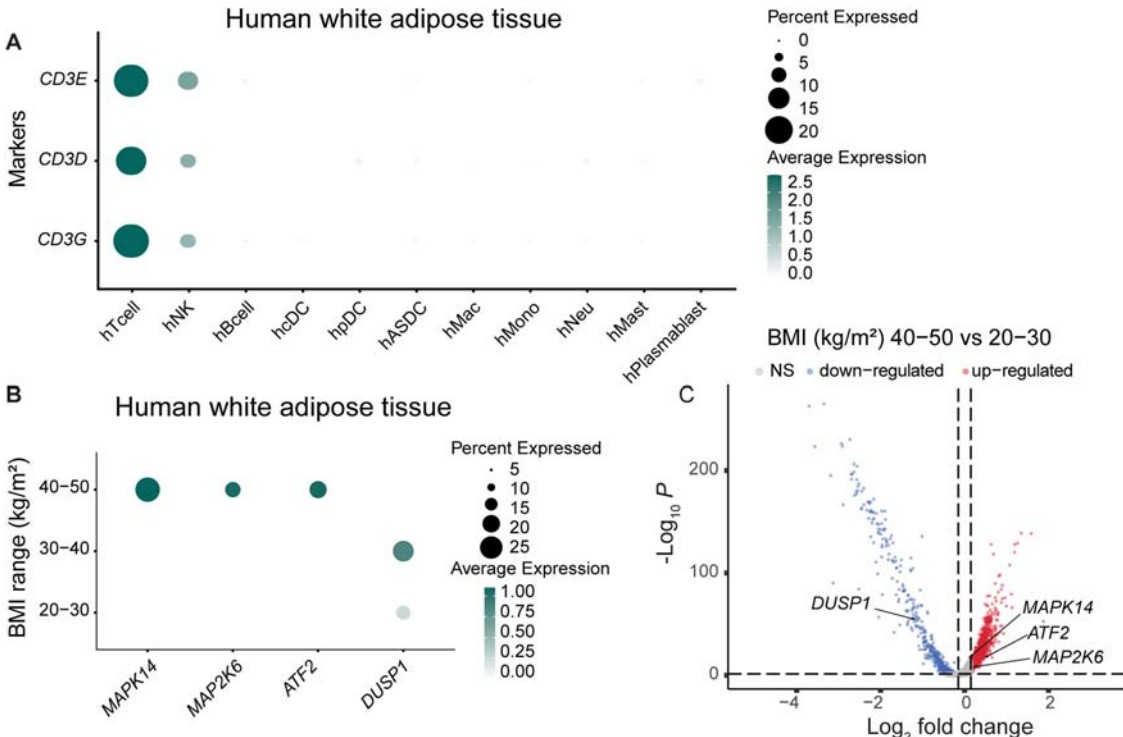

**Figure 1. p38 MAPK pathway is upregulated in T cells in obese human white adipose tissue.**

(A) Dot plot illustrating the expression of the indicated T cell marker genes in the different cell type clusters in human white AT single-cell RNA-seq data from Emont et al (2022). (B) Dot plot showing the expression of the indicated genes by BMI range in human white AT T cell cluster shown in (A). (C) Volcano plot of differentially expressed genes in human white AT T cell cluster in severe obese (BMI 40–50 kg/m²) versus non-obese (BMI 20–30 kg/m²) subjects. The vertical dashed line indicates a log2 fold change cut-off of 0.15. The horizontal dashed line indicates a −log10 adjusted p-value (using Bonferroni correction) cut-off of 1.3 (adj p-value < 0.05). Data information: Differentially expressed genes between BMI ranges were identified with a non-parametric Wilcoxon rank sum test. Obese (BMI 40–50 kg/m²) n = 6 biologically independent patients; Non-obese (BMI 20–30 kg/m²) n = 10 biologically independent patients; DC: dendritic cells; Mac: macrophages; Mono: monocytes; Neu: neutrophils; Mast: mastocytes. Source data are available online for this figure.

In addition, we observed protection against DIO in MKK3/6^CD4-KO mice compared with their MKK3/6^f/f littermates (Fig. 3A,B). The results from metabolic cages indicated that DIO was not due to impaired intestinal absorption of lipids or to higher locomotor activity, as HFD-fed MKK3/6^CD4-KO mice actually showed higher lipid absorption and reduced locomotor activity (Fig. 3C,D). Interestingly, HFD-fed MKK3/6^CD4-KO mice exhibited higher BAT temperature compared to littermates (Fig. 3E). Moreover, we observed that the protection against obesity and the increase in BAT temperature also occur in HFD-fed MKK3/6^CD4-KO female mice (Fig. EV2D).

### Deficiency of MKK3/6 in T cells increases adipose tissue thermogenesis and browning

Our group has shown that the immune system regulates body weight by influencing AT thermogenesis (Crespo et al, 2023), and we observed higher BAT temperature in mice lacking p38 activation in T cells. To confirm these differences under thermo-neutral conditions, we placed CD4-Cre and MKK3/6^CD4-KO mice at 30 °C throughout the entire HFD feeding. We observed that under isothermal housing, MKK3/6^CD4-KO mice maintained their protection against HFD-induced obesity (Fig. 4A) showing reduced body and fat mass evaluated by MRI (Fig. 4B), reduced fat depots

(Fig. 4C), adiposity (Fig. 4D) and liver steatosis (Fig. 4D). In addition, MKK3/6^CD4-KO mice preserved higher BAT temperature at thermoneutrality (Fig. 4E), confirming that protection against obesity is autonomous and independent of animal housing temperature. In addition, we injected mice with norepinephrine (NE, 1 mg/kg of BW) to evaluate thermogenic capacity of mice at thermoneutrality. Our results showed that HFD-fed MKK3/6^CD4-KO mice maintained higher basal BAT temperature after 4 weeks at thermoneutrality; when stimulated with NE, they reached the same peak BAT temperature peak as control MKK3/6^f/f mice (Appendix Fig. S3A,B). Moreover, we analyzed the expression of thermogenic genes in BAT at thermoneutrality and we did not find a significant increase in their expression in MKK3/6^CD4-KO mice compared to the control group, although there is a tendency for increased PGC1α expression (Appendix Fig. S3C).

Given that MKK3/6^CD4-KO mice showed heightened BAT temperature in comparison to both littermates (Fig. 3E) and CD4-Cre (Fig. 5A), we conducted a histological analysis of BAT morphology. The histological analysis unveiled increased cellularity and maintained BAT morphology in MKK3/6^CD4-KO mice, characterized by typical multilocular lipid droplets (Fig. 5B). Multi-locular cells with high energy-dispensing activity have been linked to elevated expression of the mitochondrial BAT marker UCP1 (Leiva et al, 2020). UCP1 is a crucial mitochondrial protein

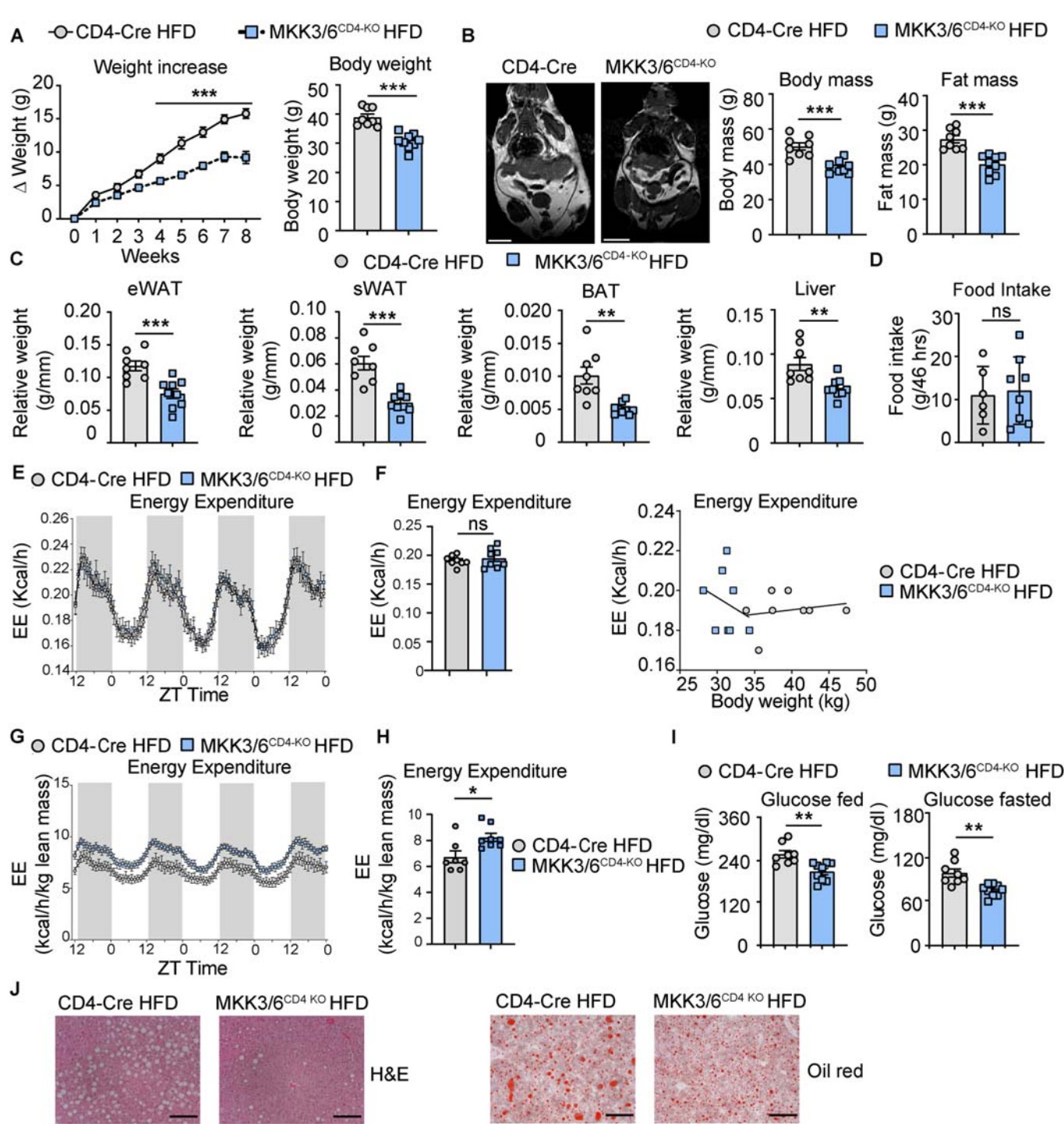

involved in BAT thermogenesis, which uncouples oxidative phosphorylation from ATP production, therefore dissipating chemical energy as heat (Enerbäck et al, 1997). Quantitative RT-PCR and western blot analysis confirmed high UCP1 expression in BAT from HFD-fed MKK3/6[CD4-KO] mice (Fig. 5C,D), correlating with BAT activation and elevated thermogenesis-related genes, *Fgf21* and its target *Ppargc1a* in HFD-fed MKK3/6[CD4-KO] mice (Fig. 5D). It has been shown that FGF21 acts as an autocrine factor in adipocytes, inducing UCP1 expression and promoting the

thermogenic program (Reilly et al, 2021). These results may indicate that deficiency of p38 activation in T cells promotes BAT thermogenesis thus restraining obesity development.

T cells have been shown to regulate the browning of WAT (Kohlgruber et al, 2018; Moysidou et al, 2018), as well as adipose-derived FGF21 in an autocrine/paracrine manner (Fisher et al, 2012). Therefore, we evaluated whether the lack of p38 activation in T cells induced browning. We found significant upregulation of the browning-signature markers *Ucp1*, PR domain-containing16

**Figure 2. MKK3/6 deficiency in T cells protects against HFD-induced obesity.**

(A–J) MKK3/6$^{CD4-KO}$ and CD4-Cre mice were fed a high-fat diet (HFD) for 8 weeks (starting at 8–10 weeks old). (A) Body weight evolution in MKK3/6$^{CD4-KO}$ and CD4-Cre male mice for 8 weeks. Data are presented as the increase above initial weight (left) and absolute weight at the end of the experiment (right). (B) MRI analysis of body and fat mass in CD4-Cre and MKK3/6$^{CD4-KO}$ mice after 8 weeks of HFD. Representative images are shown on the left. (C) eWAT, sWAT, BAT, and liver mass relative to tibia length. (D) Food intake during 46 h. (E, F) Comparison of energy balance between HFD-fed MKK3/6$^{CD4-KO}$ and CD4-Cre mice examined in metabolic cages. Hour-by-hour variation in EE (kcal/h) (E); mean EE (Kcal/h) and ANCOVA analysis of EE (kcal/h) (F). (G, H) Comparison of energy balance between HFD-fed MKK3/6$^{CD4-KO}$ and CD4-Cre mice examined in metabolic cages. Hour-by-hour lean–mass-corrected variation in EE (kcal/h/kg) (G) and mean lean mass-corrected EE (kcal/h/kg) (H). (I) Blood glucose concentration after 8 weeks of HFD in mice fed (left) or fasted overnight (right). (J) Representative haematoxylin–eosin and oil-red O staining of liver sections. Data Information: Data are presented as mean ± SEM, *$p < 0.05$, **$p < 0.01$, ***$p < 0.001$, ns: not significant. Analysis by 2-way ANOVA coupled to the Bonferroni post-test (A) or coupled to the Sidak's multiple comparison post-test (E, G); and by $t$ test or by the Welch test when variances were different (A–C, F, H, I). $n = 6$–9 biologically independent mice for each group, represented as single dots in the graphs (A–I). Scale bar: 1 cm (B), 100 µm (J). Source data are available online for this figure.

(*Prdm16*), and peroxisome proliferator-activated receptor gamma (*Ppargc1*) in eWAT and sWAT from HFD-fed MKK3/6$^{CD4-KO}$ mice (Fig. 5E,G). We also observed smaller adipocyte size in both eWAT and sWAT from HFD-fed MKK3/6$^{CD4-KO}$ mice (Fig. 5F,H). Considering the association of obesity with impaired AT metabolism (Leiva et al, 2020) and the emerging evidence for the role of T lymphocytes in the regulation of AT (Mathis, 2013), we profiled metabolism-related gene expression in eWAT and sWAT from HFD-fed MKK3/6$^{CD4-KO}$. RT-PCR revealed increased expression of genes involved in adipogenesis (*AdipoQ* and *Plin1*), lipogenesis (*Ppard*, *Acaca*, and *Scd1*), β-oxidation (*Acox1* and *Cpt1*), and glycolysis (*Pepck*) in both eWAT and sWAT (Fig. EV3A,B). These results provide evidence for a T cell–AT crosstalk, regulated by p38 activation, that controls AT metabolism and browning, especially in obesity.

## T cell MKK3/6-deficiency increases Treg accumulation and prevents CD8$^+$ infiltration into adipose tissues

Obesity is considered as low-grade inflammation condition, so we next examined the cellular mechanism underlying the observed DIO protection by flow cytometry (gating strategy presented in Appendix Fig. S4). First, we analyzed CD4$^+$, CD8$^+$ and Treg population in the spleen, blood, and AT draining lymph node after HFD. While there were no differences in CD4$^+$ and CD8$^+$ cells in the observed lymphoid compartments and circulation (Figs. EV4A–C), we found more Treg cells in the spleen, blood, and lymph nodes in mice deficient for p38 activation in T cells after HFD (Figs. EV4A–C), consistent with previously published data showing that p38 deficiency enhances Treg induction (Hayakawa et al, 2017). Obesity leads to decrease of protective Treg cells in AT (Feuerer et al, 2009), hence we analyzed if the higher frequency of Treg cells observed in MKK3/6$^{CD4-KO}$ mice was also evident in AT. Our results demonstrate that p38 activation in T cells inhibits Treg cell accumulation in AT and, as a result, HFD-fed MKK3/6$^{CD4-KO}$ mice show AT Treg cell expansion (Fig. 6A; Appendix Fig. S5). To rule out that these changes were due to a leaner phenotype, we analyzed AT Treg cell population in mice after 2 weeks of HFD where there were no differences in body or fat depots weight (Appendix Fig. S6A,B). We found an increased frequency of Treg cells in lymph nodes and circulation (Appendix Fig. S6C,D), suggesting that Treg cells were migrating to inflamed tissue, such as AT during obesity. Although we found a lower percentage of Treg cells in eWAT in MKK3/6$^{CD4-KO}$ mice, the number of cells per gram

of tissue was higher compared to littermates (Appendix Fig. S6E). These data suggest that while MKK3/6$^{CD4-KO}$ mice have more Treg cells in eWAT, the number of Treg cells declines in littermates shortly after starting the HFD. To corroborate this, we checked whether p38 signaling pathway was differentially expressed in the cluster of Treg cells in the White Adipose Atlas (Emont et al, 2022) (Fig. EV5A). We observed higher expression of p38α (*MAPK14*) in AT Treg cells from human with obesity (BMI 30–50) or with severe obesity (BMI 40–50) compared with lean individuals (BMI 20–30) (Figs. EV5B–EV5E and Datasets EV2 and EV3). We also found that its upstream activators, MKK6 (*MAP2K6*, Fig. EV5B) and MKK3 (*MAP2K3*), were upregulated in Treg cells from human with obesity (Figs. EV5B,F,G and Dataset EV4). In Treg cells from humans with severe obesity, the expression of phosphatase *DUSP1* was downregulated (Fig. EV5B), while *ATF-2* was also upregulated in AT Treg cells from humans with a BMI 40–50 (Fig. EV5B). Altogether, these data indicate that p38 signaling pathway is upregulated in Treg cells from the AT of humans with obesity.

Since Treg cells possess immunosuppressive properties and may play a role in suppressing infiltration of CD8$^+$ obesogenic T cells in AT (Burzyn et al, 2013; Feuerer et al, 2009), we measured CD8$^+$ T cells in eWAT by FACS. HFD-fed MKK3/6$^{CD4-KO}$ mice exhibit lower infiltration of CD8$^+$ cells in eWAT than HFD-fed CD4-Cre mice (Fig. 6B). CD8$^+$ cell infiltration precedes macrophage accumulation and promotes pro-inflammatory macrophage recruitment (Nishimura et al, 2009). In line with this, eWAT from HFD-fed MKK3/6$^{CD4-KO}$ mice presented less abundance of obesogenic, pro-inflammatory macrophages and NK cells (Fig. 6C). Notably, we found a higher frequency of anti-inflammatory macrophages (F4/80$^+$CD11b$^+$CD206$^+$) after 2 weeks of HFD, a condition that is maintained after 10 weeks of HFD (Appendix Fig. S6E). These data suggest that p38 activation deficiency in T cells prevents AT inflammation.

## p38 activation in Treg cells represses the expression of *Il-35* via the mTOR pathway

To investigate the molecular mechanisms through which Treg cells control AT inflammation, we assessed the expression of IL-35, an immunosuppressive cytokine produced by Treg cells, in both the AT and stromal vascular fraction (SVF). The expression of *Ebi3* and *p35*, the two subunits of the IL-35, were upregulated in HFD-fed MKK3/6$^{CD4-KO}$ SVF (Fig. 7A,B), highlighting p38s as regulators of IL-35 production in Treg cells. To confirm this, we analyzed

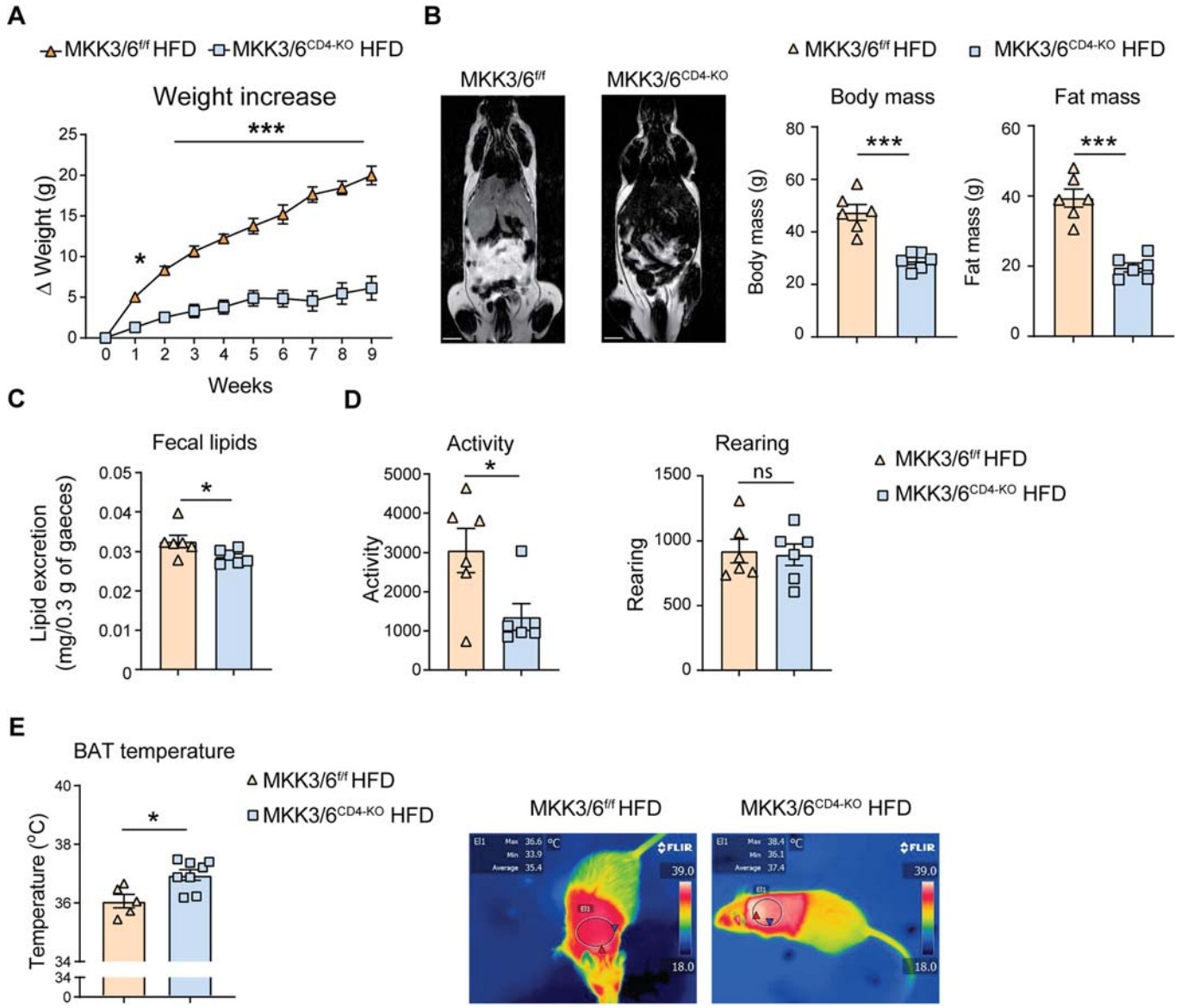

**Figure 3. MKK3/6 deficiency in T cells protects against HFD-induced obesity by increasing BAT temperature.**

(A–E) MKK3/6$^{CD4-KO}$ and MKK3/6$^{f/f}$ mice were fed a high-fat diet (HFD) for 9 weeks. (A) Body weight increased for 9 weeks. (B) MRI analysis of body and fat mass and representative images on the left. (C) Fecal lipid excretion over 5 days. (D) Locomotor activity and rearing over 24 h. (E) Skin temperature surrounding interscapular BAT. Right panels show representative infrared thermal images. Data Information: Data are presented as mean ± SEM, *p < 0.05, ***p < 0.001, ns: not significant. Analysis by 2-way ANOVA coupled to the Sidak's multiple comparison post-test (A) or t test or by the Welch test when variances were different (B–E). n = 6–8 biologically independent I'mmice for each group, represented as single dots in the graphs (A–E). Scale bar: 1 cm (B). Source data are available online for this figure.

IL-35$^{+}$ Treg cells in AT draining lymph nodes after HFD, and we observed a higher frequency of IL-35$^{+}$ Treg cells in HFD-fed MKK3/6$^{CD4-KO}$ mice (Fig. 7C). To gain deeper insights into the molecular mechanism underlying the control of IL-35 production by p38 activation, we next measured Il-35 expression in in Hvitro induced Treg cells (iTregs) (Fig. 7D). Consistent with the in vivo data from SVF and lymph nodes, the expression of Il-35 (p35) was higher in MKK3/6$^{CD4-KO}$ Treg cells than in their CD4-Cre counterparts (Fig. 7D). To confirm the action of this pathway in human Treg cells, we inhibited p38 with the p38 pan inhibitor BIRB796

(Kuma et al, 2005). As expected, p38 inhibition in human Treg cells induced the expression of Il-35 (P35) (Fig. 7E).

Previous studies have proposed a potential role of mTOR in regulating IL-35 levels (Zhang et al, 2017). In agreement, we found that Treg cells from mice with increased mTOR activation (TSC1$^{CD4-KO}$ mice) presented higher levels of IL-35 (Fig. 7F). mTOR has been shown to be modulated by p38 (Gonzalez-Teran et al, 2016a). In agreement, western blot analysis demonstrated that the lack of p38 activation in Treg cells leads to increased mTOR activation and higher S6 protein phosphorylation (Fig. 7G).

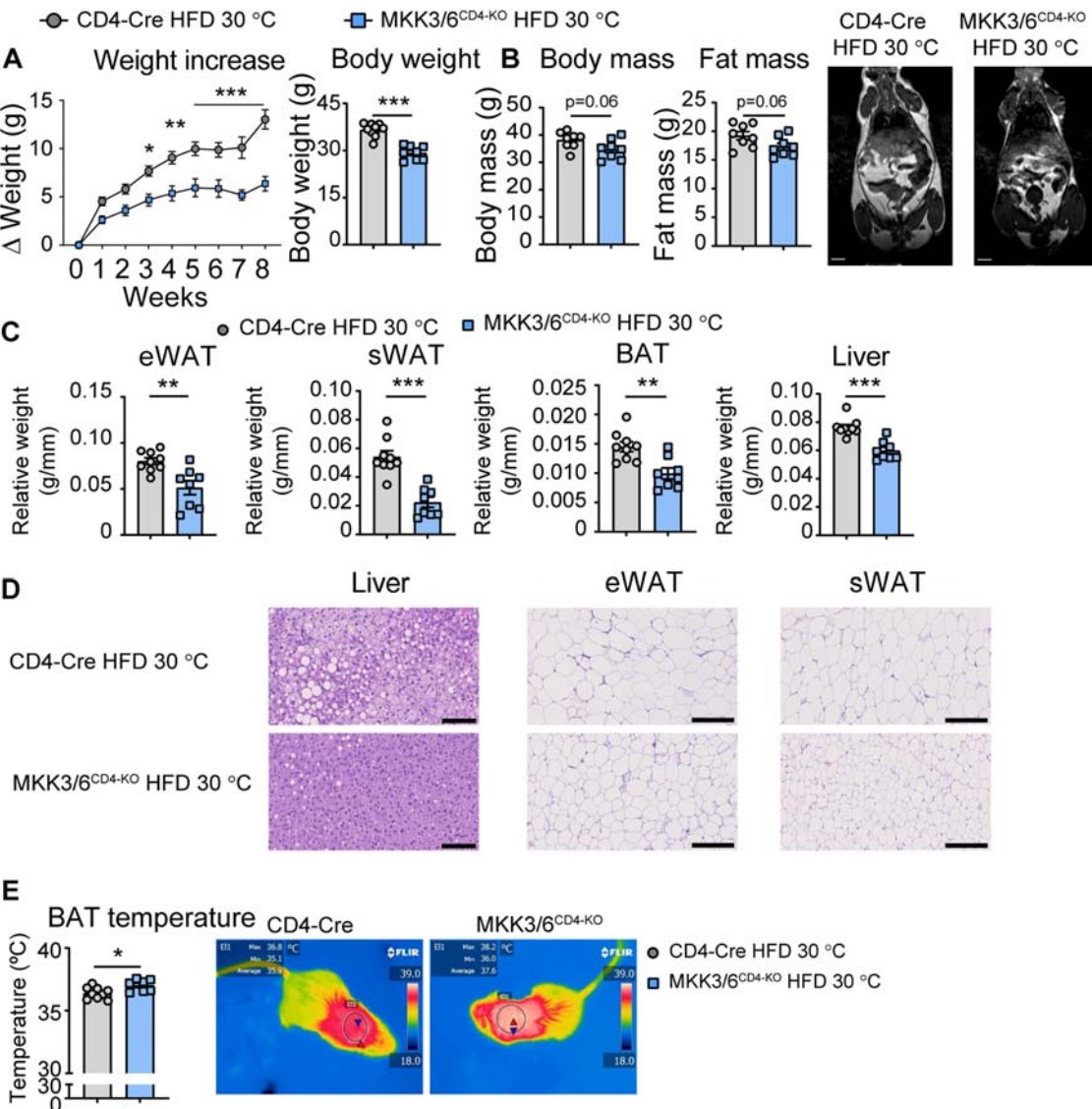

**Figure 4. MKK3/6 deficiency in T cells protects against HFD-induced obesity in isothermal housing.**

(A–E) MKK3/6[CD4-KO] and CD4-Cre mice were fed a high-fat diet (HFD) for 8 weeks (starting at 8–10 weeks old) and housed at 30 °C during whole course of HFD. (A) Body weight evolution in CD4-Cre and MKK3/6[CD4-KO] male mice for 8 weeks. Data are presented as the increase above initial weight (left) and absolute weight at the end of the experiment (right). (B) MRI analysis of body and fat mass in MKK3/6[CD4-KO] and CD4-Cre mice after 8 weeks of HFD. Representative images are shown on the right. (C) eWAT, sWAT, BAT, and liver mass relative to tibia length. (D) Representative haematoxylin–eosin and oil-red O staining of liver, eWAT, and sWAT sections. (E) Skin temperature surrounding interscapular BAT. Right panels show representative infrared thermal images. Data information: Data are presented as mean ± SEM, *$p < 0.05$, **$p < 0.01$, ***$p < 0.001$. Exact p-values are shown. Analysis by 2-way ANOVA coupled to the Bonferroni post-test (B) or $t$ test or by the Welch test when variances were different (A–C, E). $n = 7$–9 biologically independent mice for each group, represented as single dots in the graphs. Scale bar: 1 cm (B), 100 μm for liver and 250 μm for eWAT and sWAT (D). Source data are available online for this figure.

To evaluate whether mTOR activation in MKK3/6[CD4-KO] Treg cells was responsible for the increase in IL35 expression, we inhibited mTOR pathway with rapamycin. We found that IL-35 expression is dependent on mTOR activation, since treating Treg cells lacking p38 activation (MKK3/6[CD4-KO]) with rapamycin diminished the IL-35 expression (Fig. 7H). In summary, we described novel mechanism by which p38 kinases inhibits the mTOR signaling pathway that, in turn, reduces IL-35 production in mouse and human Treg cells. Inhibiting of p38s, either genetically or chemically, increased IL-35 mRNA and protein levels.

## Treg-derived IL-35 triggers adipose tissue browning and is reduced in obese patients

As MKK3/6[CD4-KO] mice expressed higher levels of IL-35 in AT, we postulated that IL-35 might have a role in AT thermogenesis and, consequently, in protection against obesity. Interestingly, we found a tendency for lower p35 expression in visceral AT obtained from individuals with obesity compared to lean individuals (Fig. 8A). In addition, MKK3/6[CD4-KO] mice exposed to cold preserved their body and BAT temperature, while the temperature of control CD4-Cre

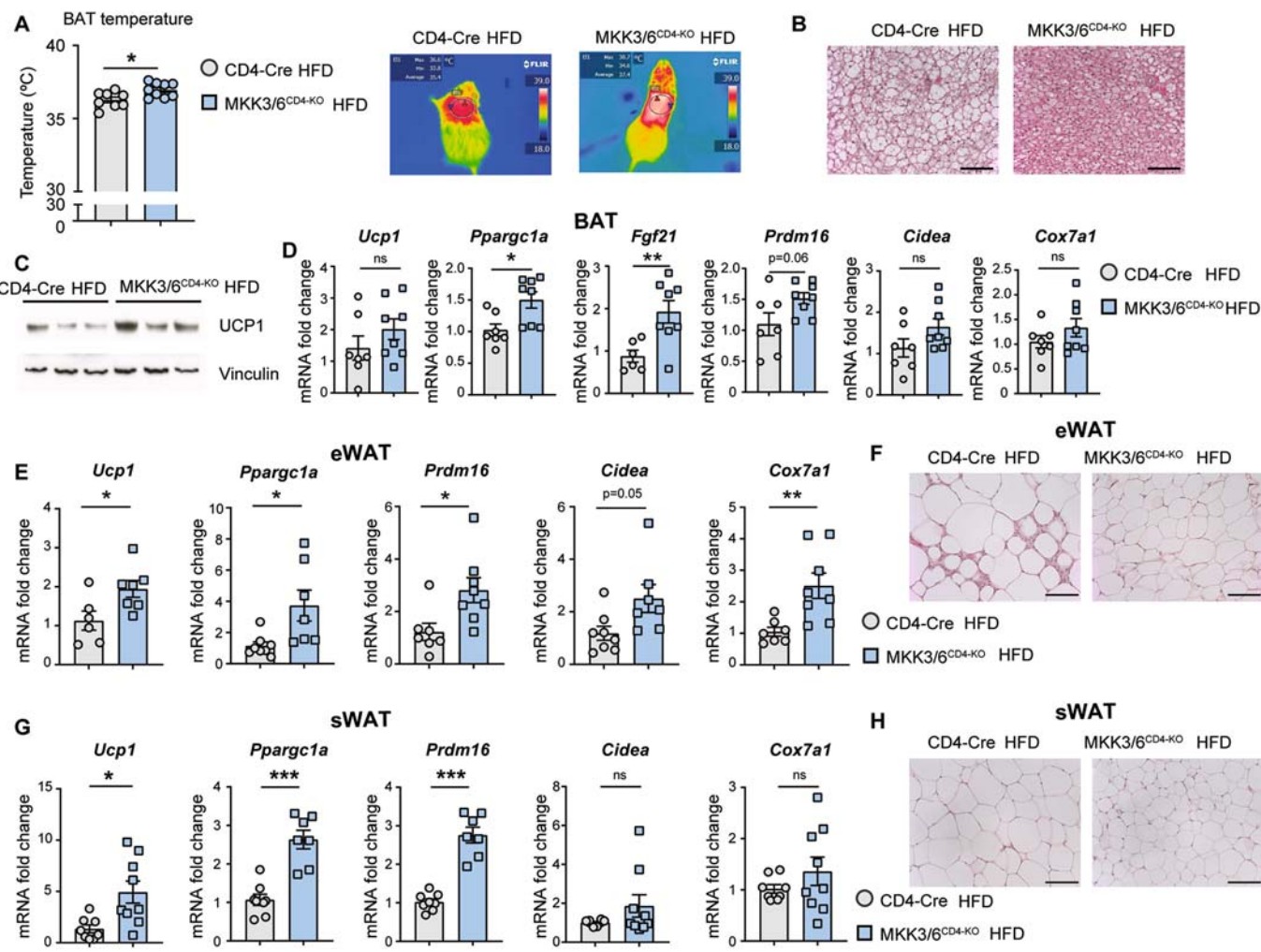

**Figure 5. Lack of MKK3/6 in T cells increases BAT thermogenesis and adipose tissue browning.**

(A–H) MKK3/6^CD4-KO and control CD4-Cre mice were HFD-fed for 8 weeks. (A) Skin temperature surrounding interscapular BAT. Right panels show representative infrared thermal images. (B) Representative H&E staining of BAT sections. (C) Western blot analysis of UCP1 in BAT. (D) qRT-PCR analysis of thermogenic gene mRNA expression in BAT isolated from control or MKK3/6^CD4-KO mice. mRNA expression was normalized to the expression of β-actin mRNA and presented as fold increase compared to CD4-Cre. (E) qRT-PCR analysis of browning genes mRNA expression from eWAT isolated from control or MKK3/6^CD4-KO mice. mRNA expression was normalized to β-actin mRNA and presented as fold increase compared to CD4-Cre. (F) Representative H&E staining of eWAT sections. (G) qRT-PCR analysis of browning genes mRNA expression from sWAT isolated from control or MKK3/6^CD4-KO mice. mRNA expression was normalized to β-actin mRNA and presented as fold increase compared to CD4-Cre. (H) Representative H&E staining of sWAT sections. Data information: Data are presented as mean ± SEM, *$p < 0.05$, **$p < 0.01$, ***$p < 0.001$, ns: not significant. Exact $p$-values are shown. Analysis by t test or by the Welch test when variances were different. $n = 7$–9 biologically independent mice for each group, represented as single dots in the graphs. Scale bar: 100 μm (B, F, H). Source data are available online for this figure.

mice gradually dropped during cold challenge (Fig. 8B). We next evaluated the functional role of IL-35 in thermogenesis in vivo. IL-35 injection triggered thermogenesis in mice compared to the PBS-treated group (Fig. 8C). To understand mechanistically how IL-35 cytokine controls thermogenesis in adipocytes, we treated brown adipocytes with IL-35 and found increased levels of UCP1 and FGF21, suggesting an activation of the brown thermogenic program (Fig. 8D). IL-35 treatment also resulted in ATF-2 phosphorylation (Fig. 8E), the transcription factor that controls UCP1 and PGC1A expression in BAT (Cao et al, 2004). This increase in *Ucp1* expression, induced by IL-35, was downregulated when ATF2 activation in adipocytes was blunted by using inhibitor of upstream activator of ATF2 (SB203580, p38α/β inhibitor) (Fig. 8F).

Furthermore, IL-35 also promoted adipogenesis in differentiated primary adipocytes, which was reduced when ATF2 activation was suppressed (Fig. 8G). In summary, we identified possible novel role of Treg-derived cytokine IL-35 in regulating AT thermogenesis through the pATF-2/UCP1/FGF21 axis.

## Discussion

Our data highlight a new role of p38 pathway in Treg cells that impairs BAT activation and browning during obesity. Our findings suggest that the molecular mechanism by which p38 activation regulates the production of IL-35 by Tregs is through the mTOR

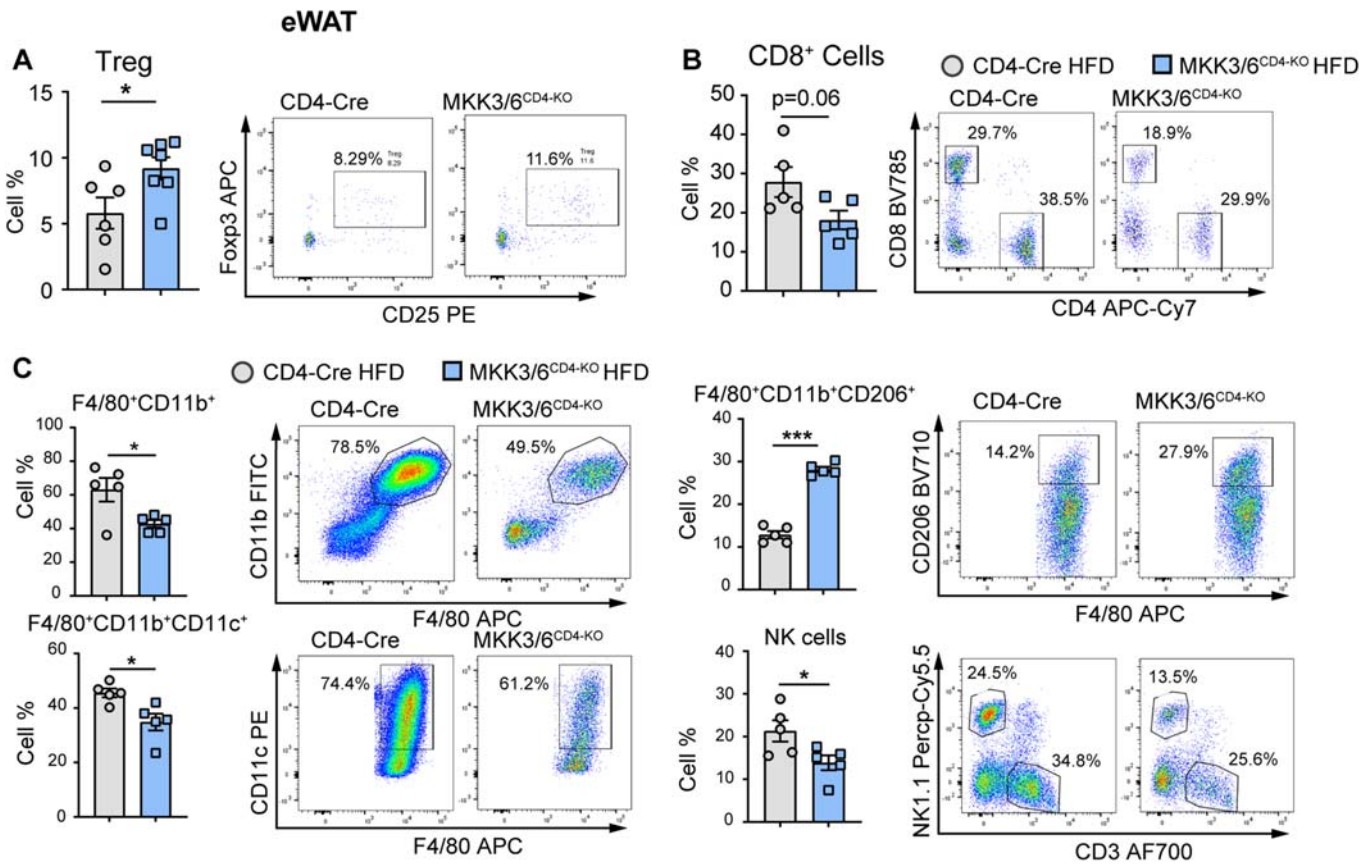

**Figure 6. MKK3/6 deficiency in T cells promotes Treg cell accumulation in adipose tissue.**

(A–C) MKK3/6$^{CD4-KO}$ and CD4-Cre mice were fed a high-fat diet (HFD) for 8 weeks. (A) FACS quantification and representative dot plots of Treg cells (CD4$^+$CD25$^+$Foxp3$^+$) in SVF and (B) CD8$^+$ T cells in SVF. (C) FACS quantification and representative dot plots of myeloid (F4/80$^+$CD11b$^+$), M1 macrophages (Mφ) (F4/80$^+$CD11b$^+$CD11c$^+$), M2 Mφ (F4/80$^+$CD11b$^+$CD206$^+$), and NK (NK1.1$^+$CD3$^-$) cells in SVF. Data Information: Data are presented as mean ± SEM, *$p < 0.05$, ***$p < 0.001$. Exact $p$-values are shown. Analysis by $t$ test or by the Welch test when variances were different. $n = 5$–7 biologically independent mice for each group, represented as single dots in the graphs (A–C). Source data are available online for this figure.

protein pathway. Furthermore, our findings suggest a potential novel role of IL35 in influencing BAT temperature. We have demonstrated that IL-35 treatment not only activates ATF-2 but also leads to increased expression of UCP1 and FGF21, coinciding with elevated BAT temperature in mice. In line with this, individuals with obesity exhibit lower expression of this cytokine in visceral fat. In summary, our findings suggest a possible role of IL-35 derived from Treg cells in controlling AT metabolism and function.

The activation of stress kinases is associated with obesity and metabolic diseases in mouse and human in a tissue-specific manner (Nikolic et al, 2020). The importance of these kinases in myeloid cells in obesity and associated comorbidities has been widely studied. For example, p38 kinases activation polarizes macrophages towards a pro-inflammatory phenotype, leading to pro-inflammatory cytokine production in a model of liver steatosis (Zhang et al, 2019) and LPS-induced acute hepatitis (Gonzalez-Teran et al, 2013). In neutrophils, p38 kinase family members control their migration and promote nonalcoholic fatty liver diseases development (Gonzalez-Teran et al, 2016b). However, our knowledge about the role of p38 kinases in T cells and their

contribution to obesity development was insufficient. By analyzing the single-cell RNA sequencing data from human WAT was obtained from Emont et al (Emont et al, 2022), we focused on the expression of p38 signaling pathway in T cells and Treg cells within AT. We observed upregulated expression of p38α (*MAPK14*), its activators MKK3 (*MAP2K3*), and p38 substrate *ATF-2* in humans with obesity (BMI > 30), while *DUSP1* phosphatase, which deactivates p38s, was downregulated, suggesting a possible obesogenic role of p38 signaling pathway in both T cells and Treg cells. Furthermore, our study results confirmed p38 activation in T cells promotes obesity, as the deficiency of the upstream activators of p38 kinases, specifically in T cells, impeded obesity development and improved metabolic profiles. Our data also indicate that p38 blockade in T cells is sufficient to blunt obesity-induced AT inflammation, thereby protecting HFD-fed MKK3/6$^{CD4-KO}$ mice against the AT metabolic dysfunction known to be a driver of obesity comorbidities. Treg cells increase in lean individuals and are thought to prevent AT inflammation. Interestingly, we found a higher number of Treg cells in the blood, lymph nodes and AT of HFD-fed MKK3/6$^{CD4-KO}$ mice, resembling the phenotype typical for lean mice (Feuerer et al, 2009).

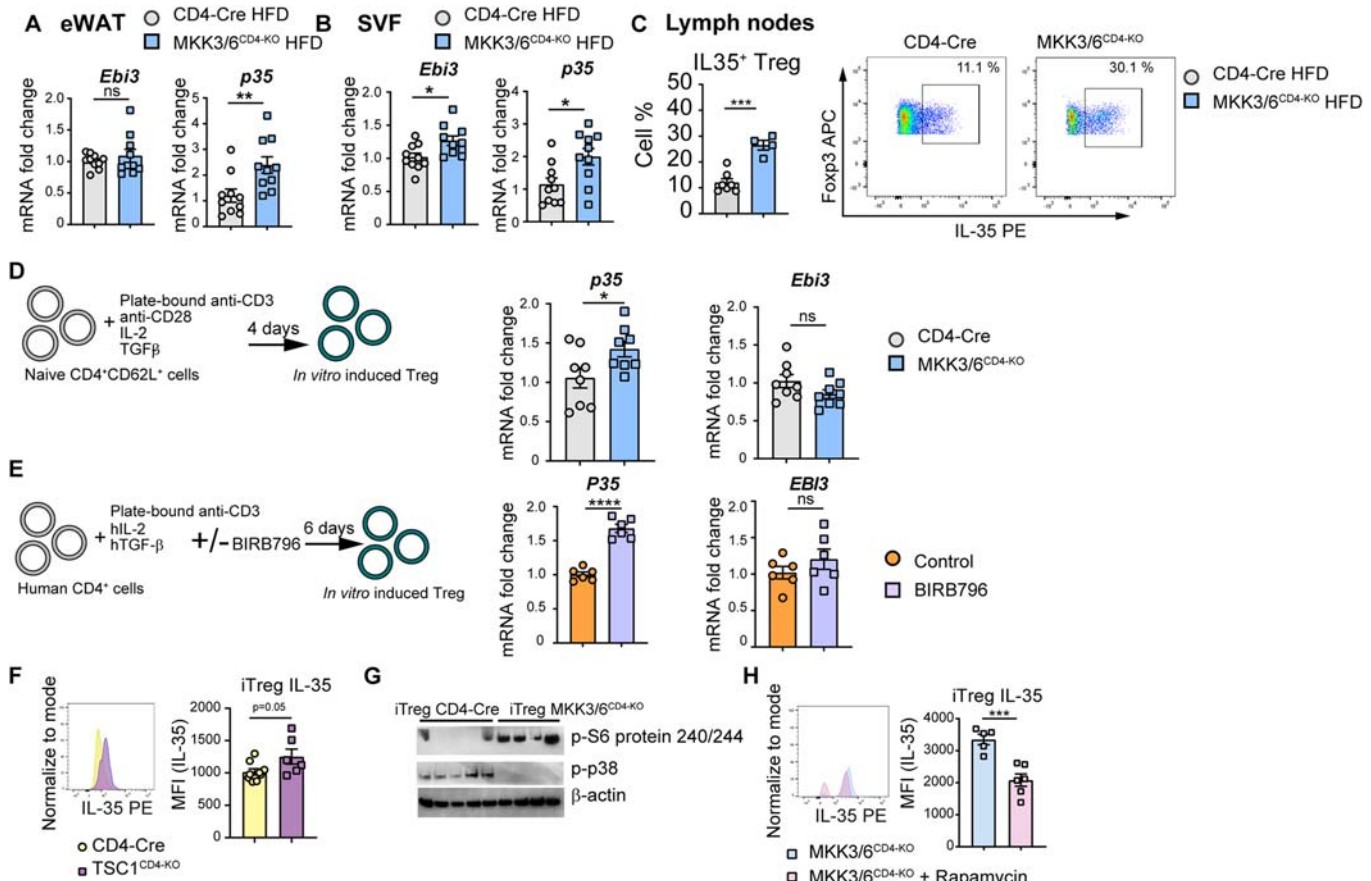

**Figure 7.  p38 activation in Treg cells inhibits IL-35 production.**

(A–C) MKK3/6[CD4-KO] and CD4-Cre mice were fed a high-fat diet (HFD) for 8 weeks. (A, B) qRT-PCR analysis of mRNA expression in (A) eWAT and (B) SVF isolated from control or MKK3/6[CD4-KO] mice. mRNA expression was normalized to *b-actin* mRNA. (C) FACS quantification and representative dot plots of IL-35[+] Treg cells in lymph nodes. (D) In vitro Treg cell induction (iTreg). Naive CD4[+] T cells were isolated from the spleens of CD4-Cre and MKK3/6[CD4-KO] mice stimulated for 96 h with plate-bound anti-CD3, soluble anti-CD28 + IL-2 + TGFβ. qRT-PCR analysis of *p35* and *Ebi3* mRNA in iTregs derived from control or MKK3/6[CD4-KO] mice. mRNA expression was normalized to *b-actin* mRNA. (E) Induction of iTregs from CD4[+] T cells isolated from healthy human donor buffy coats and stimulated with plate-bound anti-CD3, soluble hIL-2 + hTGFβ for 6 days in the presence or absence of the p38 pan inhibitor BIRB796. qRT-PCR analysis of *P35* and *EBI3* mRNA in iTregs. mRNA expression was normalized to *GAPDH* mRNA. (F) FACS analysis of IL-35 MFI in in vitro induced iTreg cells from CD4-Cre and TSC1[CD4-KO] mice. (G) Western blot analysis of p-s6 protein S240/244 and p-p38 Thr180/Tyr182 in iTreg cells from CD4-Cre and MKK3/6[CD4-KO] mice. Loading control for p-p38 was run on different gel and not presented. (H) FACS analysis of IL-35 MFI in in vitro induced iTreg cells from MKK3/6[CD4-KO] mice in the presence or absence of rapamycin for 4 h. Data Information: Data are presented as mean ± SEM, *$p < 0.05$, **$p < 0.01$, ***$p < 0.001$, ns: not significant. Exact p-values are shown. Analysis by t test. $n = 4$–10 biologically independent mice (A–C) or $n = 4$–9 biologically independent wells (D–H) for each group, represented as single dots in the graphs. Source data are available online for this figure.

Treg cells are important immunosuppressive cells which maintenance self-tolerance, immune homeostasis and limit chronic inflammation (Sakaguchi et al, 2010; Takeuchi and Nishikawa, 2016; Wing and Sakaguchi, 2010). The observed accumulation of Treg cells in HFD-fed MKK3/6[CD4-KO] mice restrained CD8[+] T cell infiltration in AT, consequently limiting AT inflammation. Namely, the infiltration of CD8[+] T cells in AT is one of the crucial steps in obesity development which precedes the activation and influx of pro-inflammatory macrophages (Nishimura et al, 2009). In agreement, we observed lower accumulation of CD8[+] T cells in mice lacking p38 activation, which reflected in decreased infiltration of pro-inflammatory macrophages and NK cells.

Recently, cytokine IL-35 was identified as a novel inhibitory cytokine produced by Treg cells, which potentiates their suppressive activity and suppresses T cell proliferation (Collison et al, 2007). Here, we found that its expression was p38-dependent, as

both chemical inhibition and genetic deficiency of p38 activation resulted in increased IL-35 expression in human and mouse Treg cells, thereby limiting AT inflammation in HFD-fed MKK3/6[CD4-KO] mice. In addition, we described a novel mechanism by which p38 kinases regulate IL-35 expression through the inhibition of the mTOR pathway.

Finally, we have demonstrated a novel role for the Treg-derived cytokine IL-35 in AT functionality. Levels of this cytokine are decreased in obese AT in both humans and mice. These results are consistent with the recently described role for another member of the same IL family, IL-27, which promotes adipocyte thermogenesis (Wang et al, 2021). Notably, the lack of p38 activation in T cells correlates with an increase in BAT temperature, protecting mice against obesity, with some limitations. One such limitation of the study is that, although we did not observe a total increase in EE in MKK3/6[CD4-KO] mice, when corrected by lean mass, MKK3/6[CD4-KO]

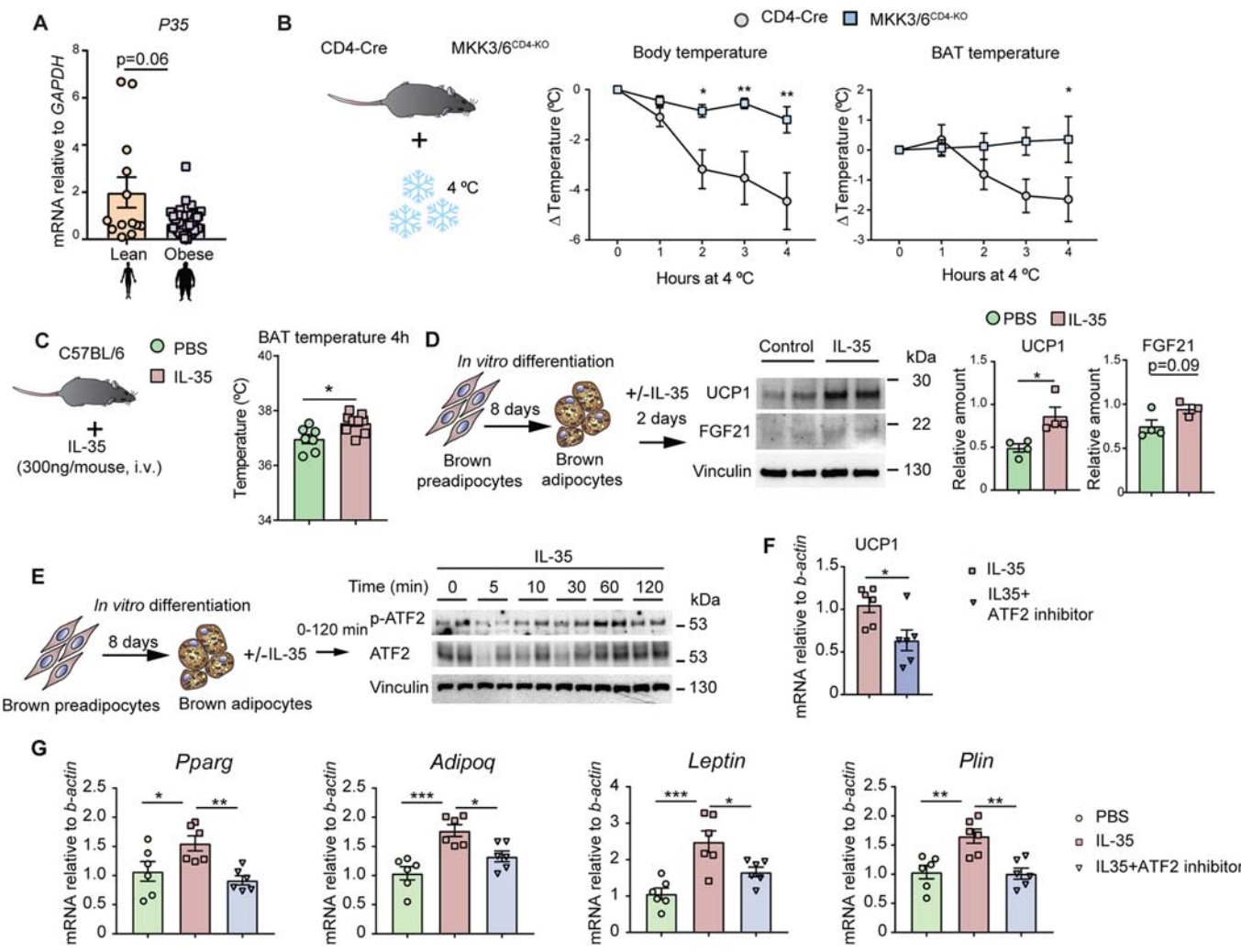

**Figure 8.  Treg-derived IL-35 promotes thermogenesis by increasing ATF-2 phosphorylation and UCP1 and FGF21 levels.**

(A) mRNA expression of p35 subunit of IL-35 in human visceral fat isolated from lean and obese patients. (B) MKK3/6[CD4-KO] and control CD4-Cre mice were exposed to cold for 4 h. Body and BAT temperature was measured every hour. (C) C57BL6 mice were treated with recombinant IL-35 i.v. (300 ng per mouse) and BAT temperature was measured 4 h later (D) Immortalized brown preadipocytes were differentiated in vitro. Once differentiated, cells were stimulated in the presence or absence of IL-35 (100 ng/ml) for 48 h and UCP1 and FGF21 levels were analyzed by immunoblot. Loading control for UCP1 was run on different gel and not presented. (E) Immortalized brown preadipocytes were differentiated in vitro. Once differentiated, cells were stimulated in the presence or absence of IL-35 (100 ng/ml) for 0–120 min and ATF2 phosphorylation was analyzed by immunoblot. (F) Differentiated adipocytes were stimulated with IL-35 (100 ng/ml) for 48 h in the presence or absence of SB203580 inhibitor (10 µM). The expression of *Ucp1* level was measured by qRT-PCR and relativized to *b-actin*. (G) Primary white preadipocytes were isolated from C57BL6 mice and differentiated in vitro. Once differentiated, cells were stimulated with PBS or with IL-35 (100 ng/ml) for 48 h in the presence or absence of SB203580 inhibitor (10 µM). The expression of principal adipogenic markers (*Pparg, Adipoq, Leptin, Perlinipin*) level was measured by qRT-PCR and relativized to *b-actin*. Data Information: Data are presented as mean ± SEM, *$p < 0.05$, **$p < 0.01$, ***$p < 0.001$. Exact *p*-values are shown. Analysis by *t* test (**A, C, D, F**), 2-way ANOVA (**B**), or 1-way ANOVA (**G**). Lean $N = 12$ biologically independent patients; Obese $N = 52$ biologically independent patients (**A**). $n = 5–9$ biologically independent mice (**B, C**) or $n = 2–6$ biologically independent wells (**D–G**) for each group, represented as single dots in the graphs. Source data are available online for this figure.

showed higher EE. Typically, heavier animals exhibit higher EE, whereas MKK3/6[CD4-KO] mice, being smaller, exhibit comparable EE to control mice, indicating that they have higher EE per gram of body weight. This protection was maintained when the mice were placed under isothermal conditions, confirming that the observed phenotype is dependent on p38 T cells. Since increased BAT thermogenesis is still present under isothermal housing conditions, this suggests that the effect of Treg cells/IL-35 on BAT thermogenesis is likely mediated by alternative mechanisms, which need to be confirmed by further studies. However, it is important to note

that we cannot rule out the contribution of alternative mechanisms controlled by the lack of p38 activation in T cells, independent of Treg-derived IL-35, which may also play a role in protecting against obesity.

Our results suggest that the higher expression of IL-35 in MKK3/6[CD4-KO] mice might be responsible for their improved adaptive thermogenesis. In fact, we found that IL-35 is able to induce adipocyte thermogenic program. Mechanistically, IL-35 triggers the phosphorylation of the transcription factor ATF-2, leading to increased UCP1 and FGF21 protein levels in adipocytes.

While previous studies using the loss of adipose Treg cells (Bapat et al, 2015; Cipolletta et al, 2012; Feuerer et al, 2009) or gain-of function Treg cells (Matsumoto et al, 2017), did not result in significant protection against obesity, our study showed that p38 signaling pathway is important in Treg cells and may modulate their function. This ultimately led to an upregulation of IL-35 expression, which could be a contributing factor to the significant obesity protection observed in MKK3/6^CD4-KO mice. We believe that IL-35's effects on energy balance and thermogenesis are critical components of the observed protection against obesity in this model.

In summary, our findings unveil the importance of p38 signaling in the regulation of T cell function and identify the manipulation of this pathway is a promising therapeutic strategy for metabolic diseases. We demonstrate that p38 kinases reduce Treg cells in circulation and AT draining lymph nodes, hence leading to Treg reduction in AT and obesity development. Moreover, we show that Treg-derived IL-35 has an important role in regulating AT browning via the ATF-2/UCP1/FGF21 axis, thereby improving metabolism and proposing this cytokine as a promising target for anti-obesity immunotherapy.

# Methods

**Reagents and tools table**

| Reagent/Resource | Reference or Source | Identifier or Catalog Number |
|---|---|---|
| **Experimental Models** | | |
| C57BL/6J background | Jackson Laboratory | Cat# 000664 RRID:IMSR_JAX:000664 |
| Tg (Cd4-cre)1Cwi/BfluJ | Jackson Laboratory | Cat# 017336 RRID:IMSR_JAX:017336 |
| Map2k3 f/f | N/A | |
| Map2k6 f/f | N/A | |
| **Recombinant DNA** | | |
| **Antibodies** | | |
| APC/Cy7 anti-mouse CD4 antibody (clone RM4-5) | BioLegend | Cat# 100525 RRID:AB_312726 |
| Brilliant Violet 785™ anti-mouse CD8a (Clone 53-6.7) | BioLegend | Cat# 100749 RRID:AB_11218801 |
| Brilliant Violet 510™ anti-mouse CD8a Antibody (Clone 53-6.7) | BioLegend | Cat# 100752 RRID: AB_2563057 |
| PE anti-mouse CD25 (Clone PC61) | BioLegend | Cat# 102007 RRID:AB_312856 |
| BV421 anti-mouse CD25 (Clone A18246A) | BioLegend | Cat# 113705 |
| PERCP-CY5.5 anti-mouse NK1.1 (Clone PK136) | BioLegend | Cat# 108728 RRID:AB_2132705 |
| Alexa fluor 700 anti-mouse CD3 (Clone 17A2) | BioLegend | Cat# 100216 RRID:AB_493697 |
| Brilliant Violet 711™ anti-mouse CD206 (MMR) (Clone C068C2) | BioLegend | Cat# 141727 RRID:AB_2565822 |
| Brilliant Violet 785™ anti-mouse/human CD11b Antibody (Clone: M1/70) | BioLegend | Cat# 101243 RRID: AB_2561373 |
| PE Hamster Anti-Mouse CD11c (Clone HL3) | BD bioscience | Cat# 553802 RRID:AB_395061 |
| V450 anti-mouse CD45 (Clone 30-F11) | BD bioscience | Cat# 560501 RRID:AB_1645275 |
| FITC Rat anti-mouse CD11b (Clone M1/70) | BD bioscience | Cat# 557396 RRID:AB_396679 |
| Alexa 647 anti-Mouse F4/80 (Clone A3-1) | BioRad | Cat# MCA497A647 RRID:AB_323931 |
| PE anti-mouse CD115 (Clon AFS98) | eBioscience | Cat #: 12-1152-82 |
| PE-Cy7 F4/80 Monoclonal Antibody (Clone BM8) | eBioscience | Cat# 25-4801-82 RRID: AB_469653 |
| APC anti-mouse FOXP3 (Clone FJK-16s) | eBioscience™,ThermoFisher | Cat# 17-5773-80 RRID:AB_469456 |
| FITC anti-mouse FOXP3 (Clone FJK-16s) | eBioscience™,ThermoFisher | Cat# 11-5773-82 RRID: AB_465243 |
| PE IL-12 p35 Monoclonal Antibody (Clone 27537) | Invitrogen, ThermoFisher | Cat# MA5-23559 RRID: AB_2609031 |
| Anti-mouse UCP1 | Abcam | Cat# AB10983 RRID:AB_2241462 |
| Anti-mouse FGF21 | BioVendor | Cat# RD281108100 |
| Anti-mouse ATF2 | Cell Signaling Technology | Cat# 9226 S |
| Anti-mouse p-s6 S240/244 | Cell Signaling Technology | Cat# 5364 S |
| Anti-mouse p-p38 T180/Y182 | Cell Signaling Technology | Cat# 9211 S |
| Anti-mouse b-actin | Santa Cruz Technology | Cat# sc-47778 |
| Anti-mouse Vinculin | Sigma | Cat# V9131 RRID:AB_477629 |
| Anti-mouse MKK6 | Enzo Life Sciences | Cat# ADI-KAP-MA014-E RRID:AB_11179962 |
| Anti-mouse MKK3b | Cell Signaling Technology | Cat# 9238 RRID:AB_2140797 |
| Polyclonal goat anti-mouse IgG (H + L) Secondary Antibody, HRP | ThermoFisher | Cat# 31430 RRID:AB_228307 |
| Polyclonal goat anti-rabbit IgG (H + L) Secondary Antibody, HRP | ThermoFisher | Cat# 31460 RRID:AB_228341 |

| Reagent/Resource | Reference or Source | Identifier or Catalog Number |
|---|---|---|
| **Oligonucleotides and other sequence-based reagents** | | |
| Primers for qRT-PCR | Sigma-Aldrich | |
| Gene | Forward | Reverse |
| m_b-actin | GGCTGTATTCCCCTCCATCG | CCAGTTGGTAACAATGCCATGT |
| m_Ucp1 | GTGAACCCGACAACTTCCGAA | TGAAACTCCGGCTGAGAAGAT |
| m_Ppargc1a | TATGGAGTGACATAGAGTGTGCT | CCACTTCAATCCACCCAGA |
| m_Cox7a1 | GCTCTGGTCCGGTCTTTTAGC | GTACTGGGAGGTCATTGTCGG |
| m_Cidea | TGACATTCATGGGATTGCAGAC | GGCCAGTTGTGATGACTAAGAC |
| m_Cox7a1 | GCTCTGGTCCGGTCTTTTAGC | GTACTGGGAGGTCATTGTCGG |
| m_Prdm16 | CCACCAGCGAGGACTTCAC | GGAGGACTCTCGTAGCTCGAA |
| m_Leptin | GAGACCCCTGTGTCGGTTC | CTGCGTGTGTGAAATGTCATT |
| m_Plin1 | ACAGCAGAATATGCCGCCAA | GGCTGACTCCTTGTCTGGTG |
| m_Pepck | CCATCACCTCCTGGAAGAACA | ACCCTCAATGGGTACTCCTTC |
| m_G6pc | CGACTCGCTATCTCCAAGTGA | GTTGAACCAGTCTCCGACCA |
| m_Acox1 | CCGCCACCTTCAATCCAGAG | CAAGTTCTCGATTTCTCGACG |
| m_Ppardelta | CTCGTACTTGAGCTTCATGCG | GAGCACACCCTTCCTTCCAG |
| m_Fasn | GCGGGTTCGTGAAACTGATAA | GCAAAATGGGCCTCCTTGATA |
| m_Acaca | GATGAACCATCTCCGTTGGC | GACCCAATTATGAATCGGGAGTG |
| m_Elovl6 | GAGCAGAGGCGCAGAGAAC | ATGCCGACCACCAAAGATAA |
| m_Scd1 | TTCTTGCGATACACTCTGGTGC | CGGGATTGAATGTTCTTGTCGT |
| m_Dgat2 | GCGCTACTTCCGAGACTAC | GGGCCTTATGCCAGGAAACT |
| m_Adipoq | TGTTCCTCTTATCCTGCCCA | CCAACCTGCACAAGTTCCCTT |
| m_Cpt1a | CTCCGCCTGAGCCATGAAG | CACCAGTGATGATGCCATTCT |
| m_Ebi3 | GCTCCCCTGGTTACACTGAA | ACGGGATACCGAGAAGCAT |
| m_p35 | TCAGAATCACAACCATCAGCA | CGCCATTATGATTCAGAGACTG |
| H_Gapdh | CCATGAGAAGTATGACAACAG | GGGTGCTAAGCAGTTGGTG |
| H_P35 | TGCCTTCACCACTCCCAAAACC | CAATCTCTTCAGAAGTGCAAGGG |
| H_EBI3 | CTGGATCCGTTACAAGCGTCAG | CACTTGGACGTAGTACCTGGCT |
| **Chemicals, Enzymes and other reagents** | | |
| Fast SYBR Green Master Mix | Applied Biosystems | Cat# 4385616 |
| Protein marker PS10 Plus (5 × 500 µl) | Attenbio | Cat# PL-5 |
| Purified NA/LE Hamster Anti-Mouse CD28 (clon 37.51) | BD Bioscience | Cat# 553294 |
| BD Pharmingen™ Purified NA/LE Hamster Anti-Mouse CD3e (clon 145-2C11) | BD Bioscience | Cat# 553057 |
| Ultra-LEAF™ Purified anti-human CD3 Antibody (clon OKT3) | Biolegend | Cat# 317326 |
| Recombinant Human TGF-β1 (carrier-free) | BioLegend | Cat# 580702 |
| Acrilamide | Bio-Rad | Cat# 161-0156 |
| Rapamycin | Biorbyt | Cat# orb154705 |
| EDTA | Calbiochem | Cat# 324503 |
| MOPS-SDS running buffer (20X), 1 L | Fisher | Cat# 15435159 |
| Amersham ECL Prime Western Blotting Detection Reagent | GE Healthcare | Cat# RPN2232 |
| Ficoll-Paque PLUS* | GE Healthcare | Cat# 17-1440-03 |
| Fetal Bovine Serum (FBS) | Gibco | Cat# 1027-106 |
| L-Glutamine | Hyclone | Cat# SH30034.01 |
| Amphotericin B (Fungizone) solution | Hyclone | Cat# SV30078.01 |
| Hepes | Hyclone | SH30237.01 |
| Non-essential amino acids | Hyclone | Cat# SH30238.01 |
| Methanol | Honeywell | Cat# 24229 |
| Tri reagent (Trizol) | Invitrogen | Cat# 15596-026 |
| Insulin Humulina 100 UI/ml | Lilly | N/A |
| Ammonium Persulfate (APS) | Merck | Cat# 101200 |

| Reagent/Resource | Reference or Source | Identifier or Catalog Number |
| --- | --- | --- |
| β-mercaptoethanol | Merck | Cat# 805740 |
| Bromophenol blue | Merck | Cat# 108122 |
| NaF | Merck | Cat# B0590249 |
| 10% NEUTRAL BUFFERED FORMALIN | Proquinorte S.A. | Cat# BAF-0010-10X |
| Recombinant Human IL-2 | Peprotech | Cat# 200-02 |
| Aprotinin | Roche | Cat# 11583794001 |
| Leupeptin | Roche | Cat# 11017128001 |
| IL-35 EBI3 Mouse Recombinant Protein | Rockland | Cat# 010-001-B66 |
| Dexamethasone | Sigma | Cat# D1756-100MG |
| 3-isobutyl-1-methylxanthine | Sigma | Cat# I5879-250MG |
| Indomethacin | Sigma | Cat# I7378-100G |
| Insulin | Sigma | Cat# I9278 |
| Ionomycin from Streptomyces conglobatus | Sigma | Cat# I9657-1MG |
| LIBERASE TL RESEARCH GRADE | Sigma | Cat# 5401020001 |
| Ponceau | Sigma | Cat# P3504-10G |
| Bovine Serum Albumin | Sigma | Cat# A7906 |
| Norepinephrine | Sigma | Cat# A7257-500MG |
| Na pyrophosphate | Sigma | Cat# 221368-100 G |
| Na orthovanadate | Sigma | Cat# S6508-10G |
| Sucrose | Sigma | Cat# S9378 |
| Sodium dodecyl sulfate (SDS) | Sigma | Cat# L5750 |
| PMSF | Sigma | Cat# P7626 |
| Phorbol 12-myristate 13-acetate (PMA) | Sigma | Cat# P8139-1MG |
| Oil-Red | Sigma | Cat# O0625-25G |
| RPMI | Sigma | Cat# 21875-034 |
| Hematoxylin | Sigma | Cat# H3136 |
| Tween-20 | Sigma | Cat# P1379-1L |
| DNase Type II-S | Sigma | Cat# D4513 |
| Penicilin-Strepomicin | Sigma | Cat# P4333 |
| T3 3,3′,5-Triiodo-L-Thyronine Free acid | Sigma | Cat# T2877-250MG |
| Temed | Sigma | Cat# T9281 |
| Tissue-Tek® optimum cutting temperature (O.C.T.) | SAKURA FINETEK USA INC | Cat# 25608-930 |
| Doramapimod (BIRB 796) | SELLECK CHEMICALS LLC | Cat# S1574 |
| Adezmapimod (SB203580) | SELLECK CHEMICALS LLC | Cat# S1076 |
| Troglitazone | Tocris Bioscience | Cat# 3114/10 |
| HBSS, calcium, magnesium, no phenol red | Thermofisher | Cat# 14025-050 |
| Eosin Y Alcoholic | Thermo Scientific | Cat# 6766008 |
| **Software** | | |
| GraphPad PRISM 7 | GraphPad Software | RRID:SCR_002798 |
| Photoshop CS6 | Adobe | RRID:SCR_014199 |
| Adobe Illustrator | Adobe | RRID:SCR_010279 |
| Seahorse Wave | Agilent | N/A |
| FlowJo | FlowJo | https://www.flowjo.com/ RRID:SCR_008520 |
| Fiji | ImageJ | RRID:SCR_002285 |
| FlirIR software | FLIR | RRID:SCR_016330 |
| METABOLISM Software | Panlab | Cat# 760817 |
| Seurat 4.0 | Satijalab | N/A |
| Sequence Detection System v2.4 | Applied Biosystems | Cat# 4350490 |

| Reagent/Resource | Reference or Source | Identifier or Catalog Number |
|---|---|---|
| **Other** | | |
| Human buffy coats | Blood Transfusion Center of Comunidad de Madrid | N/A |
| Foxp3 Transcription Factor Staining Buffer Kit | eBioscience™,ThermoFisher | Cat# A25866A |
| EasySep™ Mouse Naive CD4 + T Cell Isolation Kit | StemCell Technologies | Cat# 19765 |
| EasySep™ Human Naive CD4 + T Cell Isolation Kit II | StemCell Technologies | Cat# 17555 |
| RNa easy Mini Kit | Qiagen | Cat# 74106 |
| Zombie Aqua™ Fixable Viability Kit | BioLegend | Cat# 423101 |
| DAPI (for nucleic acid staining) | Sigma | Cat# D9542-5MG |
| RNa easy Mini Kit | Qiagen | Cat# 74106 |
| Rna easy Micro kit | Qiagen | Cat# 217084 |
| High-Capacity cDNA Reverse Transcription Kit | Applied Biosystems | Cat# 4368814 |
| Breeding & Maintenance diet for nude rats and mice and transgenic strains | Altromin | Cat# 1410 |
| Glucometer | Ascensia Breeze 2 glucose meter | N/A |
| 7900 Fast Real Time thermocycler | Applied Biosystems | |
| Glucose test strips Contour Next (50UN) | Bayer (Ascensia) | Cat# 84167836 |
| Nitrocellulose Membrane (Pkg of 1 roll, 0.2 μm, 30 cm × 3.5 m) | Bio-Rad | Cat# 162-0112 |
| Mini-PROTEAN® Tetra Vertical Electrophoresis Cell for Mini Precast Gels, 2-gel | Bio-Rad | Cat# 1658005 |
| Mini Trans-Blot Module | Bio-Rad | Cat# 1703935 |
| BD FACSymphony SORP | BD Bioscience | N/A |
| BD FACS Aria II SORP | BD Bioscience | RRID:SCR_018934 |
| 70 μM cell strainers | Corning Falcon | Cat# 352350 |
| Paraffin embedding station | Leica Microsystems | EG 1150 H |
| Leica microscope | Leica Microsystems | Leica DM2500 |
| Paraffin embedding station | Leica Microsystems | EG 1150 H |
| Tissue magnalyser | Roche | N/A |
| HFD with 60 kcal + 1.5% cholesterol | Research Diets Inc/BROGAARDEN | Cat# D11103002i |
| Metabolic cages | TSE LabMaster, TSE Systems, Germany | N/A |
| Small animal magnetic resonance scanner 7 Tesla | Varian-Agilent | N/A |

## Human visceral fat samples

For the analysis of visceral fat, the study population included 65 patients (52 adult patients with BMI > 35) recruited from patients who underwent elective bariatric surgery at the University Hospital of Salamanca. Patients were excluded if they had a history of alcohol use disorders or excessive alcohol consumption (>30 g/day in men and >20 g/day in women) or had chronic hepatitis C or B. Control subjects ($n = 12$) were recruited among patients who underwent laparoscopic cholecystectomy for gallstone disease. The patient data is presented in Table 1.

## Study approval

Human peripheral blood mononuclear cells (PBMC) were isolated from buffy coats obtained from healthy donors according to standard procedures. Buffy coats of healthy donors were received from the Blood Transfusion Center of the Comunidad de Madrid, and all donors signed their consent for the use of samples for research purposes. All procedures using primary human cells were approved by the Ethics Committee of Hospital Universitario de la Princesa. For visceral fat samples the population study was approved by the Ethics Committee of the University Hospital of

Salamanca and the Carlos III (CEI PI 09_2017-v3) with the all subjects providing written informed consent to undergo visceral fat biopsy under direct vision during surgery. Data were collected on demographic information (age, sex, and ethnicity), anthropomorphic measurements (BMI), smoking and alcohol history, coexisting medical conditions, and medication use. All animal procedures conformed to EU Directive 86/609/EEC and Recommendation 2007/526/EC regarding the protection of animals used for experimental and other scientific purposes, enacted under Spanish law 1 1201/2005. The protocols are CNIC-07/18 and PROEX 215/18.

## Isolation, culture, and stimulation of human peripheral blood CD4+ T Cells

Human peripheral blood mononuclear cells (PBMC) were collected from buffy coats from healthy donors by Ficoll density gradient separation, as described (Schmidt et al, 2016). CD4+ cells were isolated using the EasySep™ Human Naive CD4+ T Cell Isolation Kit II. For in vitro Treg differentiation, isolated cells were activated with plate-bound anti-CD3 (3 μg/ml) and RPMI 1640 medium containing 10% FBS, 1% penicillin/streptomycin, 50 μM 2-mercaptoethanol, 2 mM L-glutamine, 1% non-essential AA, 1% anti-

**Table 1. Characteristics of obese patients and controls for visceral fat analysis.**

| Variable | Obese patients ($n = 52$) | Controls ($n = 12$) | $p$ |
|---|---|---|---|
| BMI (kg/m²) | 49.39 (7.39) | 25.71 (3.64) | <0.0001 |
| Body fat (%) | 53.83 (4.13) | 32.26 (8.30) | <0.0001 |
| Fasting blood sugar (mg/dL) | 114.04 (47.91) | 94.73 (10.39) | 0.191 |
| Insulin (pmol/L) | 978.49 (1722.735) | 371.90 (433.92) | 0.488 |
| HOMA-IR | 8.46 (25.09) | 2.75 (3.64) | 0.478 |
| AST (IU/L) | 24.61 (12.46) | 21.45 (5.61) | 0.415 |
| ALT (IU/L) | 32.60 (18.29) | 24.73 (11.82) | 0.178 |
| Total cholesterol (mg/dL) | 193.80 (39.05) | 199.5 (43.82) | 0.683 |
| Triglycerides (mg/dL) | 156.24 (88.95) | 147.39 (56.12) | 0.765 |
| LDL-cholesterol (mg/dL) | 113.81 (38.126) | 121 (35.366) | 0.604 |
| HDL-cholesterol (mg/dL) | 47.12 (13.11) | 48.61 (18.84) | 0.773 |

Variables are presented as mean (standard deviation) and analyzed by *t*-test.
*BMI* body mass index, *AST* aspartate aminotransferase, *ALT* alanine aminotransferase.

mycotic, 2.5 ng/ml TGF-b, and 50 U/ml IL-2 with or without 10 μM BIRB796 for 6 days in a humidified atmosphere (5% $CO_2$, 95% air) at 37 °C.

## Mice

Floxed mutant mice for *Map2k6* (*Mkk6*) genes were as described (Matesanz et al, 2017). Mice with a germ-line mutation in the *Map2k3* gene and LoxP elements inserted into two introns (*Map2k3*LoxP) were generated after homologous recombination in ES cells, obtained from EUCOMM clon EPD0160_3_H09. The ES cell clones were injected into C57BL/6J blastocysts to create chimeric mice that transmitted the mutated *Map2k3* allele through the germ line. The Flp NeoR cassette was excised by crossing these mice with ACTB:FLPe B6;SJL mice, which express a FLP1 recombinase gene under the direction of the human ACTB promoter. These animals were crossed with Tg (CD4-cre)1Cwi/BfluJ mice on the C57BL/6J background (Jackson Laboratory) to generate mice lacking MKK3 and MKK6 in T cells (MKK3/6[CD4-KO]). TSC1[CD4-KO] (Tsc1_lox (lox/lox) Tg.Cd4-Cre) were from Dr. Alejo Efeyan (CNIO). All mice were maintained on a C57BL/6J background (back-crossed for 10 generations). Genotype was confirmed by PCR analysis of genomic DNA. Mice were fed an ND or an HFD (Research Diets Inc.) for 2 weeks or 8–10 weeks ad libitum. As a control, we used age-matching either CD4-Cre mice or MKK3/6[f/f] (Cre negative) mice which is clearly indicated in results and figure legends. Mice were randomly assigned to cages, with inclusion or exclusion criteria based on instances of animal fights or harm during the study. Mice were housed at temperature standard for our animal facility (23–25 °C). For thermoneutrality experiments, mice were housed them at 30 °C and fed HFD for 4 or

8 weeks (until the end of the experiment). All analysts were blinded to experimental group or treatment.

## BAT temperature measurement

BAT-adjacent interscapular temperature was quantified from thermographic images captured with a FLIR T430sc Infrared Camera (FLIR Systems, Inc., Wilsonville, OR) and analyzed with FLIR software.

## NE administration

MKK3/6[CD4-KO] and MKK3/6[f/f] mice were housed at 30 °C and fed HFD for 4 weeks and then injected norepinephrine (NE 1 mg/kg of BW, i.p., Sigma-Aldrich). BAT-adjacent interscapular temperature was measured before the NE administration and 10 and 20 min after NE and quantified with FLIR software. In between capturing thermogenic images, mice were kept at 30 °C to avoid that room temperature affects BAT-adjacent interscapular temperature.

## IL-35 administration

WT (C57BL6) mice were injected with IL-35 (300 ng/mouse, 010-001-B66, Rockland Immunochemicals, Inc.) or PBS i.v. in the retro-orbital sinus. 4 h later BAT-adjacent interscapular temperature was measured and quantified.

## Isolation, culture, and in vitro induction of murine Treg cells

For in vitro Treg differentiation, naive CD4+ T cells were isolated from the spleens of CD4-Cre, MKK3/6[CD4-KO], or TSC1[CD4-KO] mice using the EasySep™ Mouse Naive CD4+ T Cell Isolation Kit, STEMCELL Technologies. Cells were plated on plates previously coated with 2 μg/mL anti-CD3 antibody and incubated with 2 μg/mL anti-CD28, 2 ng/mL TGF-β1 and 20 ng/mL IL-2 in RPMI1640 supplemented with 10% FBS, 1% penicillin/streptomycin, 50 μM 2-mercaptoethanol, 2 mM L-glutamine, and 1% anti-mycotic for 96 h in a humidified atmosphere (5% $CO_2$, 95% air) at 37 °C. At the end of the experiment, iTreg cells were collected for RNA and protein isolation or treated with rapamycin (100 nm) for 4 h and used for FACS analysis.

## Real-time bioenergetic measurements in response to activation

We measured real-time naive CD4+ T cells activation measurements by Seahorse XFe96 as described (van der Windt et al, 2016). In brief, naive CD4+ T cells were isolated from the spleens of CD4-Cre and MKK3/6[CD4-KO] mice using the EasySep™ Mouse Naive CD4+ T Cell Isolation Kit, STEMCELL Technologies. Cells were plated on Seahorse XFe96 plates previously coated with 50 μg/mL poly-L-lysine (Sigma) in DMEM containing 25 mM glucose, 2 mM L-glutamine and 1 mM sodium pyruvate and placed 1 h in non-$CO_2$ incubator at 37 °C. Cells were stimulated with 50 ng/ml of PMA and 500 ng/ml of Ionomycin which was injected after third measurement (the protocol was 3 min mix–2 min wait–3 min measure) and analyzed using Seahorse Wave software (Agilent).

## Adipocyte in vitro differentiation and IL-35 stimulation

Immortalized brown preadipocytes from WT mice (mycoplasma free cells) were differentiated to brown adipocytes in 10% FCS medium supplemented with 20 nM insulin, 1 nM T3, 125 µM indomethacin, 2 µg/ml dexamethasone, and 50 mM IBMX for 48 h and maintained with 20 nM of insulin and 1 nM of T3 for 8 days. Primary white preadipocytes were isolated from C57BL/6 mice were differentiated to adipocytes for 9 days in 8% FCS medium supplemented with 5 µg/ml insulin, 25 µg/ml IBMX, 1 µg/ml dexamethasone, and 1 µM troglitazone (we changed medium every other day and supplemented with insulin and troglitazone, or the last changes were with insulin only). Differentiated adipocytes were stimulated in the presence or absence of 100 ng/ml IL-35 (010-001-B66, Rockland) for 0–2 h or for 48 h and then the cells were lysed for western blot. In some experiments we used 10 µM SB20253580 for 48 h together with IL-35.

## Glucose measurement

Mice were starved overnight (fasted) or for 1 h (fed) and blood glucose levels were quantified with an Ascensia Breeze 2 glucose meter.

## GTT

Overnight-starved mice were injected intraperitoneally with 1 g/kg of body weight of glucose, and blood glucose levels were quantified with an Ascensia Breeze 2 glucose meter at 0, 15, 30, 60, 90, and 120 min post injection.

## ITT

ITT was performed by injecting intraperitoneally 0.75 IU/kg of insulin (Lilly) at mice starved for 1 h and detecting blood glucose levels with a glucometer at 0, 15, 30, 60, and 90 min post injection.

## Indirect calorimetry system

Energy expenditure, respiratory exchange, locomotor activity, and food intake were quantified using the indirect calorimetry system (TSE LabMaster, TSE Systems, Germany) for 3 days.

## Measurement of fecal lipids

Fecal lipids were extracted from dried feces (300 mg) collected from individually housed mice during 5 days according to Kraus D et al (Kraus et al, 2015). In brief, collected feces were dried overnight at 55 °C, and lipids were extracted by 2 ml chloroform : methanol (2:1) and using tissue magnalyser (Roche) followed by centrifugation 1000 × g for 10 min in plastic 15-ml falcon tubes. The lower phase containing lipids with chloroform and methanol were collected by carefully insertion of a 21 G needle through the tube wall into the lower, lipid phases and placed in a new 2 ml tubes (previously measured the weight of empty tubes). The tubes were left for 72 h at 37 °C until all liquid was evaporated and weighed again. The lipid content was measured by subtracting the weight of empty tubes to lipid mass per 300 mg of feces.

## Nuclear magnetic resonance imaging analysis (MRI)

Body, fat, and lean mass were measured by nuclear magnetic resonance (Varian-Agilent, MA, USA) and analyzed with Fiji software (Image J).

## Histology

Fresh livers, eWAT, sWAT, and BAT were fixed in 10% formalin, included in paraffin, and cut into 5 µm slides followed by haematoxylin–eosin staining. Fat droplets in liver were detected by oil-red staining (0.7% in propylenglycol) in 8 µm liver slides included in OCT compound (Tissue-Tek).

## Tissue processing for flow cytometry

At the end of experiments, mouse axillar and inguinal lymph nodes, blood (100 µl), and spleens were collected, and single-cell suspensions were obtained by passing through a 70-µm cell strainer. Erythrocytes in pellets from all tissues (except lymph nodes) were lysed with a red cell lysis buffer, and the remaining cells were subsequently resuspended in flow cytometry buffer (2 PBS + 1% FBS + 2 mM EDTA). Epidydimal white adipose tissue (eWAT) was carefully excised, minced, and digested with 1 mg/mL liberase + 2 U/ml DNAse in HBSS for 25-30 min at 37 °C with shaking at 1200 rpm. Digestion was stopped by addition of PBS + 10% FBS, and cells were passed through a 70-µm cell strainer and centrifuged for 5 min at 500 × g to obtain the stromal-vascular fraction (SVF). SVF pellets were also used for RNA isolation and subsequent qRT-PCR analysis.

## Flow cytometry and cell sorting

Single-cell suspensions were stained with surface antibodies, and dead cells were excluded by nuclear staining with DAPI or Zombi Aqua™. Fluorochrome-conjugated antibodies to surface proteins were as follows: anti-CD45 V450 (clone 30-F11, BD Bioscience, Cat# 560501), anti-CD3 AF700 (clone 17A2, Biolegend, Cat# 141727), anti-CD4 APC-Cy7 (clone RM4-5, Biolgened, Cat# 100525), anti-CD8 BV785 (clone 53-6.7, Biolegend, Cat# 100749), anti-CD8 BV510 (clone: 53-6.7), anti-CD25 PE (clone PC61, Biolegend, Cat# 102007), anti-CD25 BV421 (clone: A18246A, Biolegend, Cat# 113705), anti-CD206 BV711 (clone C068C2, Biolegend, Cat# 141727), anti-F4/80 AF647 (clone Cl:A3-1, BioRad, Cat# MCA497A647), anti-F4/80 PE-Cy7 (clone: BM8, Thermofisher, Cat# 25-4801-82), anti-CD11b FITC (clone M1/70, BD Bioscience, Cat# 557396), anti-CD11b BV785 (clone: M1/70, Biolegend, Cat# 101243), anti-CD11c PE (clone HL3, BD Bioscience, Cat# 553802), anti-IL-35 PE (clone 27537, Thermofisher, Cat# MA5-23559), and anti-NK1.1 PerCP-Cy5.5 (clone PK136, Biolegend, Cat# 108728). For intranuclear staining, cells were fixed and permeabilized with the eBioscience Foxp3 Transcription Factor Staining Buffer Kit followed by anti-Foxp3 APC (clone FJK-16s) or anti-Foxp3 FITC (clone FJK-16s) staining. For intracellular staining of IL-35, cells were activated with PMA/ionomycin/Brefeldin A for hours before surface staining. After staining, cells were passed through 70 µm filters, and data were acquired with a BD FACSymphony flow cytometer and analyzed

with FlowJo software. The gating strategy is presented in Appendix Fig. S4A,B.

Single-cell suspensions of splenocytes from mice were stained with the following surface antibodies: anti-CD4 APC-Cy7 (clone RM4-5), anti-CD8 BV785 (clone 53-6.7), anti-CD3 AF700 (clone 17A2), anti-CD11b FITC (clone M1/70), and anti-CD45 V450 (clone 30-F11). Nuclei were stained with DAPI to distinguish live and dead cells. Cells were sorted with a BD FACS Aria cell sorter into the following populations: CD4$^+$ T cells (CD45$^+$CD3$^+$CD4$^+$CD8$^-$), CD8$^+$ T cells (CD45$^+$CD3$^+$CD8$^+$CD4$^-$) and NK cells (CD45$^+$NK1.1$^+$CD3$^-$). Sorted cells were collected, lysed, and analyzed by western blot to check MKK3 and MKK6 deletion.

## Western blot

Samples were lysed in Triton lysis buffer [20 mM Tris (pH 7.4), 1% Triton X-100, 10% glycerol, 137 mM NaCl, 2 mM EDTA, 25 mM β-glycerophosphate, 1 mM sodium orthovanadate, 1 mM phenyl-methylsulphonyl fluoride, and 10 μg/mL aprotinin, and leupeptin]. Extracts (20–50 μg protein) were examined by immunoblot. Primary antibodies used in the study: anti-mouse MKK3 (Cat# 9238, Cell Signaling Technology), anti-mouse MKK6 (Cat# ADI-KAP-MA014-E, Enzo Life Sciences), anti-mouse UCP1 (Cat# AB10983, Abcam), anti-mouse FGF21 (Cat# RD281108100, BioVendor), anti-mouse p-ATF2 T69/71 (Cat# 9225S, Cell Signaling Technology), anti-mouse ATF2 (Cat# 9226S, Cell Signaling Technology), anti-mouse p-s6 S240/244 (Cat# 5364S, Cell Signaling Technology), anti-mouse p-p38 T180/Y182 (Cat# 9211S, Cell Signaling Technology), anti-mouse b-actin (Cat# sc-47778, Santa Cruz Technology), anti-mouse vinculin (Cat# V9131, Sigma) and secondary antibodies used in the study: goat anti-mouse (Cat# 31430, ThermoFisher) and goat anti-rabbit (Cat# 31460, Thermo-Fisher). Reactive bands were detected by chemiluminescence.

## qRT-PCR

RNA (1 μg) extracted with the RNeasy Plus Mini kit or RNeasy Plus Micro kit (Quiagen) was transcribed to cDNA, and qRT-PCR was performed using the Fast Sybr Green probe (Applied Biosystems) and the appropriate primers in a 7900 Fast Real Time thermocycler (Applied Biosystems). Data were analyzed with SDS2.4 software (Applied Biosystems), and relative mRNA expression was normalized to *GAPDH* (human data) or *b-actin* (mouse data) mRNA measured in each sample. Primers used are listed in the reagents table.

## Single-cell RNA-Seq processing

Single-cell RNA sequencing data from human WAT was obtained from Emont et al (Emont et al, 2022) (https://gitlab.com/rosen-lab/white-adipose-atlas). The Seurat RDS file for immune cells was downloaded and processed with Seurat 4.0 (Hao et al, 2021) (https://satijalab.org/seurat/). The clusters of T cells and Treg cells were subset and exported for further analysis. As the clusters were already identified, integration was not re-run, but we applied a linear transformation to scale the data from the T cells and Treg cells isolated cluster, followed by dimension reduction analysis

(PCA). Following PCA analysis, the first 20 and 30 dimensions were used for further analysis in the T cells and Treg cells, respectively. We performed non-linear dimension reduction analysis using the UMAP algorithm. The analysis was done using the first 20 and 30 dimensions for the T cells and Treg cells clusters, respectively, and a resolution of 0.5. Differentially expressed genes between BMI ranges were identified with a non-parametric Wilcoxon rank sum test.

## Statistical analysis

Results are expressed as mean ± SEM. Statistical differences were analyzed by the Student *t* test or 2-way ANOVA, with differences at $p < 0.05$ considered significant. When variances were different, ANOVA coupled with Bonferroni's post-tests. When variances were different in *t* test, we used Welch's test. All analyses were performed with Excel (Microsoft Corp.) and GraphPad PRISM 7 software. Statistical details for individual the experiments were indicated in the figure legends.

# Data availability

No primary datasets have been generated and deposited.

The source data of this paper are collected in the following database record: biostudies:S-SCDT-10_1038-S44319-024-00149-y.

# Peer review information

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

## Acknowledgements

We thank S Bartlett for English editing and F Sanchez Madrid for providing
human buffy coats. We thank the staff at the CNIC Cellomics and Advanced
Imaging units for technical support and help with data analysis. IN was funded
by EFSD/Lilly grants (2017 and 2019), the CNIC IPP FP7 Marie Curie
Programme (PCOFUND-2012-600396), an EFSD Rising Star award (2019), and
grant MINECO IJC2018-035390-I. IR FPI-MCIN PRE2020-092784. MC was an
FPI-MINECO fellow (BES-2017–079711). RRB is a fellow of the FPU Program
(FPU17/03847). ML was supported by Spanish grant MINECO-FEDER
SAF2015-74112-JIN and Fundación AECC: INVES20026LEIV. ABP-G MG was
awarded with FPI BES-2017 − 081381. JV was supported by MICIU/AEI/
10.13039/501100011033 and by European Union (grants PID2021-122348NB-
I00, PLEC2022-009298, PLEC2022-009235 and EQC2021-007053-P), by the
Comunidad de Madrid (S2022/BMD-7333-CM), and by "La Caixa" Foundation
(LCF/PR/HR22/52420019). MM has been funded by Instituto de Salud Carlos
III (ISCIII) through the project PI20/00743 and INT21/00065 to MM and co-
funded by the European Union and by Junta de Castilla y León, Spain through
projects GRS 2388/A/21 and GRS 2648/A/22 to MM. PM is supported by
MICIN-ISCIII-Fondo de Investigación Sanitaria (PI22/01759; PMPTA22/
00090-BIOCARDIOTOX) and Comunidad de Madrid (P2022/BMD-7209-
INTEGRAMUNE-CM; Spain). GS is a EMBO YIP member, received funding
from the following programs and organizations: MCIN PGC2018-097019-B-
I00; European Union Seventh Framework Programme (FP7/2007-2013) under
grant agreement ERC 260464; the EFSD/Lilly European Diabetes Research
Programme; Fundación AECC PROYE19047SABI; BBVA Foundation Leonardo
Grants Program for Researchers and Cultural Creators (Investigadores-BBVA-
2017) IN[17]_BBM_BAS_0066; MINECO-FEDER SAF2016-79126-R and
PID2019-104399RB-I00, MICIN-FEDER PID2022-138525OB-I00 2023-26; and
the Comunidad de Madrid (IMMUNOTHERCAN-CM S2010/BMD-2326 and
B2017/BMD-3733), PMP21/00057. GS. MM, JV, JT has been awarded
with Infraestructura de Medicina de Precisión asociada a la Ciencia y
Tecnología IMPACT-2021 PMP21/00113. Instituto de Salud Carlos III.,
PDC2021-121147-I00.Convocatoria: Proyectos Prueba de Concepto 2021.

Ministerio de Ciencia e Innovación. The CNIC is supported by the Instituto de
Salud Carlos III (ISCIII), the Ministerio de Ciencia e Innovación (MCIN) and the
Pro CNIC Foundation), and is a Severo Ochoa Center of Excellence (grant
CEX2020-001041-S funded by MICIN/AEI/10.13039/501100011033).

## Author contributions

**Ivana Nikolic**: Conceptualization; Resources; Data curation; Formal analysis;
Supervision; Funding acquisition; Validation; Investigation; Visualization;
Methodology; Writing—original draft; Writing—review and editing. **Irene Ruiz-
Garrido**: Resources; Data curation; Formal analysis; Supervision; Validation;
Investigation; Visualization; Methodology; Writing—original draft; Writing—
review and editing. **María Crespo**: Data curation; Formal analysis; Supervision;
Validation; Investigation; Visualization; Methodology; Writing—review and
editing. **Rafael Romero-Becerra**: Data curation; Formal analysis; Supervision;
Validation; Investigation; Visualization; Methodology; Writing—review and
editing. **Luis Leiva-Vega**: Investigation; Methodology; Writing—review and
editing. **Alfonso Mora**: Data curation; Formal analysis; Supervision; Validation;
Investigation; Visualization; Methodology; Writing—review and editing.
**Marta León**: Investigation; Methodology; Writing—review and editing.
**Elena Rodríguez**: Investigation; Methodology; Writing—review and editing.
**Magdalena Leiva**: Data curation; Formal analysis; Supervision; Validation;
Investigation; Visualization; Methodology; Writing—review and editing.
**Ana Belén Plata-Gómez**: Resources; Data curation; Formal analysis;
Supervision; Validation; Investigation; Visualization; Methodology; Writing—
review and editing. **Maria Beatriz Alvarez Flores**: Resources; Formal analysis;
Validation; Investigation; Visualization; Methodology; Writing—review and
editing. **Jorge L Torres**: Resources; Data curation; Formal analysis; Supervision;
Validation; Investigation; Visualization; Methodology; Writing—review and
editing. **Lourdes Hernández-Cosido**: Resources; Data curation; Formal analysis;
Supervision; Validation; Investigation; Visualization; Methodology; Writing—
review and editing. **Juan Antonio López**: Resources; Data curation; Formal
analysis; Supervision; Validation; Visualization; Methodology; Writing—review
and editing. **Jesús Vázquez**: Resources; Data curation; Formal analysis;
Supervision; Funding acquisition; Validation; Investigation; Visualization;
Methodology; Writing—review and editing. **Alejo Efeyan**: Resources; Data
curation; Formal analysis; Supervision; Funding acquisition; Validation;
Investigation; Visualization; Methodology; Writing—review and editing.
**Pilar Martin**: Resources; Formal analysis; Supervision; Funding acquisition;
Investigation; Visualization; Methodology; Writing—review and editing.
**Miguel Marcos**: Resources; Data curation; Formal analysis; Supervision;
Funding acquisition; Validation; Investigation; Visualization; Methodology;
Project administration; Writing—review and editing. **Guadalupe Sabio**:
Conceptualization; Resources; Data curation; Formal analysis; Supervision;
Funding acquisition; Validation; Investigation; Visualization; Methodology;
Writing—original draft; Project administration; Writing—review and editing.

Source data underlying figure panels in this paper may have individual
authorship assigned. Where available, figure panel/source data authorship is
listed in the following database record: biostudies:S-SCDT-10_1038-S44319-
024-00149-y.

## Disclosure and competing interests statement

The authors declare no competing interests.

# Expanded View Figures

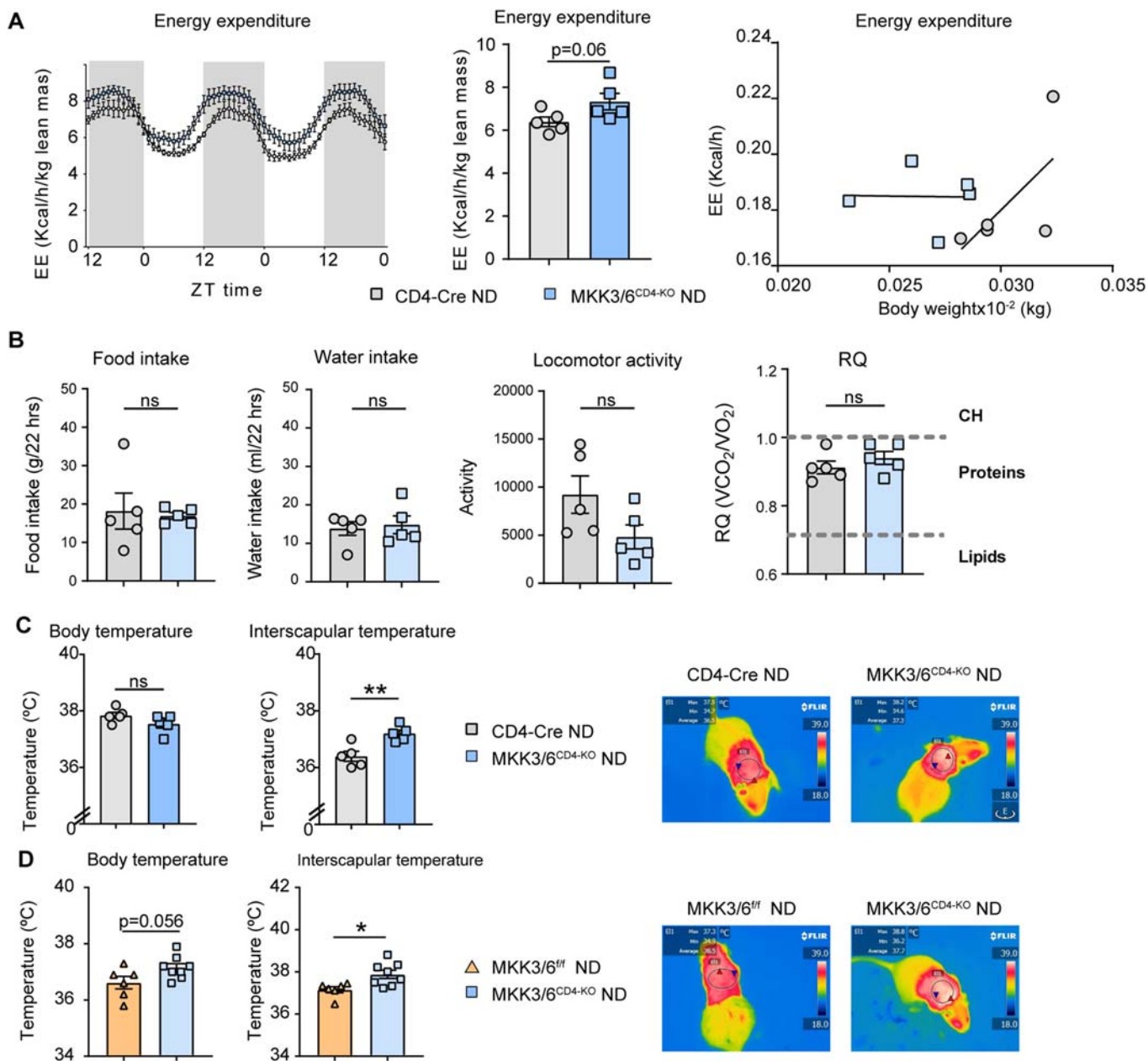

**Figure EV1. MKK3/6 deficiency in T cells increases energy expenditure and BAT temperature.**

(A) Comparison of energy balance between ND-fed MKK3/6[CD4-KO] and CD4-Cre mice examined in a metabolic cage over a 3-day period. Hour-by-hour lean–mass-corrected variation in energy expenditure (EE) (left panel); mean lean-mass-corrected EE (middle panel); and ANCOVA analysis of EE (kcal/h) (right panel). (B) Food and water intake, locomotor activity, and respiratory quotient obtained from metabolic cages. (C, D) Body temperature and skin temperature surrounding interscapular BAT. Right panels show representative infrared thermal images in (C) CD4-Cre and MKK3/6 [CD4-KO] mice and (D) in littermates (MKK3/6[f/f]) and MKK3/6[CD4-KO] mice fed chow diet. Data information: Data are presented as mean ± SEM, *$p < 0.05$, **$p < 0.01$, ns: not significant. Exact $p$-values are shown. Analysis by 2-way ANOVA coupled to the Sidak's multiple comparison post-test (A) or by $t$ test or by the Welch test when variances were different (B–D). $n = 5$–8 biologically independent mice for each group, represented as single dots in the graphs (A–D). Source data are available online for this figure.

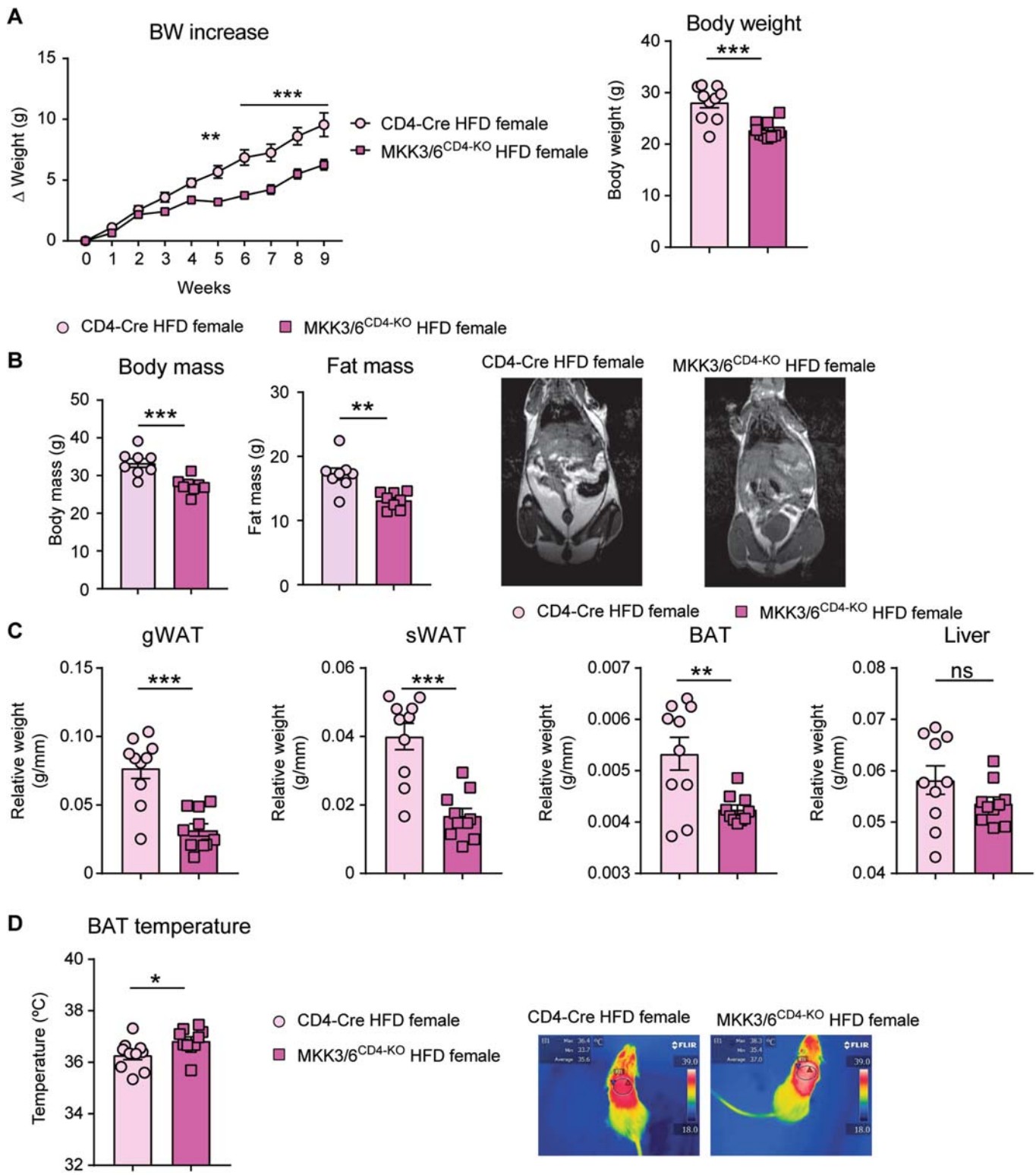

Figure EV2.   MKK3/6 deficiency in T cells protects females against HFD-induced obesity.

(A–D) Female MKK3/6^CD4-KO and CD4-Cre mice were fed a high-fat diet (HFD) for 9 weeks (starting at 8–10 weeks old). (A) Body weight evolution in MKK3/6^CD4-KO and CD4-Cre female mice for 9 weeks. Data are presented as the increase above initial weight (left) and absolute weight at the end of the experiment (right). (B) MRI analysis of body and fat mass in MKK3/6^CD4-KO and CD4-Cre mice after 8 weeks of HFD. Representative images are shown on the right. (C) eWAT, sWAT, BAT, and liver mass relative to tibia length. (D) Skin temperature surrounding interscapular BAT. Right panels show representative infrared thermal images. Data Information: Data are presented as mean ± SEM, *$p < 0.05$, **$p < 0.01$, ***$p < 0.001$, ns: not significant. Analysis by 2-way ANOVA coupled to the Bonferroni post-test (A) or $t$ test or by the Welch test when variances were different (A–D). $n = 8$–10 biologically independent mice for each group, represented as single dots in the graphs (A–D). Source data are available online for this figure.

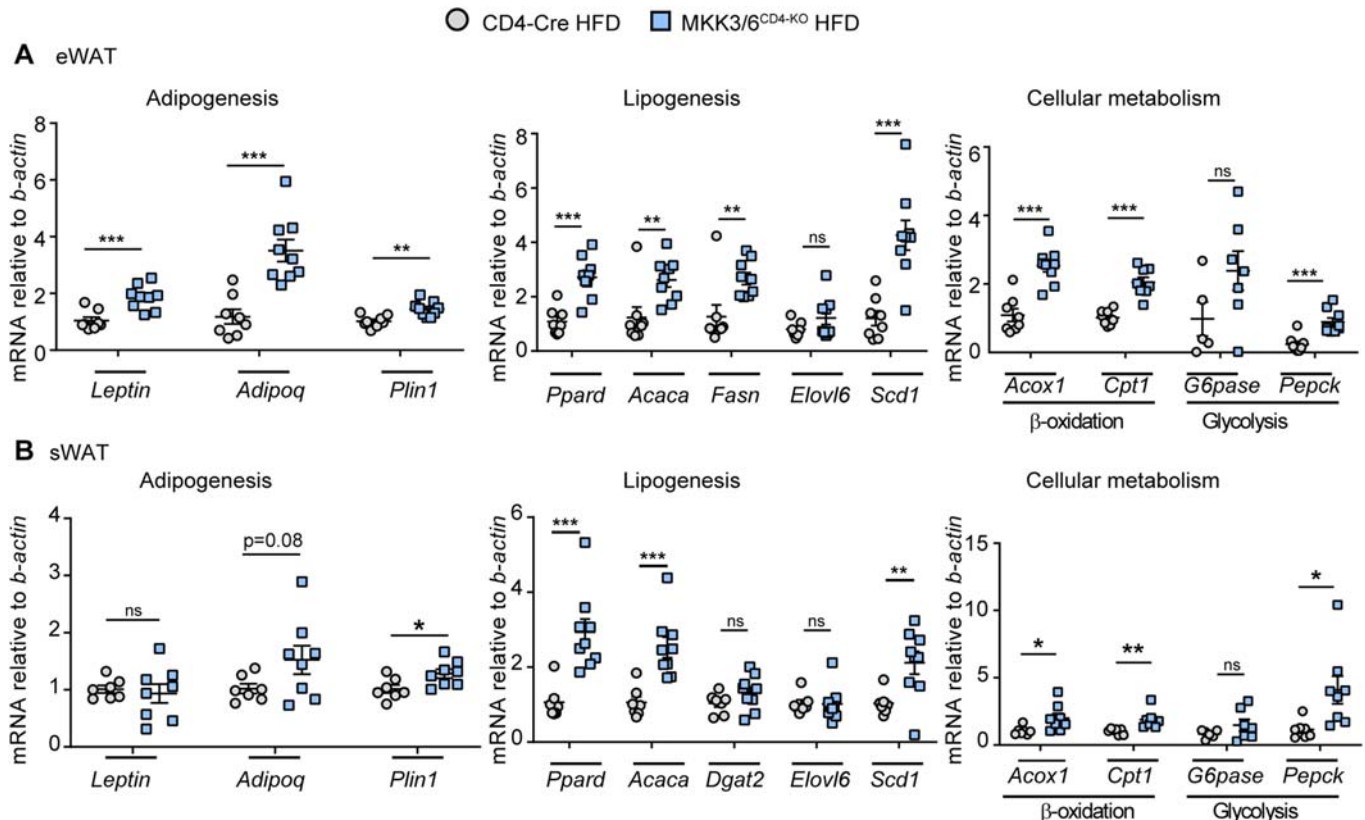

**Figure EV3. Lack of MKK3/6 in T improves adipose tissue metabolic homeostasis.**

(A, B) MKK3/6$^{CD4-KO}$ and control CD4-Cre mice were fed an HFD for 8 weeks. qRT-PCR analysis of adipogenic, lipogenic, β-oxidation, and glycolytic genes mRNA expression from (A) eWAT and (B) sWAT isolated from control CD4-Cre or MKK3/6$^{CD4-KO}$ mice. mRNA expression was normalized to the amount of *b-actin* mRNA. Data Information: Data are presented as mean ± SEM, *$p < 0.05$, **$p < 0.01$, ***$p < 0.001$, ns: not significant. Exact *p* values are shown. Analysis by *t* test or Welch's test when variances were different. $n = 8$-9 biologically independent mice for each group, represented as single dots in the graphs. Source data are available online for this figure.

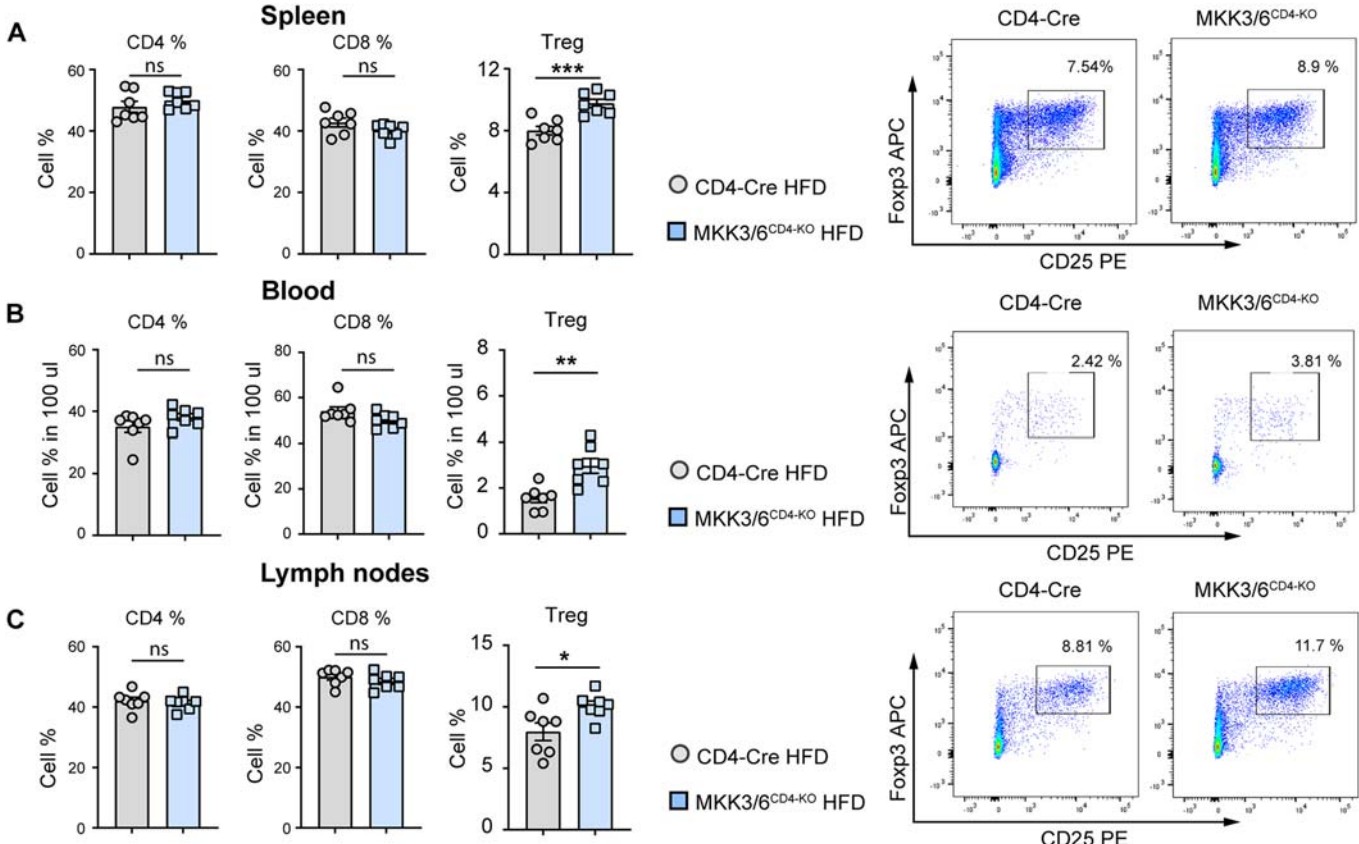

**Figure EV4. MKK3/6 deletion in T cells increases Treg cell population in blood and lymph nodes.**

(A–C) MKK3/6$^{CD4-KO}$ and CD4-Cre mice were fed a high-fat diet (HFD) for 8 weeks. FACS quantification and representative dot plots of CD4$^+$, CD8$^+$ and Treg cells (CD4$^+$CD25$^+$Foxp3$^+$) in spleen (A), blood (B), and lymph nodes (C). Data Information: Data are presented as mean ± SEM, *$p < 0.05$, **$p < 0.01$, ***$p < 0.001$, ns: not significant. Analysis by $t$ test or Welch's test when variances were different. $n = 7$ biologically independent mice for each group, represented as single dots in the graphs. Source data are available online for this figure.

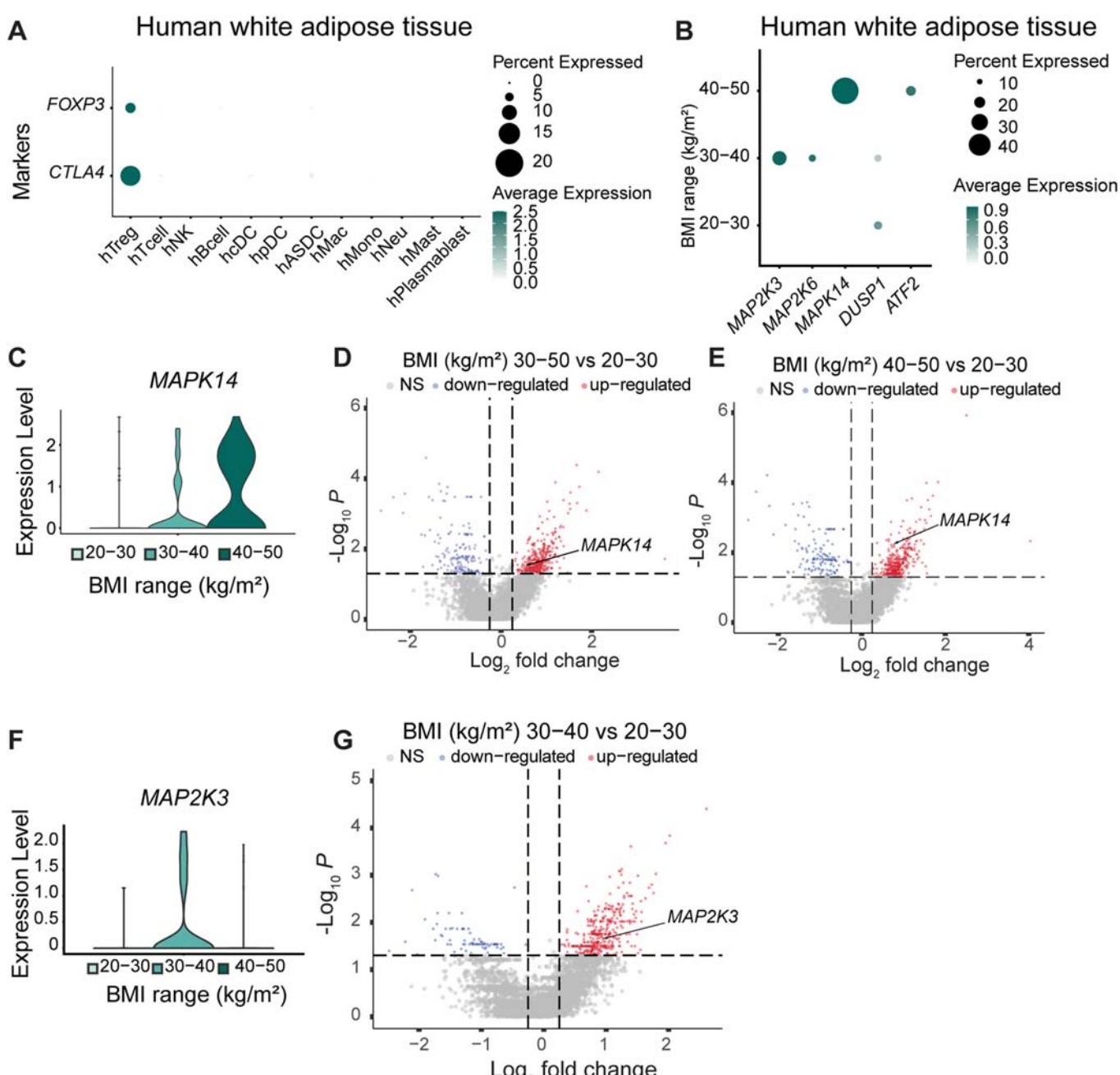

**Figure EV5. p38 MAPK pathway is upregulated in Treg cells in obese human adipose tissue.**

(A–G) The analysis was performed using human white adipose tissue single-cell RNA-seq data from Emont et al (Emont et al, 2022). (A) Dot plot of the expression of the indicated regulatory T cell (Treg) marker genes in the different cell type clusters. (B) Dot plot of the expression of the indicated genes by BMI range in human white adipose tissue Treg cluster shown in (A). (C) Violin plot showing the level of *MAPK14* gene expression by BMI range in the human white adipose tissue Treg dataset. (D–E) Volcano plots of differentially expressed genes in human white adipose tissue Treg subcluster in obese (BMI 30–40 kg/m²) versus non-obese (BMI 20–30 kg/m²) subjects (D) and in severe obese (BMI 40–50 kg/m²) versus non-obese (BMI 20–30 kg/m²) subjects (E). (F) Violin plot showing the level of *MAP2K3* gene expression by BMI range in the human white adipose tissue Treg dataset. (G) Volcano plots of differentially expressed genes in human white adipose tissue Treg subcluster in class 1 and 2 obesity (BMI 30–40 kg/m²) versus non-obese (BMI 20–30 kg/m²) subjects. The vertical dashed lines in (D, E, G) indicate a log2 fold change cut-off of 0.25. The horizontal dashed lines in (D, E, G) indicate a −log10 p-value cut-off of 1.3 (p-value < 0.05). Data information: Differentially expressed genes between BMI ranges were identified with a non-parametric Wilcoxon rank sum test. Obese (BMI 30–40 kg/m²) $N = 3$ biologically independent patients; Severe obese (BMI 40–50 kg/m²) $N = 6$ biologically independent patients; non-obese (BMI 20–30 kg/m²) $N = 5$ biologically independent patients. DC: dendritic cells; Mac: macrophages; Mono: monocytes; Neu: neutrophiles; Mast: mastocytes.

