## [Peer Review File · EMBO Reports]

Lack of p38 activation in T cells increases IL-35 and protects against obesity by promoting thermogenesis

Ivana Nikolic, Irene Ruiz-Garrido, María Crespo, Rafael Romero-Becerra, Luis Leiva-Vega, Alfonso Mora, Marta Leon, Elena Rodríguez, Magdalena Leiva, Ana Belen Plata-Gomez, Maria Beatriz Alvarez, Jorge Torres, Lourdes Hernández-Cosido, Juan Antonio López, Jesús Vázquez, Alejo Efeyan, Pilar Martin, Miguel Marcos, and Guadalupe Sabio

Corresponding author(s): Guadalupe Sabio (gsabio@cnic.es), Ivana Nikolic (INikolic@cnic.es)

Review Timeline:

Transfer Date:	11th Jan 24
Editorial Decision:	12th Jan 24
Revision Received:	31st Jan 24
Editorial Decision:	21st Feb 24
Revision Received:	5th Mar 24
Editorial Decision:	5th Apr 24
Revision Received:	15th Apr 24
Accepted:	17th Apr 24

Editor: Achim Breiling

Transaction Report: This manuscript was transferred to EMBO reports following peer review at Review Commons and The EMBO Journal.

The logo for Review Commons, featuring the word "Review" in a large, blue, serif font and "COMMONS" in a smaller, blue, sans-serif font below it.

Review #1

1. Evidence, reproducibility and clarity:

Evidence, reproducibility and clarity (Required)

In this study, Nikolic et al. show a novel role for p38 signaling in Treg cells, which impacts adipocytes through IL-35. This mechanism seems to be important for adipose tissue browning and metabolic health and could be potentially therapeutically exploited.

****Major comments:****

1. A control group of mice fed chow diet is needed to distinguish the effects of the genotype from those caused by diet. What is the phenotype of regular chow-fed mice in terms of energy metabolism and thermogenesis?
2. While an increase in BAT temperature (as demonstrated here by infrared imaging) in line with increased thermogenesis, it will be critical to verify this hypothesis by indirect calorimetry. Energy expenditure, food intake, and activity measures should be added for regular and DIO mice. Please follow the guidelines for ANCOVA analysis and measurements explained in PMID: 22205519 and PMID: 21177944.
3. That the phenotype is still seen at isothermal housing is interesting but should be backed up by direct assessment of thermogenic capacity (see PMID: 21177944). In the end, it could also be increased heat loss, independently of heat production. If the browning is cause or consequence remains unclear, then.
4. Regarding the in vitro data, a thermogenic phenotype should be functionally verified by Seahorse analysis.
5. Mechanistically, there is epistasis type of experiment that IL-35 influences Ucp1 levels via ATF2 as the data remain associative in nature.
6. What are other consequences of injecting IL-35? Is it good or bad? What is the therapeutic potential in DIO mice? Also, in these experiments (Fig. 7) indirect calorimetry as described would be supportive of the claims.

****Minor comments:****

7. The authors claim that their HFD-fed MKK3/6CD4-KO mice are protected against hyperglycemia, but only fasted/fed blood glucose tests are performed. Lower glucose levels could be explained due to a hyperinsulinemic state in response to growing insulin resistance in the presence of HFD. It would be sensible to perform both glucose and insulin tolerance tests to back up your statement.

8. Please provide the loading control for p38 and S6 blots (Figure 6G).
9. Statistical test from Figure 7B should be a t-test, since it is only comparing 2 variables (PBS vs IL-35), and not a 2-way ANOVA as described in the legend.
10. Label correctly the panels in the figures -examples: Fig 3, panels C and D are interchanged; reference in the text to Fig S1G even though the figure only as panels A-F; Fig 7 legend refers to the statistical test of panel E when the figure only has A-D.
11. There are several typos along the text, please revise (example: page 4; line 4 - "tremorgenic")

****Referees cross-commenting****

I think we three reviewers are pretty much on the same page - mouse energy metabolism explored too little and the mechanistic insight a bit thin considering the relatively strong claims.

2. Significance:

Significance (Required)

The manuscript is well written, and the research conducted properly, even though a thorough analysis of energy metabolism in mice and cells is missing and the mechanistic claims are based on relatively thin data.

The immune system and inflammation play important roles for obesity and insulin resistance, yet the roles they play in thermogenic adipocytes remains unclear. This work adds novel aspects to this relationship.

3. How much time do you estimate the authors will need to complete the suggested revisions:

Estimated time to Complete Revisions (Required)

(Decision Recommendation)

More than 6 months

4. Review Commons values the work of reviewers and encourages them to get credit for their work. Select 'Yes' below to register your reviewing activity at Web of Science

Reviewer Recognition Service (formerly Publons); note that the content of your review will not be visible on Web of Science.

Yes

Review #2

1. Evidence, reproducibility and clarity:

Evidence, reproducibility and clarity (Required)

This manuscript by Nikolic et al sought to investigate the role of p38 activation in adipose tissue Treg cells and obesity. They found that the expression of p38a, its upstream kinase MKK6, and downstream substrate ATF2 was upregulated specifically in adipose T cells associated with human obesity. They generated T cell-specific knockout MKK3/6 in mice and found these animals were protected from diet-induced obesity as a result of increased BAT thermogenesis. Mechanistically, loss of p38a activation promoted adipose tissue accumulation of Treg cells, leading to elevated IL-35 availability and UCP1 expression.

****Major comments:****

1. They attributed the obesity protection to energy expenditure; however, food intake and intestinal absorption were never tested. Immune cells particularly Treg cells are important modulates of nutrient uptake.
2. At thermoneutrality, BAT is inactive even though UCP1 expression is still present (not activated). MKK3/6 deficiency in T cells still confer protection against obesity at thermoneutrality suggests it regulates other energy balance components in addition to BAT thermogenesis.
3. Loss of adipose Treg cells (such as Pparg KO, Foxp3-DTR) did not lead to obvious obesity phenotypes. Gain-of-function Treg cells (such as adoptive transfer, IL-2/IL-2 Ab) did not results in profound obesity protection as observed in MKK3/6 CD4-KO mice. It suggests that MKK3/6 KO in T cells causes other immune defects (besides Tregs).

4. The increase in IL-35 seemed to be very moderate, compared to the metabolic phenotypes. It raises the question if IL-35 is responsible for BAT activation and reduced weight gain. It is unclear what systemic and local levels of IL-35 were reached after recombinant IL-35 treatment (Fig. 7B). IL-35 antibody blockade experiment in KO mice is recommended.
5. IL-35 induced p-ATF2 is acute and transient (Fig. 7D) and it was able to increase BAT temperature in just 4 h (Fig. 7B). However, Ucp1 transcription and translation generally take much longer time (e.g. 2d in Fig. 7C). IL-35 may increase energy expenditure through UCP1-independent mechanisms.

****Minor comments:****

1. The gating of Treg cells should exclude CD25⁻ cells. Single positive (CD25⁺ or Foxp3⁺) cells are progenitors of Tregs. In addition to number, phenotypic activation of Treg cells should also be determined.
2. ATF is also important for adipogenesis, is the adipogenic differentiation of BAT SVF cells affected by MKK3/6 KO or IL-35 treatment?
3. Metabolic cage experiments are desired to determine whole-body energy balance, including food intake, physical activity, and heat production.
4. Total UCP1 expression (both RNA and protein) in the whole BAT from an animal should be determined (since BAT is smaller in KO mice).
5. Fig. 6C, IL-35-expressing Treg cells should be quantified from adipose tissue.

****Referees cross-commenting****

I agree with Reviewer #1. In addition to energy metabolism and mechanistic action of IL-35, more rigor characterization of adipose Treg cells is needed.

2. Significance:

Significance (Required)

The manuscript is innovative in defining the novel role of p38 activation in the T cell compartment and its metabolic regulation. The involvement of Treg cells in adipose tissue homeostasis has been well documented and Treg cell-derived IL-35 has been demonstrated in immune regulation. The authors provided a relatively thorough description of the altered metabolism in these Mkk3/6 CD4-KO mice; however, the reviewer has doubts if Treg cells and IL-35 are primary mechanisms of the observed protection from obesity. The manuscript would be much stronger if the model were Treg cell-specific KO and/or IL-35 deficiency in Treg cells reverses obesity resistance conferred by MKK3/6 deficiency. It is also suspected that BAT thermogenesis is not the

major reason, as BAT deficiency or UCP1 KO results in much milder phenotypes in mice, even at thermoneutrality.

3. How much time do you estimate the authors will need to complete the suggested revisions:

Estimated time to Complete Revisions (Required)

(Decision Recommendation)

Between 3 and 6 months

Yes

Review #3

1. Evidence, reproducibility and clarity:

Evidence, reproducibility and clarity (Required)

Nikolic et al. examine the metabolic outcome of T cell specific deletion of MKK3/6 (MKK3/6 CD4-KO), which are the main activators of p38. Previous studies have demonstrated that MKK3/6 CD4-KO leads to Treg expansion and that Tregs in adipose tissues are associated with improved metabolic homeostasis. In line with these observations, the authors show that MKK3/6 CD4-KO mice gain less weight and have more active brown fat thermogenesis on a HFD both at the room temperature and 30C housing conditions. They also find more Tregs and M2 macrophages in eWAT of

MKK3/6 CD4-KO. All of the metabolic parameters are compared to CD4-cre mice as the wild type controls. Mechanistically, the authors suggest that reduced p38 activation by MKK3/6 CD4-KO leads to increased IL-35 production by Tregs, which induces beiging/browning of adipose tissues to promote metabolic health.

The authors have spent most of the efforts conducting metabolic phenotyping of MKK3/6 CD4-KO mice. One potential issue is whether the non-littermate CD4-cre mice are the proper controls for the comparison. In addition, the mechanistic link of the IL-35-ATF2-UCP1/FGF21 axis has only been superficially addressed.

****Specific comments:****

1. It's important to use proper controls for mouse metabolic studies. The authors stated that CD4-Cre and MKK3/6 CD4-KO mice are all in the C57B/6L background. However, it would appear that these two lines were bred separately. The difference in the genetic background, despite minor, can lead to the observed phenotype, notably weight gain. Since the metabolic phenotypes seem to be driven by the weight difference, it is even more critical to include additional controls to validate the findings. For instance, crossing MKK3/6 f/f with one copy of CD4-Cre with MKK3/6 f/f to generate age-matched MKK3/6 CD4-KO and MKK3/6 f/f controls should be used to repeat major in vivo studies similar to those in Fig. 2-4.
2. The assessment of adipose tissue immune cell population in Fig. 5 was conducted after HFD-induced obesity. As mentioned above, the change in Treg and M2 cell percentage could be due to the body weight difference. The experiment should be repeated (with proper controls) in normal chow and after a few weeks of HFD when Treg numbers start to decline.
3. Data related to the mechanistic link in Fig. 6/7 are not robust and require a large amount of additional work to substantiate the claim. First of all, the role of IL-35 in BAT thermogenesis remains unclear. It's somewhat surprising to see a single dose of IL-35 i.v. injection is sufficient to increase BAT temperature in Fig. 7B. Minimally, the authors need to demonstrate that IL-35 treatment (perhaps after a few daily doses) is able to increase browning/beiging of fat cells and improve cold tolerance when placing the mice at 4 degree of several hours (and up to 3 days). Serum FGF21 level should also be measured after/during IL-13 treatment. Secondly, ATF2 knockout or knockdown in brown preadipocytes should be employed to demonstrate that IL-35 induced UCP1 and FGF21 expression is ATF2 dependent. Another key experiment is to use IL-35 deficient Treg model to definitively demonstrate the requirement of Treg IL-35 to maintain thermogenesis. However, this can be done in a follow up study.

2. Significance:

Significance (Required)

Dissipating energy as heat through brown or beige adipocyte-mediated thermogenesis is believed to be an effective way to combat obesity. The current study aims to characterize the p38 signaling pathway in T cells as a potential target to modulate browning or beiging of adipose tissues. This would be of interest to the basic biomedical research community, particularly in the area of immunometabolism. A major limitation is the concern of improper controls for the mouse models, which makes data interpretation difficult. In addition, the mechanistic studies lack in depth analyses to support the conclusion.

3. How much time do you estimate the authors will need to complete the suggested revisions:

Estimated time to Complete Revisions (Required)

(Decision Recommendation)

More than 6 months

Yes

Full Revision

Manuscript number: RC-2023-02032

Corresponding author(s): Guadalupe, Sabio and Ivana, Nikolic

1. General Statements [optional]

We have performed the revision trying to respond to as many reviewers' requestees as possible.

This section is mandatory. Please insert a point-by-point reply describing the revisions that were already carried out and included in the transferred manuscript.

Reviewer #1 (Evidence, reproducibility and clarity (Required)):

Major comments:

1. A control group of mice fed chow diet is needed to distinguish the effects of the genotype from those caused by diet. What is the phenotype of regular chow-fed mice in terms of energy metabolism and thermogenesis?

We thank the reviewer 1 for this interesting question. We have measured body and BAT temperature in chow-fed mice and found similar phenotype as in HFD-fed mice MKK3/6^{CD4-KO} mice. Mice lacking MKK3/6 in T cells have higher BAT temperature, compared both to chow-fed CD4-Cre and littermates (MKK3/6^{ff}). We also observed higher energy expenditure in chow-fed MKK3/6^{CD4-KO} mice. These data are included in Figure 1 of this Full Revision letter and in Supplementary figure 2.

Figure 1. MKK3/6 deficiency in T cells increases brown adipose tissue temperature. (A) Comparison of energy balance between ND-fed CD4-Cre and MKK3/6^{CD4-KO} mice examined in a metabolic cage over a 3-day period. Hour-by-hour lean–mass-corrected variation in energy expenditure (EE) (left panel; mean ± SEM; CD4-Cre n = 5 mice; MKK3/6^{CD4-KO} n = 5 mice); mean lean–mass-corrected EE (middle panel; mean ± SEM; CD4-Cre n = 5 mice; MKK3/6^{CD4-KO} n = 5 mice); and ANCOVA analysis of EE as a function of body weight (right panel; mean ± SEM; CD4-Cre n = 5 mice; MKK3/6^{CD4-KO} n = 5 mice). (B) Body temperature and skin temperature surrounding interscapular BAT. Right panels show representative infrared thermal images in (B) CD4-Cre and MKK3/6^{CD4-KO} mice and (C) littermates (MKK3/6^{fl/fl}) and MKK3/6^{CD4-KO} mice fed chow diet. Data are mean ± SEM, t test *p<0.05, **p<0.01.

2. While an increase in BAT temperature (as demonstrated here by infrared imaging) in line with increased thermogenesis, it will be critical to verify this hypothesis by indirect calorimetry. Energy expenditure, food intake, and activity measures should be added for regular and DIO mice. Please follow the guidelines for ANCOVA analysis and measurements explained in PMID: 22205519 and PMID: 21177944.

Thank you again for this important question. As the reviewer asked, we have performed experiments in metabolic cages to determinate energy expenditure, food intake and activity

measurements. In line with increased thermogenesis, our results showed that DIO MKK3/6^{CD4-KO} mice have increased energy expenditure, without significant differences in food and water intake. In addition, they have reduced locomotor activity, but without differences in rearing behavior between genotypes. These results suggest that protection against DIO observed in MKK3/6^{CD4-KO} mice is due to higher energy expenditure which is used to produce increased thermogenesis. These results are now in a revised figure 3 of the paper and in this Full Revision letter.

Figure 2. MKK3/6 deficiency in T cells protects against HFD-induced obesity compared to littermates. MKK3/6^{CD4-KO} and MKK3/6^{f/f} mice were fed a high-fat diet (HFD) for 9 weeks. (A) Comparison of energy balance between HFD-fed MKK3/6^{f/f} and MKK3/6^{CD4-KO} mice examined in metabolic cages. Hour-by-hour lean-mass-corrected variation in energy expenditure (EE) (left panel; mean ± SEM; MKK3/6^{f/f} n = 6 mice; MKK3/6^{CD4-KO} n = 6 mice); mean BW-corrected EE (middle panel; mean ± SEM; MKK3/6^{f/f} n = 6 mice; MKK3/6^{CD4-KO} n = 6 mice); and ANCOVA analysis of EE as a function of body weight (right panel; mean ± SEM MKK3/6^{f/f} n = 6 mice; MKK3/6^{CD4-KO} n = 6 mice). (B) Food and drink intake during 24h. (C) Locomotor activity and rearing over 24 h. Data are mean ± SEM, *p < 0.05, ** p < 0.01, ***p < 0.001 MKK3/6^{f/f} versus MKK3/6^{CD4-KO}. Analysis by 2-way ANOVA coupled to the Sidak's multiple comparison post-test (B) or t test or by the Welch test when variances were different (A, B, C).

3. That the phenotype is still seen at isothermal housing is interesting but should be backed up by direct assessment of thermogenic capacity (see PMID: 21177944). In the end, it could also

be increased heat loss, independently of heat production. If the browning is cause or consequence remains unclear, then.

Thank you for raising this important point. According to your suggestion, we housed the animals at 30 °C for four weeks and subsequently injected norepinephrine (NE, 1 mg/kg of BW, i.p.) to evaluate thermogenesis capacity while measuring brown adipose tissue (BAT) temperature.

Our results showed that while HFD-fed MKK3/6^{CD4-KO} mice after 4 weeks at thermoneutrality have higher basal BAT temperature when stimulated with NE reach the same maximum of BAT temperature peak as control MKK3/6^{ff} mice. These results are now in Supplementary figure S5 and Figure 3 of this Full Revision letter.

Figure 2. Mice lacking MKK3/6 in T cells have a tendency to higher BAT temperature after NE stimulation. MKK3/6^{CD4-KO} and MKK3/6^{ff} mice were fed a high-fat diet (HFD) for 4 weeks and housed at thermoneutrality. (A) Interscapular BAT temperature at basal conditions and (B) after NE injection (1 mg/kg of BW, i.p.). Data are mean ± SEM, *p < 0.05, MKK3/6^{ff} versus MKK3/6^{CD4-KO}. Analysis by t test (A) or 2-way ANOVA coupled to the Bonferroni post-test (B).

4. Regarding the in vitro data, a thermogenic phenotype should be functionally verified by Seahorse analysis.

In agreement with the suggestion, we performed seahorse analysis in differentiated adipocytes treated with or without IL-35 for 48h. We observed slightly increased basal metabolism and response to isoproterenol (ISO) stimulation of β3 adrenergic receptors in adipocytes after IL-35 treatment. This result is showed in figure 4 of this Full Revision letter.

Figure for referee with unpublished data and its description has been removed upon request by the authors

5. Mechanistically, there is epistasis type of experiment that IL-35 influences Ucp1 levels via ATF2 as the data remain associative in nature.

Thank you for this comment. It has been shown that the ATF2 phosphorylation leads to its translocation to the nucleus and expression of UCP1 (PMID: 11369767. and PMID: 15024092). To study more profoundly the mechanism how IL-35 controls UCP1 levels, we treated adipocytes with IL-35 in the presence or absence inhibitor of ATF2 pathway and found reduced expression of *Ucp1*. These results are now in Figure 8F of the paper and Figure 5 of this Full revision letter.

Figure 5. ATF2 mediates IL-35 controlled UCP1 expression in adipocytes. Primary white preadipocytes were isolated from C57BL6 mice and differentiated in vitro. Once differentiated, cells were stimulated with IL-35 (100 ng/ml) for 48h in the presence or absence of ATF2 inhibitor (SB203580, 10 uM, the upstream inhibitor of ATF2 pathway). *Ucp1* level was measured by qRT-PCR and relativized to *b-actin*. Data are mean \pm SEM, t test * $p < 0.05$.

6. What are other consequences of injecting IL-35? Is it good or bad? What is the therapeutic potential in DIO mice? Also, in these experiments (Fig. 7) indirect calorimetry as described would be supportive of the claims.

Regarding the consequences of injecting IL-35, we have already performed experiments to analyze its effect. Our findings indicate that IL-35 increases thermogenesis in BAT (Figure 8C), suggesting that it may play a role in promoting energy expenditure, which could be beneficial in combating diet-induced obesity (DIO) in mice. Importantly, we did not observe any negative effects of IL-35 in our experiments.

Based on these promising results, we are expecting the therapeutic potential of IL-35 in DIO mice. By promoting thermogenesis in BAT, IL-35 may offer a novel approach to manage obesity and related metabolic disorders. However, we acknowledge that further comprehensive studies are needed to fully understand its therapeutic benefits and potential side effects.

In our future works, we plan to evaluate a targeted delivery system for IL-35. We are currently generating IL-35 loaded metal-organic frameworks (MOFs) labeled with adipose tissue-specific peptides. This innovative strategy aims to enhance the delivery of IL-35 to adipose tissue, potentially maximizing its effects in the relevant areas. Our ongoing work with IL-35 loaded MOFs may offer a promising avenue for targeted delivery.

Minor comments:

7. The authors claim that their HFD-fed MKK3/6CD4-KO mice are protected against hyperglycemia, but only fasted/fed blood glucose tests are performed. Lower glucose levels could be explained due to a hyperinsulinemic state in response to growing insulin resistance in the presence of HFD. It would be sensible to perform both glucose and insulin tolerance tests to back up your statement.

We appreciate the reviewer's comment. We have performed GTT and ITT analysis in HFD fed MKK3/6^{CD4-KO} mice and observed slightly improved glucose tolerance and significant insulin sensitivity in HFD-fed MKK3/6^{CD4-KO} mice. These results are now included below and in the Supplementary figure S4.

Figure 6. Deficiency of p38 activation in T cells slightly improves glucose tolerance and insulin sensitivity. MKK3/6^{CD4-KO} and control (CD4-Cre) mice were HFD-fed for 8 weeks. Mice were fasted overnight (for GTT) or 1 hour (for ITT), and blood glucose concentration was measured in mice given intraperitoneal injections of glucose (1 g/kg of total body weight) or insulin (0.75 U/kg of total body weight). Data are mean ± SEM, 2-way ANOVA *p < 0.05, ***p < 0.001.

8. Please provide the loading control for p38 and S6 blots (Figure 6G).

Thank you for the comment. The loading control we used is β-actin. These results are in Figure 7G of the revised paper.

Fig. 7. p38 activation in Treg cell inhibits IL-35 production. In vitro Treg cell induction (iTreg). Naïve CD4⁺ T cells were isolated from the spleens of CD4-Cre and MKK3/6CD4-KO

Full Revision

mice stimulated for 96h with plate-bound anti-CD3, soluble anti-CD28 + IL-2 +TGF β . Western blot analysis of p-s6 protein S240/244 and p-p38 Thr180/Tyr182 in iTreg cells from CD4-Cre (n=5) and MKK3/6^{CD4-KO} mice (n=4).

9. Statistical test from Figure 7B should be a t-test, since it is only comparing 2 variables (PBS vs IL-35), and not a 2-way ANOVA as described in the legend.

We thank the reviewer for the comment, it was a mistake in the text. We have performed t-test, however we have made mistake when writing the legend. Now it is corrected.

10. Label correctly the panels in the figures -examples: Fig 3, panels C and D are interchanged; reference in the text to Fig S1G even though the figure only as panels A-F; Fig 7 legend refers to the statistical test of panel E when the figure only has A-D.

We apologize for the mistakes that could have caused difficulties while reading the article and for potentially misleading the results. We appreciate the reviewer #1 for pointing out the errors in our manuscript. We have now corrected the article.

11. There are several typos along the text, please revise (example: page 4;line 4 -"tremorgenic")

We apologize for the typos, we have revised the article.

****Referees cross-commenting****

I think we three reviewers are pretty much on the same page - mouse energy metabolism explored too little and the mechanistic insight a bit thin considering the relatively strong claims.

Reviewer #1 (Significance (Required)):

The manuscript is well written, and the research conducted properly, even though a thorough analysis of energy metabolism in mice and cells is missing and the mechanistic claims are based on relatively thin data.

The immune system and inflammation play important roles for obesity and insulin resistance, yet the roles they play in thermogenic adipocytes remains unclear. This work adds novel aspects to this relationship.

We would like to say thank you for reviewer's thoughtful feedback. We appreciate the reviewer's acknowledgment of the manuscript's strong points and the insightful suggestions for improvement. We hope that our novel data now strengthen the manuscript hypothesis that Treg cells and IL-35 have a novel role in thermogenesis and metabolism.

Reviewer #2 (Evidence, reproducibility and clarity (Required)):

This manuscript by Nikolic et al sought to investigate the role of p38 activation in adipose tissue Treg cells and obesity. They found that the expression of p38a, its upstream kinase MKK6, and downstream substrate ATF2 was upregulated specifically in adipose T cells associated with human obesity. They generated T cell-specific knockout MKK3/6 in mice and found these animals were protected from diet-induced obesity as a result of increased BAT thermogenesis. Mechanistically, loss of p38a activation promoted adipose tissue accumulation of Treg cells, leading to elevated IL-35 availability and UCP1 expression.

Major comments:

1. They attributed the obesity protection to energy expenditure; however, food intake and intestinal absorption were never tested. Immune cells particularly Treg cells are important modulates of nutrient uptake.

We thank the reviewer #2 for this important suggestion. We have performed experiments in metabolic cages to determinate energy expenditure, food intake and activity measurements. In line with increased thermogenesis, our results showed that HFD fed MKK3/6^{CD4-KO} mice have increased energy expenditure, without significant differences in food and water intake. In addition, they have reduced locomotor activity, but without differences in rearing behavior between genotypes. These results suggest that protection against obesity observed in HFD fed MKK3/6^{CD4-KO} mice is due to higher energy expenditure which is used to produce increased thermogenesis. In addition, and following the reviewer suggestion, we also measured intestinal absorption by analyzing the lipid content in faces. Interestingly and opposite to the protection against obesity we observed and increase in lipid absorption in MKK3/6^{CD4-KO} mice. Future work would be necessary to understand this interesting effect in intestinal absorption. These results are now in a revised figure 3 of the paper and in figure 8 of this Full Revision letter.

Figure 8. MKK3/6 deficiency in T cells protects against HFD-induced obesity compared to littermates. MKK3/6^{CD4-KO} and MKK3/6^{fl/fl} mice were fed a high-fat diet (HFD) for 9 weeks. (A) Comparison of energy balance between HFD-fed MKK3/6^{fl/fl} and MKK3/6^{CD4-KO} mice examined in metabolic cages. Hour-by-hour lean–mass-corrected variation in energy expenditure (EE) (left panel; mean ± SEM; MKK3/6^{fl/fl} n = 6 mice; MKK3/6^{CD4-KO} n = 6 mice); mean BW-corrected EE (middle panel; mean ± SEM; MKK3/6^{fl/fl} n = 6 mice; MKK3/6^{CD4-KO} n = 6 mice); and ANCOVA analysis of EE as a function of body weight (right panel; mean ± SEM MKK3/6^{fl/fl} n = 6 mice; MKK3/6^{CD4-KO} n = 6 mice). (B) Food and drink intake during 24h. (C) Locomotor activity and rearing over 24 h. (D) Faecal lipid excretion over 5 days. Each point represents a data from individual mouse. Data are mean ± SEM, *p < 0.05, ** p < 0.01, ***p < 0.001 MKK3/6^{fl/fl} versus MKK3/6^{CD4-KO}. Analysis by 2-way ANOVA coupled to the Sidak's multiple comparison post-test (B) or t test or by the Welch test when variances were different (A, B, C, D).

2. At thermoneutrality, BAT is inactive even though UCP1 expression is still present (not activated). MKK3/6 deficiency in T cells still confer protection against obesity at thermoneutrality suggests it regulates other energy balance components in addition to BAT thermogenesis.

Thanks for the comment. We believe that the effects of IL35 on thermogenesis are likely partly mediated by alternative mechanisms, as we did not observe an increase in UCP1 gene expression in BAT in vivo (Figure 3D of the manuscript), and the increase in thermogenesis is still present even at thermoneutrality where UCP1 is inactive (Figure 4E of the manuscript). This suggests that IL35 might regulate other alternative pathways that control BAT thermogenesis.

While our current findings provide valuable insights, further experiments may be necessary to fully understand the underlying mechanisms. For instance, conducting experiments with transgenic mice expressing IL35 or using IL35 knockout (KO) mice could shed more light on the specific pathways through which IL35 exerts its effects on thermogenesis and energy balance.

In addition, we housed the animals at 30 °C for four weeks and subsequently injected norepinephrine (NE, 1 mg/kg of BW, i.p.) to evaluate thermogenesis capacity while measuring brown adipose tissue (BAT) temperature.

Our results showed that while HFD-fed MKK3/6^{CD4-KO} mice after 4 weeks at thermoneutrality have higher basal BAT temperature when stimulated with NE reach the same maximum of BAT temperature peak as control MKK3/6^{ff} mice. These results are now in Supplementary figure S5 and Figure 9 of this letter.

Figure 9. Mice lacking MKK3/6 in T cells have a tendency to higher BAT temperature after NE stimulation. MKK3/6^{CD4-KO} and MKK3/6^{ff} mice were fed a high-fat diet (HFD) for 4 weeks and housed at thermoneutrality. (A) Interscapular BAT temperature at basal conditions and (B) after NE injection (1 mg/kg of BW, i.p.). Data are mean \pm SEM, * $p < 0.05$, MKK3/6^{ff} versus MKK3/6^{CD4-KO}. Analysis by t test (A) or 2-way ANOVA coupled to the Bonferroni post-test (B).

In conclusion, we hypothesize that IL35's effects on thermogenesis are mediated partly by alternative mechanisms beyond UCP1 activation, and its ability to enhance thermogenesis even at thermoneutrality highlights its potential as a regulator of energy balance. We plan to further investigate the specific mechanisms through which IL35 impacts thermogenesis and energy balance. To achieve this, we will consider conducting experiments with transgenic mice expressing IL35 or using IL35 knockout (KO) mice in follow up studies. This is now discussed in our manuscript.

3. Loss of adipose Treg cells (such as Pparg KO, Foxp3-DTR) did not lead to obvious obesity phenotypes. Gain-of-function Treg cells (such as adoptive transfer, IL-2/IL-2 Ab) did not result in profound obesity protection as observed in MKK3/6 CD4-KO mice. It suggests that MKK3/6 KO in T cells causes other immune defects (besides Tregs).

We agree with the referee's assessment that the lack of obvious obesity phenotypes in above mentioned animal models. The results we observed in our MKK3/6^{CD4-KO} mice suggest that p38 signaling pathway in T cells may modulate their function, leading to an upregulation of IL35 expression, which could be a contributing factor to the significant obesity protection observed in MKK3/6^{CD4-KO} mice. We believe that IL35's effects on energy balance and thermogenesis are critical components of the observed protection against obesity in this model.

Regarding the studies with PPAR KO in Treg cells, it is important to note that they did not specifically focus on the effect of thermogenesis. While they observed a general tendency of increased fat deposition when treated with a PPAR agonist in the Treg deficient PPAR KO mice, these findings were not extensively studied in that particular paper. Thus, additional research is necessary to specifically evaluate thermogenesis in these mice and further understand the role of PPAR in Treg-mediated thermogenic processes.

We also acknowledge the presence of contradictory results from loss-of-function experiments of Treg cells in mice. The observed metabolic changes may be context-dependent, and the impact of Treg cells on metabolism might vary under different physiological conditions. For instance, in lean conditions where adipose tissue inflammation is low, a decrease in VAT Treg cells might not lead to significant metabolic changes. However, under certain circumstances, such as obesity, VAT Treg cells may play a critical role in regulating metabolism. In this context increasing that population that is reduced during obesity could result in improved metabolic performance.

In conclusion, our findings suggest that the lack of p38 activation in Treg cells may prevent the dramatic down-regulation and loss of function observed in Treg cells during obesity. This preservation of Treg function could be a significant factor driving the observed protection against obesity in MKK3/6^{CD4-KO} mice.

While further studies are required to elucidate the precise timing and spatial aspects of the specific functions of adipose-resident Treg cells, it is evident that these cells play a crucial role in maintaining immune and metabolic homeostasis. They achieve this, in part, by regulating adipose inflammation, insulin sensitivity, lipolysis, and thermogenesis. This is now discussed in our manuscript.

4. The increase in IL-35 seemed to be very moderate, compared to the metabolic phenotypes. It raises the question if IL-35 is responsible for BAT activation and reduced weight gain. It is

unclear what systemic and local levels of IL-35 were reached after recombinant IL-35 treatment (Fig. 7B). IL-35 antibody blockade experiment in KO mice is recommended.

Physiological changes in cytokines can indeed have a significant impact on the metabolic profile due to their continuous and intricate interactions. Even minor alterations in the overall cytokine milieu can result in substantial changes in metabolism (doi.org/10.1073/pnas.1215840110). In fact, it is well-established that in humans, small changes in cytokine profiles between genders, in obesity, and during aging can play a critical role in the development of pathology. These cytokines often operate in a chronic manner, exerting long-term effects on various physiological processes (doi.org/10.1038/s41467-020-14396-9).

In summary, the dynamic of cytokines in metabolism can lead to significant metabolic changes even with subtle alterations in their levels. While the increase in IL-35 may appear moderate, our findings using recombinant IL35 indicate that IL-35 increases thermogenesis in BAT, suggesting that it may play a role in promoting energy expenditure, which could be beneficial in combating diet-induced obesity (DIO) in mice. Importantly, we did not observe any negative effects of IL-35 in our experiments.

We are sorry as ethical reason in our animal facility prevent us to perform long term studies injecting mice with IL35 or the antibody.

5. IL-35 induced p-ATF2 is acute and transient (Fig. 7D) and it was able to increase BAT temperature in just 4 h (Fig. 7B). However, Ucp1 transcription and translation generally take much longer time (e.g. 2D in Fig. 7C). IL-35 may increase energy expenditure through UCP1-independent mechanisms.

Thanks for the comment. As previously mentioned, we believe that the effects of IL35 on thermogenesis might be mediated by alternative mechanisms, as we did not observe an increase in UCP1 gene expression in BAT, and the increase in thermogenesis is still present even at thermoneutrality where UCP1 is inactive. This suggests that IL35 might regulate other alternative pathways that control BAT thermogenesis.

While our current findings provide valuable insights, further experiments may be necessary to fully understand the underlying mechanisms. For instance, conducting experiments with transgenic mice expressing IL35 or using IL35 knockout (KO) mice could shed more light on the specific pathways through which IL35 exerts its effects on thermogenesis and energy balance. We plan to further investigate the specific mechanisms through which IL35 impacts thermogenesis and energy balance. To achieve this, we will consider conducting experiments with transgenic mice expressing IL35 or using IL35 knockout (KO) mice in follow up studies. This is now discussed in our manuscript.

Importantly we find that IL-35 increases serum levels of FGF21. Interestingly has been shown that FGF21 mediates the resistance to DIO and the remodeling of iWAT ('browning') in UCP1

Full Revision

KO mice suggesting that has thermogenic effect independently of UCP1 (doi: 10.1038/s41467-019-14069-2).

Figure for referee with unpublished data and its description has been removed upon request by the authors

Minor comments:

1. The gating of Treg cells should exclude CD25⁻ cells. Single positive (CD25⁺ or Foxp3⁺) cells are progenitors of Tregs. In addition to number, phenotypic activation of Treg cells should also be determined.

Thank you for the comment. We have reanalyzed our data by excluding CD25⁻ cells and included now in the figure 11 of this letter and 6A of the manuscript. We also checked CD69⁺ and KLRG1⁺ Treg cells and observed no differences between genotypes (Figure 11B of this letter).

Figure 11. Treg cells are increased in eWAT of mice lacking p38 activation. MKK3/6^{CD4-KO} and CD4-Cre mice were fed a high-fat diet (HFD) for 8 weeks. (A) FACS quantification and representative dot plots of Treg cells (CD4+CD25+Foxp3+) and (B) CD69+ and KLRG1+ Treg cells in SVF (mean ± SEM; CD4-Cre n = 7 mice; MKK3/6^{CD4-KO} n = 7 mice).

2. ATF is also important for adipogenesis, is the adipogenic differentiation of BAT SVF cells affected by MKK3/6 KO or IL-35 treatment?

We agree with the reviewer that ATF is also important for adipogenesis. We differentiated adipocytes in vitro and treated with IL-35 in the presence or absence of inhibitor of upstream activator of ATF. We observed that IL-35 treatment increased adipogenic markers, such as Pparg, Adipoq, Leptin and Perilipin, while inhibiting ATF activation reduced expression of adipogenic markers. These data are now in the figure 8G of the manuscript and in the Figure 12 of this Full Revision letter.

Figure 12. ATF2 mediates IL-35 controlled adipogenesis. Primary white preadipocytes were isolated from C57BL6 mice and differentiated in vitro. Once differentiated, cells were stimulated with IL-35 (100 ng/ml) for 48h in the presence or absence of SB203580 inhibitor (10 μM). The expression of principal adipogenic markers (*Pparg*, *Adipoq*, *Leptin*, *Perilipin*) level was measured by qRT-PCR and relativized to *b-actin*. Data are mean ± SEM, 1-way ANOVA *p < 0.05, **p < 0.01, ***p < 0.001.

3. Metabolic cage experiments are desired to determine whole-body energy balance, including food intake, physical activity, and heat production.

Thank you for this important comment. We have now performed experiments in metabolic cages to determinate energy expenditure, food intake and activity measurements. In line with increased thermogenesis, our results showed that ND and DIO MKK3/6^{CD4-KO} mice have increased energy expenditure, without significant differences in food and water intake. In

addition, they have reduced locomotor activity, but without differences in rearing behavior between genotypes. These results suggest that protection against DIO observed in MKK3/6^{CD4-KO} mice is due to higher energy expenditure which is used to produce increased thermogenesis. We also measured intestinal absorption by analyzing the lipid content in feces and observed increased lipid absorption MKK3/6^{CD4-KO} mice. These results are now in a revised Supplementary figure S2 and Figure 3 of the paper and in figures 13-14 of this Full Revision letter.

Figure 13. MKK3/6 deficiency in T cells increases brown adipose tissue temperature. (A) Comparison of energy balance between ND-fed CD4-Cre and MKK3/6^{CD4-KO} mice examined in a metabolic cage over a 3-day period. Hour-by-hour lean-mass-corrected variation in energy expenditure (EE) (left panel; mean ± SEM; CD4-Cre n = 5 mice; MKK3/6^{CD4-KO} n = 5 mice); mean lean-mass-corrected EE (middle panel; mean ± SEM; CD4-Cre n = 5 mice; MKK3/6^{CD4-KO} n = 5 mice); and ANCOVA analysis of EE as a function of body weight (right panel; mean ± SEM; CD4-Cre n = 5 mice; MKK3/6^{CD4-KO} n = 5 mice). (B) Body temperature and skin temperature surrounding interscapular BAT. Right panels show representative infrared thermal images in (B) CD4-Cre and MKK3/6^{CD4-KO} mice and (C) in littermates (MKK3/6^{fl/fl}) and MKK3/6^{CD4-KO} mice fed chow diet. Data are mean ± SEM, t test *p<0.05, **p<0.01.

Figure 14. MKK3/6 deficiency in T cells increases energy expenditure in HFD-induced obesity compared to littermates. MKK3/6^{CD4-KO} and MKK3/6^{ff} mice were fed a high-fat diet (HFD) for 9 weeks. (A) Comparison of energy balance between HFD-fed MKK3/6^{ff} and MKK3/6^{CD4-KO} mice examined in metabolic cages. Hour-by-hour lean-mass-corrected variation in energy expenditure (EE) (left panel; mean ± SEM; MKK3/6^{ff} n = 6 mice; MKK3/6^{CD4-KO} n = 6 mice); mean BW-corrected EE (middle panel; mean ± SEM; MKK3/6^{ff} n = 6 mice; MKK3/6^{CD4-KO} n = 6 mice); and ANCOVA analysis of EE as a function of body weight (right panel; mean ± SEM MKK3/6^{ff} n = 6 mice; MKK3/6^{CD4-KO} n = 6 mice). (B) Food and drink intake during 24h. (C) Locomotor activity and rearing over 24 h. (D) Fecal lipid excretion over 5 days. Each point represents a data from individual mouse. (E) Skin temperature surrounding interscapular BAT. Right panels show representative infrared thermal images. Data are mean ± SEM, *p < 0.05, **p < 0.01, ***p < 0.001 MKK3/6^{ff} versus MKK3/6^{CD4-KO}. Analysis by 2-way ANOVA coupled to the Sidak's multiple comparison post-test (B) or t test or by the Welch test when variances were different (A, B, C, D, E).

4. Total UCP1 expression (both RNA and protein) in the whole BAT from an animal should be determined (since BAT is smaller in KO mice).

Thank you for this comment. We have measured UCP1 expression in the whole BAT from the animals without observing changes. It is in the figure 4C and 4D and here in Figure 15 of Full Revision letter.

Figure 15. Deficiency of p38 activation in T cells increases UCP1 at protein and RNA level in brown adipose tissue. MKK3/6^{CD4-KO} and control (CD4-Cre) mice were HFD-fed for 8 weeks. UCP1 levels in BAT analyzed by western blot and qRT-PCR. mRNA expression was normalized to the expression of β -actin mRNA and presented as fold increase compared to CD4-Cre.

5. Fig. 6C, IL-35-expressing Treg cells should be quantified from adipose tissue.

We appreciate the referee's suggestion to quantify IL-35-expressing Treg cells from adipose tissue in Fig. 6C. While we agree that this would be valuable information, we encountered technical challenges that made it impractical to measure IL-35 directly in Treg cells from the visceral adipose tissue (VAT).

One of the main technical challenges we encountered is the low number of Treg cells present in the adipose tissue, making it difficult to obtain sufficient cell material for accurate quantification of IL-35. Treg cells are relatively rare compared to other immune cell populations in the adipose tissue, and their extraction and analysis can be technically demanding.

****Referees cross-commenting****

I agree with Reviewer #1. In addition to energy metabolism and mechanistic action of IL-35, more rigor characterization of adipose Treg cells is needed.

Reviewer #2 (Significance (Required)):

The manuscript is innovative in define the novel role of p38 activation in the T cell compartment and its metabolic regulation. The involvement of Treg cells in adipose tissue homeostasis has been well documented and Treg cell-derived IL-35 has been demonstrated in immune regulation. The authors provided a relatively thorough description of the altered metabolism in these Mkk3/6 CD4-KO mice; however, the reviewer has doubts if Treg cells and IL-35 are primary mechanisms of the observed protection from obesity. The manuscript would be much stronger if the model were Treg cell-specific KO and/or IL-35 deficiency in Treg cells reverses obesity resistance conferred by MKK3/6 deficiency. It also suspected that BAT thermogenesis is not the major reason, as BAT deficiency or UCP1 KO results in much milder phenotypes in mice, even at thermoneutrality.

We would also like to say thank you to reviewer #2 for the insightful suggestions and feedback. We agree that IL-35 deficiency or Treg specific KO models would be useful to confirm our data, but we believe that now with the new experiments we performed, we have improved our study and confirmed the novel role of p38 activation in T cells in metabolism and obesity.

Reviewer #3 (Evidence, reproducibility and clarity (Required)):

Specific comments:

1. It's important to use proper controls for mouse metabolic studies. The authors stated that CD4-Cre and MKK3/6 CD4-KO mice are all in the C57B/6L background. However, it would appear that these two lines were bred separately. The difference in the genetic background, despite minor, can lead to the observed phenotype, notably weight gain. Since the metabolic phenotypes seem to be driven by the weight difference, it is even more critical to include additional controls to validate the findings. For instance, crossing MKK3/6 f/f with one copy of CD4-Cre with MKK3/6 f/f to generate age-matched MKK3/6 CD4-KO and MKK3/6 f/f controls should be used to repeat major in vivo studies similar to those in Fig. 2-4.

We thank the reviewer for the comment. Although, every control is important using conditional mice, there are several papers indicating that all the cre expression lines have for their own effects that could be important in metabolism and there are several articles that strongly recommended to use cre+ lines as a control. For that reason, we have used the cre expressing line as a control because we really think is the best one (Jonkers and Berns, 2002). In fact, Jackson laboratory recommend to use cre expressing line as a control to avoid side effects that cre overexpression could have in the tissue of interest

(<https://biokamikazi.files.wordpress.com/2014/07/cre-lox-imp-notes.pdf>).

However, as this reviewer suggested, we checked that similar results were obtained using littermates as controls and we have now included these data in the manuscript (Supplementary Figure 2B) and Figure 16 and 17.

Figure 16. MKK3/6 deficiency in T cells results in lower body weight and increased body and BAT temperature compared to littermates. (A) Body weight of MKK3/6^{CD4-KO} mice littermates (MKK3/6^{ff}) fed chow diet. (B) Body temperature and skin temperature surrounding interscapular BAT. Right panels show representative infrared thermal images in littermates (MKK3/6^{ff}) and MKK3/6^{CD4-KO} mice fed chow diet. Data are mean +/- SEM, n=6 MKK3/6^{ff} mice and n=8 MKK3/6^{CD4-KO} mice, t test *p < 0.05, ***p < 0.001.

As the reviewer #3 suggested, we have also performed experiments in HFD-fed MKK3/6^{CD4-KO} and littermates, MKK3/6^{ff} mice, and observed protection against DIO. This was accompanied by reduced body and fat mass in MKK3/6^{CD4-KO} mice. We have also performed metabolic cages analysis to determinate energy expenditure, food intake and activity measurements. In line with increased thermogenesis, our results showed that DIO MKK3/6^{CD4-KO} mice have increased energy expenditure, without significant differences in food and water intake. In addition, they have reduced locomotor activity, but without differences in rearing behavior between genotypes. These results suggest that protection against DIO observed in MKK3/6^{CD4-KO} mice is due to higher energy expenditure which is used to produce increased thermogenesis. Interestingly, we also measured intestinal absorption by analyzing the lipid content in feces and observed increased lipid absorption MKK3/6^{CD4-KO} mice. These results are now in a revised figure 3 of the paper and in figure 17 of this Full Revision letter.

Figure 17. MKK3/6 deficiency in T cells are protected against obesity due to an increased energy expenditure compared to littermates. MKK3/6^{CD4-KO} and MKK3/6^{fl/fl} mice were fed a high-fat diet (HFD) for 9 weeks. (A) Body weight increased after starting HFD. (B) MRI analysis of body and fat mass and representative images on the left. (C) Comparison of energy balance

between HFD-fed MKK3/6^{f/f} and MKK3/6^{CD4-KO} mice examined in metabolic cages. Hour-by-hour lean–mass-corrected variation in energy expenditure (EE) (left panel; mean ± SEM; MKK3/6^{f/f} n = 6 mice; MKK3/6^{CD4-KO} n = 6 mice); mean BW-corrected EE (middle panel; mean ± SEM; MKK3/6^{f/f} n = 6 mice; MKK3/6^{CD4-KO} n = 6 mice); and ANCOVA analysis of EE as a function of body weight (right panel; mean ± SEM MKK3/6^{f/f} n = 6 mice; MKK3/6^{CD4-KO} n = 6 mice). (D) Food and drink intake during 24h. (E) Locomotor activity and rearing over 24 h. (F) Fecal lipid excretion over 5 days. Each point represents a data from individual mouse. (G) Skin temperature surrounding interscapular BAT. Right panels show representative infrared thermal images. Data are mean ± SEM, *p < 0.05, ** p < 0.01, ***p < 0.001 MKK3/6^{f/f} versus MKK3/6^{CD4-KO}. Analysis by 2-way ANOVA coupled to the Sidak’s multiple comparison post-test (A, C) or t test or by the Welch test when variances were different (B, D, E, F, G).

2. The assessment of **adipose tissue immune cell population** in Fig. 5 was conducted after HFD-induced obesity. As mentioned above, the change in Treg and M2 cell percentage could be due to the body weight difference. The experiment should be repeated (with proper controls) in normal chow and after a few weeks of HFD when Treg numbers start to decline.

Thank you for the comment. We have evaluated Treg and M2 cell population adipose tissue of age matched MKK3/6^{CD4-KO} and CD4-Cre mice fed chow diet. We observed slightly increased of Treg cells and CD206+ macrophages in MKK3/6^{CD4-KO} mice (represented as percentage and cell number as reviewer suggested).

Figure 18. MKK3/6 deficiency in T cells results tendency to increased Treg cells in eWAT in basal conditiona. FACS analysis of Treg cells and M2 macrophage population in eWAT tissue isolated from ND-fed MKK3/6^{CD4-KO} and control (CD4-Cre) mice. Data are mean +/- SEM, t test *p < 0.05.

Full Revision

We have also evaluated Treg and M2 cell population in a short HFD experiment in adipose tissue and also in lymph nodes using the littermates as controls. We fed MKK3/6f/f and MKK3/6^{CD4-KO} mice with HFD for only two weeks and analyzed Treg cells and M2 macrophages populations before there were differences in body weight between genotypes. We found increased frequency of Treg cells in lymph nodes and circulation, suggesting that Treg cells were migrating to inflamed tissue, such as adipose tissue during obesity. As suggested by the reviewer, we have corrected the number of Tregs infiltrating the tissue by fat mass, to avoid body/fat differences, and they had higher number of cells per gram of tissue compared to littermates. Furthermore, we found more CD206+ macrophages in eWAT of MKK3/6^{CD4-KO} mice, confirming protective phenotype observed even when there are no changes in the amount of fat depots and before there are differences in body weight. These results suggest that MKK3/6^{CD4-KO} mice have more Treg and M2 macrophages in adipose tissue, while Treg cell number declines in littermates shortly after HFD has been started. These results are now in Supplementary figure S9 of revised manuscript and figure 19 of this Full Revision letter.

Figure 19. MKK3/6 deficiency in T cells results increased Treg cells in lymph nodes, blood and eWAT after 2 weeks of HFD. MKK3/6^{ff} and MKK3/6^{CD4-KO} mice were fed with HFD for 2 weeks. (A) Body and (B) fat depots weight after 2 weeks of HFD. (C-D) FACS analysis of Treg cells in lymph nodes and blood. (E) FACS analysis of M2 macrophage population and Treg cells in eWAT tissue isolated from HFD-fed MKK3/6^{CD4-KO} and control (MKK3/6^{ff}) mice. Data are mean +/- SEM, t test **p* < 0.05, ****p* < 0.001.

3. Data related to the mechanistic link in Fig. 6/7 are not robust and require a large amount of additional work to substantiate the claim. First of all, the role of IL-35 in BAT thermogenesis remains unclear. It's somewhat surprising to see a single dose of IL-35 i.v. injection is sufficient

to increase BAT temperature in Fig. 7B. Minimally, the authors need to demonstrate that IL-35 treatment (perhaps after a few daily doses) is able to increase browning/beiging of fat cells and improve cold tolerance when placing the mice at 4 degree of several hours (and up to 3 days). Serum FGF21 level should also be measured after/during IL-13 treatment. Secondly, ATF2 knockout or knockdown in brown preadipocytes should be employed to demonstrate that IL-35 induced UCP1 and FGF21 expression is ATF2 dependent. Another key experiment is to use IL-35 deficient Treg model to definitively demonstrate the requirement of Treg IL-35 to maintain thermogenesis. However, this can be done in a follow up study.

We appreciate all the comments from the reviewer #3. As we are unable to perform several sequent i.v. injections in our animal facility due to ethical permission of our animal facility, we have performed cold tolerance test in our control mice and MKK3/6^{CD4-KO} mice, which are expressing higher levels of IL-35 to evaluate its role in adaptive thermogenesis. We observed that MKK3/6^{CD4-KO} mice exposed to cold preserved their body and BAT temperature, while the temperature of control CD4-Cre mice gradually dropped during cold challenge (Fig. 8B of the manuscript and figure 20 of this Full Revision letter).

Figure 20. Lack of p38 activation in T cells improves adaptive thermogenesis in response to cold challenge. MKK3/6^{CD4-KO} and control (CD4-Cre) mice were exposed to cold for 4 hours. Body and BAT temperature was measured every hour (n=5). Data are mean ± SEM, *p < 0.05, ** p < 0.01. Analysis by 2-way ANOVA.

We also measured FGF21 protein level in serum of mice after IL-35 treatment and observed mild increase compared to the group treated with PBS (Figure 21).

Figure 21. Lack of p38 activation in T cells improves adaptive thermogenesis in response to cold challenge. C57BL6 mice were treated with recombinant IL-35 i.v. (300 ng per mouse) and sacrificed 4h later (mean \pm SEM; PBS n = 7; IL-35 treated mice n = 9). FGF21 level was measured in serum and analyzed by western blot. Below is WB quantification. Data are mean \pm SEM, t test.

We agree with the reviewer #3 that using IL-35 deficient Treg model would be great approach to confirm our results, but we think that now with the additional experiments we have performed, we strength our findings that IL-35 has a novel role in controlling adipose tissue thermogenesis.

Reviewer #3 (Significance (Required)):

Dissipating energy as heat through brown or beige adipocyte-mediated thermogenesis is believed to be an effective way to combat obesity. The current study aims to characterize the p38 signaling pathway in T cells as a potential target to modulate browning or beiging of adipose tissues. This would be of interest to the basic biomedical research community, particularly in the area of immunometabolism. A major limitation is the concern of improper controls for the mouse models, which makes data interpretation difficult. In addition, the mechanistic studies lack in depth analyses to support the conclusion.

We would like to say thank you to the reviewer #3 for the feedback and suggestion to improve our study. As the reviewer suggested, we have performed the experiments using littermates as controls, and observed protection against DIO due to an increased energy expenditure and BAT temperature in mice deficient for MKK3/6 in T cells. We also found increased number of Treg cells in eWAT of MKK3/6^{C4-KO} mice early after HFD started and before differences in body or fat

Full Revision

depot weight, suggesting the effects we observed in the absence of p38 activation in T cells are intrinsic and not due to a leaner phenotype observed in these mice.

Decision Letter from The EMBO Journal

Date: 21st Dec 23 05:32:19

Last Sent: 21st Dec 23 05:32:19

Triggered By: Ieva Gailite

From: i.gailite@embojournal.org

To: gsabio@cnio.es

BCC: office@reviewcommons.org

Subject: Decision on Manuscript EMBOJ-2023-115232R | [RC-2023-02032] [D]

Message: Dear Dr. Sabio,

Thank you for submitting your manuscript for consideration by The EMBO Journal. We have now received comments from all original reviewers, which are included below for your information. Based on these comments, we unfortunately had to conclude that the revised study is not a sufficiently strong candidate for publication in The EMBO Journal.

As you can see, while reviewer #3 is generally satisfied with the revision, reviewers #1 and #2 indicate partially overlapping concerns regarding the conclusiveness of the analyses that would require a further round of revisions. Based on our earlier discussion, I conclude that such further revisions are likely not feasible and would not be in line with our "single major revision only" policy at our journal. Therefore, I am afraid that we cannot invite a second round of revision in The EMBO Journal.

While we cannot pursue this manuscript further, I have discussed your manuscript and the referee comments with my colleague Achim Breiling at our sister journal EMBO reports. I am glad to say that EMBO reports would be happy to proceed with the manuscript. EMBO reports emphasizes novel functional over detailed mechanistic insight, but asks for clear experimental support of the major conclusions. Thus, it will not be required to address points regarding more mechanism experimentally. However, it will be necessary that all points questioning the main conclusions of the study, and all technical concerns, or points regarding the experimental designs, model systems used, or data presentation, are addressed. In this case, it will be most important that the first major point of both referees #1 and #2 is sorted out. It needs to be clear from the current dataset that KO mice have higher EE, recalculating and depicting the data as suggested by the referees. Other conclusions might need to be toned down, as indicated by the referees. If you are interested in a transfer to EMBO reports, please contact Achim Breiling at a.breiling@emboreports.org with a point-by-point-response (rebuttal letter) to discuss the revision further.

Thank you in any case for the opportunity to consider this manuscript. I am sorry that I could not communicate more positive news, and I sincerely hope

that you will find the transfer option of interest.

With kind regards,

Ieva Gailite

Referee #1:

1. The authors responded to our question regarding energy expenditure with the appropriate measurements, and it seems to be a genotype effect rather than a diet related one, as they had claimed previously. However, the EE values corrected for mean lean mass can overcompensate for mass effect and lead to misinterpreted results. EE data should be plotted in relation to body mass rather than divided by it. This applies to hour-by-hour graph, histogram and linear regression. The authors used ANCOVA analysis according to PMID: 22205519 and PMID: 21177944 but the mouse numbers are too low to find a significant difference though it looks like the KOs have indeed higher EE.
2. As it is now proved that the increase in temperature is not necessarily due to brown fat activation/non-shivering thermogenesis, the conclusions on p38 activation in T cells promoting BAT thermogenesis should be rephrased.
3. Thank you for running the suggested experiment Seahorse experiment with IL-35. First, if this is about assessing increased browning and thermogenic capacity, it would be more suitable to work with a brown adipocyte cell line or a Ucp1 expressing one for that matter, rather than a white adipocyte line. There is a clear baseline difference from the beginning, so whether IL-35 sensitizes the cells to iso stimulation remains unclear.
4. This result on ATF2 looks promising for proving the causal relation, but controls are missing: no treatment and only ATF2 inhibitor should be included to make a full statement on the effect on Ucp1.

Referee #2:

This manuscript investigated the role of p38 activation in adipose tissue Treg cells and obesity. The expression of p38a, its upstream kinase MKK6, and downstream substrate ATF2 was upregulated specifically in adipose T cells associated with human obesity. T cell-specific knockout MKK3/6 mice were protected from diet-induced obesity as a result of increased BAT thermogenesis. While the protection phenotypes were characterized with rigor, major revisions are needed to establish BAT thermogenesis and IL-35 as mechanisms.

1. It is not accepted by the field to normalize energy expenditure to body weight or $BW^{0.75}$, particularly when body weights are different between groups. It is recommended to plot EE as kcal/h and analyze with ANCOVA. When performing regression (Fig. 3C and EV2A), EE in y axis should NOT be normalized to BW or lean mass. It is still questionable if KO mice have higher EE.

2. It is very not reliable to measure food intake using most metabolic cage systems, due to food spill. Highly recommended to weigh food intake daily for at least a week (particularly at the onset of diet switch) manually. By looking at Fig. 2D, it appears that KO mice increase their food intake. Some data points from the control groups were removed. And it's not possible for a mouse eat close to 0 gram of food in 24h (one mouse in the KO group), again indicating the inaccuracy of the method.

3. Do Ucp1 and other thermogenic genes show similarly upregulation in KO mice housed at 30C?

4. The authors attributed the obesity to defective BAT. However, when analyzing Tregs, only eWAT was used (Fig. 6, 7). eWAT does not contain UCP1+ beige adipocytes either. The role of MKK3/6 controlled Tregs in thermogenesis is unclear.

5. A recent paper suggest Treg control adipose lipid uptake and iron metabolism besides BAT thermogenesis (PMID: 35874700). Could MKK3/6 also regulate lipid metabolism in WAT and control food intake through lipid signals?

6. Increased IL-35 was only shown in lymph nodes (Fig. 7C). It is necessary to demonstrate IL-35 is required for obesity protection in KO mice. What ethical reasons prevent the treatment with IL-35 or antibody? At least some intro experiments are needed to demonstrate IL-35 expression is controlled by MKK3/6 and IL-35 mediates their effects on BAT.

Referee #3:

the authors have addressed most of my concerns.

Rev_Com_number: RC-2023-02032

New_manu_number: EMBOJ-2023-115232R

Corr_author: Sabio

Title: Lack of p38 activation in T cells increases IL-35 and protects against obesity by promoting thermogenesis

Dear Dr. Sabio,

Thank you for transferring your revised manuscript to EMBO reports. I now went through your manuscript, the referee reports from The EMBO Journal (attached again below) and your preliminary point-by-point-response (revision plan). Two referees have remaining concerns and suggestions to improve the manuscript, or to strengthen the data and the conclusions drawn, I ask you to address in a further revised manuscript, as indicated in your revision plan.

Moreover, I have these editorial requests I ask you to address in a final revised manuscript:

- Please provide the abstract written in present tense throughout.
- Please have your further revised manuscript text carefully proofread by a native speaker.
- Please add up to 5 keywords to the title page (below the abstract) and order the manuscript sections like this, using these names:
Title page - Abstract - Keywords - Introduction - Results - Discussion - Materials and Methods - Data availability section - Acknowledgements - Disclosure and Competing Interests Statement - References - Tables - Figure legends - Expanded View Figure legends
- Please remove the numbering of the sub-paragraphs from the results section.
- The Expanded View format, which will be displayed in the main HTML of the paper in a collapsible format, has replaced the Supplementary information. You can submit up to 5 images as Expanded View. Please follow the nomenclature Figure EV1, Figure EV2 etc. The figure legend for these should be included in the main manuscript document file in a section called Expanded View Figure Legends after the main Figure Legends section. Additional Supplementary material should be supplied as a single pdf file labeled Appendix. The Appendix should have page numbers and needs to include a table of content on the first page (with page numbers) and legends for all content. Please follow the nomenclature Appendix Figure Sx, Appendix Table Sx etc. throughout the text, and also label the figures and tables according to this nomenclature.

Presently, there are 10 EV figures. Thus, either try to combine these to have 5 final EV figures, or provide the additional data in an Appendix file as described above. Please update all callouts after these changes have been done.

- We updated our journal's competing interests policy in January 2022 and request authors to consider both actual and perceived competing interests. Please review the policy <https://www.embopress.org/competing-interests> and update your competing interests if necessary. Please name this section 'Disclosure and Competing Interests Statement' and put it after the Acknowledgements section.
- We now use CRediT to specify the contributions of each author in the journal submission system. CRediT replaces the author contribution section. Please use the free text box to provide more detailed descriptions and do NOT provide your revised manuscript text file with an author contributions section. See also our guide to authors: <https://www.embopress.org/page/journal/14693178/authorguide#authorshipguidelines>
- The Data Availability section should only contain information on large datasets that have been deposited to external repositories and all access information. Please remove the statement: 'All data generated or analyzed during this study are included in the main text or its supplementary material. Any further information is available from the corresponding author.' If no datasets have been deposited for this study, please just state here: 'No primary datasets have been generated and deposited'. Finally, please name this section 'Data availability section'.
- Please add scale bars of similar style and thickness to all the microscopic images, using clearly visible black or white bars (depending on the background). Please place these in the lower right corner of the images themselves. Please do not write on or near the bars in the image but define the size in the respective figure legend.
- Please make sure that the number "n" for how many independent experiments were performed, their nature (biological versus

technical replicates), the bars and error bars (e.g. SEM, SD) and the test used to calculate p-values is indicated in the respective figure legends (for main, EV and Appendix figures) of the final revised manuscript. Please also check that all the p-values are explained in the legend, and that these fit to those shown in the figure. Please provide statistical testing where applicable. Please avoid the phrase 'independent experiment', but clearly state if these were biological or technical replicates. Please also indicate (e.g. with n.s.) if testing was performed, but the differences are not significant. In case n=2, please show the data as separate datapoints without error bars and statistics. See also:

<http://www.embopress.org/page/journal/14693178/authorguide#statisticalanalysis>

If n<5, please show single datapoints for diagrams. Moreover:

- Please indicate the statistical test used for data analysis in the legends of figures 1c; EV10d, e, g.
- Please note that in figures 2a-d there is a mismatch between the annotated p values in the figure legend and the annotated p values in the figure file that should be corrected.
- Please note that information related to n is missing in the legends of figures 1c; EV10d, e, g.
- Please format the figure legends for main, EV and Appendix figures) according to our journal style. Moreover, please add to each legend a 'Data Information' section explaining the statistics used or providing information regarding replicates and scales.

- Please add the primer information (Table EV6) to the reagents table (see below).
- I would suggest to show Table EV5 as normal table. I.e. please name this table 'Table 1' and adjust the callouts and the legend.
- Tables EV1-EV4 are datasets. Please upload these excel files as dataset files named 'Dataset EVx' and add a title and a legend on the first TAB of each excel file. Finally, please correct all the callout for these files and remove their legends from the main manuscript text file.
- Please format the references according to our journal style:
<http://www.embopress.org/page/journal/14693178/authorguide#referencesformat>

If there are > 10 authors listed, please use et al.

- Please remove the reagents table from the main manuscript text file. I have attached templates for that in word or excel format. Please upload the filled in table to the manuscript tracking system as 'Reagent Table' file. Please also adjust any callouts to this table. The example linked below shows how the table will display in the published article and includes examples of the type of information that should be provided for the different categories of reagents and tools. Please list your reagents/tools using the categories provided in the template and do not add additional subheadings to the table. Reagents/tools that do not fit in any of the specific categories can be listed under "Other":

https://www.embopress.org/pb%2Dassets/embo-site/msb_177951_sample_FINAL.pdf

- Thank you for providing the source data. Please review the SD checklist, as 4H has been noted twice, and it seems one should be 4F. Moreover, please upload the source data files for EV and/or potential Appendix figures should ZIPed together as one folder.

In addition, I would need from you:

Best,

Achim Breiling
Senior editor
EMBO reports

Referee #1:

1. The authors responded to our question regarding energy expenditure with the appropriate measurements, and it seems to be a genotype effect rather than a diet related one, as they had claimed previously. However, the EE values corrected for mean lean mass can overcompensate for mass effect and lead to misinterpreted results. EE data should be plotted in relation to body mass rather than divided by it. This applies to hour-by-hour graph, histogram and linear regression. The authors used ANCOVA analysis according to PMID: 22205519 and PMID: 21177944 but the mouse numbers are too low to find a significant difference though it looks like the KOs have indeed higher EE.
2. As it is now proved that the increase in temperature is not necessarily due to brown fat activation/non-shivering thermogenesis, the conclusions on p38 activation in T cells promoting BAT thermogenesis should be rephrased.
3. Thank you for running the suggested experiment Seahorse experiment with IL-35. First, if this is about assessing increased browning and thermogenic capacity, it would be more suitable to work with a brown adipocyte cell line or a Ucp1 expressing one for that matter, rather than a white adipocyte line. There is a clear baseline difference from the beginning, so whether IL-35 sensitizes the cells to iso stimulation remains unclear.
4. This result on ATF2 looks promising for proving the causal relation, but controls are missing: no treatment and only ATF2 inhibitor should be included to make a full statement on the effect on Ucp1.

Referee #2:

This manuscript investigated the role of p38 activation in adipose tissue Treg cells and obesity. The expression of p38a, its upstream kinase MKK6, and downstream substrate ATF2 was upregulated specifically in adipose T cells associated with human obesity. T cell-specific knockout MKK3/6 mice were protected from diet-induced obesity as a result of increased BAT thermogenesis. While the protection phenotypes were characterized with rigor, major revisions are needed to establish BAT thermogenesis and IL-35 as mechanisms.

1. It is not accepted by the field to normalize energy expenditure to body weight or $BW^{0.75}$, particularly when body weights are different between groups. It is recommended to plot EE as kcal/h and analyze with ANCOVA. When performing regression (Fig. 3C and EV2A), EE in y axis should NOT be normalized to BW or lean mass. It is still questionable if KO mice have higher EE.
2. It is very not reliable to measure food intake using most metabolic cage systems, due to food spill. Highly recommended to weigh food intake daily for at least a week (particularly at the onset of diet switch) manually. By looking at Fig. 2D, it appears that KO mice increase their food intake. Some data points from the control groups were removed. And it's not possible for a mouse eat close to 0 gram of food in 24h (one mouse in the KO group), again indicating the inaccuracy of the method.
3. Do Ucp1 and other thermogenic genes show similarly upregulation in KO mice housed at 30C?
4. The authors attributed the obesity to defective BAT. However, when analyzing Tregs, only eWAT was used (Fig. 6, 7). eWAT does not contain UCP1+ beige adipocytes either. The role of MKK3/6 controlled Tregs in thermogenesis is unclear.
5. A recent paper suggest Treg control adipose lipid uptake and iron metabolism besides BAT thermogenesis (PMID: 35874700). Could MKK3/6 also regulate lipid metabolism in WAT and control food intake through lipid signals?
6. Increased IL-35 was only shown in lymph nodes (Fig. 7C). It is necessary to demonstrate IL-35 is required for obesity protection in KO mice. What ethical reasons prevent the treatment with IL-35 or antibody? At least some intro experiments are needed to demonstrate IL-35 expression is controlled by MKK3/6 and IL-35 mediates their effects on BAT.

Referee #3:

The authors have addressed most of my concerns.

Referee #1:

1. The authors responded to our question regarding energy expenditure with the appropriate measurements, and it seems to be a genotype effect rather than a diet related one, as they had claimed previously. However, the EE values corrected for mean lean mass can overcompensate for mass effect and lead to misinterpreted results. EE data should be plotted in relation to body mass rather than divided by it. This applies to hour-by-hour graph, histogram and linear regression. The authors used ANCOVA analysis according to PMID: 22205519 and PMID: 21177944 but the mouse numbers are too low to find a significant difference though it looks like the KOs have indeed higher EE.

Thank you for your comprehensive review and valuable comments on our manuscript. Addressing your concern regarding energy expenditure (EE), we have carefully re-evaluated our methodology. In response, we will present the EE data in relation to body mass measurements using an ANCOVA in the revised manuscript. We acknowledge the limitation of a relatively small sample size, which may hinder the statistical significance in this analysis. While there is a noticeable trend of higher EE in KO mice, we will temper our conclusions to align with these results.

Figure 1. MKK3/6 deficiency in T cells increases energy expenditure. MKK3/6^{CD4-KO} and MKK3/6^{fl/fl} mice were fed a high-fat diet (HFD) for 9 weeks. Comparison of energy balance between HFD-fed MKK3/6^{fl/fl} and MKK3/6^{CD4-KO} mice examined in metabolic cages. ANCOVA analysis of EE as a function of body weight (mean ± SEM MKK3/6^{fl/fl} n = 6 mice; MKK3/6^{CD4-KO} n = 6 mice).

However, to accurately depict the hour-by-hour graph and assess genotype differences, it is essential to incorporate body weight, given its influence on energy expenditure (EE). This approach, considering body weight in the representation of EE, is widely employed in obesity studies, as demonstrated in the literature (see relevant references).

Deadenylase-dependent mRNA decay of GDF15 and FGF21 orchestrates food intake and energy expenditure. **Cell Metabolism**, 2022

The mitochondrial calcium uniporter engages UCP1 to form a thermoporter that promotes thermogenesis **Cell Metabolism**, 2022

Yoshizawa, T., Sato, Y., Sobuz, S.U. et al. SIRT7 suppresses energy expenditure and thermogenesis by regulating brown adipose tissue functions in mice. **Nat Commun** 13, 7439 (2022). <https://doi.org/10.1038/s41467-022-35219-z>

Wang, Y., Gao, M., Zhu, F. et al. METTL3 is essential for postnatal development of brown adipose tissue and energy expenditure in mice. **Nat Commun** 11, 1648 (2020). <https://doi.org/10.1038/s41467-020-15488-2>

Yuan, Y., Fan, Y., Zhou, Y. et al. Linker histone variant H1.2 is a brake on white adipose tissue browning. **Nat Commun** 14, 3982 (2023). <https://doi.org/10.1038/s41467-023-39713-w>

Hepatic ZBTB22 promotes hyperglycemia and insulin resistance via PEPCK1-driven gluconeogenesis **EMBO reports** <https://doi.org/10.15252/embr.202256390>

Ventromedial hypothalamic OGT drives adipose tissue lipolysis and curbs obesity **SCIENCE ADVANCES** 31 Aug 2022 DOI: 10.1126/sciadv.abn8092

Seoane-Collazo, P., Liñares-Pose, L., Rial-Pensado, E. et al. Central nicotine induces browning through hypothalamic κ opioid receptor. **Nat Commun** 10, 4037 (2019). <https://doi.org/10.1038/s41467-019-12004-z>

In the graph from the manuscript “Total daily energy expenditure has declined over the past three decades due to declining basal expenditure, not reduced activity expenditure April 2023 Nature Metabolism 5(4):579-588 DOI:10.1038/s42255-023-00782-2”, it is evident that energy expenditure (EE) is dependent on the weight of the animal. This is why presenting a graph with total EE without incorporating weight can lead to misinterpretation. Thinner animals with higher EE per gram would show less total EE simply because they weigh less. This is the reason in the graph in which we put EE horu by hour we incorporate the body weight of the animals.

2. As it is now proved that the increase in temperature is not necessarily due to brown fat activation/non-shivering thermogenesis, the conclusions on p38 activation in T cells promoting BAT thermogenesis should be rephrased.

In response to the reviewer’s comment regarding the statement, “As it is now proved that the increase in temperature is not necessarily due to brown fat activation/non-shivering thermogenesis,” it appears there might be a misunderstanding. Our findings indicate an elevation in brown adipose tissue (BAT) temperature in mice with p38 pathway inactivation in T cells under 20°C, thermoneutral conditions and during cold exposure.

The reviewer's concern suggests a potential misinterpretation of our results. Our data demonstrate that the observed increase in temperature is not solely attributed to classical brown fat activation. Instead, our study suggests that this activation is also under thermoneutral conditions. We have observed an elevation in BAT temperature, not only in response to cold exposure but also under normal thermoneutral conditions, suggesting that p38 pathway inactivation in T cells influences BAT thermogenesis beyond traditional stimuli.

3. Thank you for running the suggested experiment Seahorse experiment with IL-35. First, if this is about assessing increased browning and thermogenic capacity, it would be more suitable to work with a brown adipocyte cell line or a Ucp1 expressing one for that matter, rather than a white adipocyte line. There is a clear baseline difference from the beginning, so whether IL-35 sensitizes the cells to iso stimulation remains unclear.

Thank you for your valuable feedback. We completely agree that utilizing a brown adipocyte cell line or one expressing Ucp1 would be more appropriate for assessing increased browning and thermogenic capacity. Unfortunately, we encountered challenges in culturing brown adipocytes.

Given this limitation and the observed phenotypic effects in white adipocytes, we opted to evaluate the impact of IL-35 on white adipocytes.

Thank you for highlighting the clear baseline difference observed from the beginning. We agree that further, more in-depth studies are necessary to address the function of IL-35 in both white and brown adipose tissue. We appreciate your insight and will consider this aspect in our future research.

4. This result on ATF2 looks promising for proving the causal relation, but

controls are missing: no treatment and only ATF2 inhibitor should be included to make a full statement on the effect on Ucp1.

We appreciate your recognition of the promising results regarding ATF2 and its potential causal impact. This experiment was specifically designed to assess the effects of ATF2 inhibition, aiming to understand whether these effects are linked to this pathway. We regret not including the recommended controls, such as a group with no treatment and another with only the ATF2 inhibitor.

We agree that these controls are essential for a complete interpretation of the results. We apologize for this oversight and acknowledge the need to incorporate these controls in future studies to validate and enhance our findings.

Referee #2:

This manuscript investigated the role of p38 activation in adipose tissue Treg cells and obesity. The expression of p38a, its upstream kinase MKK6, and downstream substrate ATF2 was upregulated specifically in adipose T cells associated with human obesity. T cell-specific knockout MKK3/6 mice were protected from diet-induced obesity as a result of increased BAT thermogenesis. While the protection phenotypes were characterized with rigor, major revisions are needed to establish BAT thermogenesis and IL-35 as mechanisms.

1. It is not accepted by the field to normalize energy expenditure to body weight or $BW^{0.75}$, particularly when body weights are different between groups. It is recommended to plot EE as kcal/h and analyze with ANCOVA. When performing regression (Fig. 3C and EV2A), EE in y axis should NOT be normalized to BW or lean mass. It is still questionable if KO mice have higher EE.

Thank you for your detailed review and valuable comments on our manuscript. Addressing your concern, we conducted a thorough reassessment of the energy expenditure (EE) data. Our revised approach involves presenting the EE data in relation to body mass measurements using an ANCOVA.

Figure 1. MKK3/6 deficiency in T cells increases energy expenditure. MKK3/6^{CD4-KO} and MKK3/6^{f/f} mice were fed a high-fat diet (HFD) for 9 weeks. Comparison of energy balance between HFD-fed MKK3/6^{f/f} and MKK3/6^{CD4-KO} mice examined in metabolic

cages. ANCOVA analysis of EE as a function of body weight (mean \pm SEM MKK3/6^{f/f} n = 6 mice; MKK3/6^{CD4-KO} n = 6 mice).

2. It is very not reliable to measure food intake using most metabolic cage systems, due to food spill. Highly recommended to weigh food intake daily for at least a week (particularly at the onset of diet switch) manually. By looking at Fig. 2D, it appears that KO mice increase their food intake. Some data points from the control groups were removed. And it's not possible for a mouse eat close to 0 gram of food in 24h (one mouse in the KO group), again indicating the inaccuracy of the method.

We appreciate your observations regarding the measurement of food intake using the metabolic cage system. We also acknowledge the reviewer suggestion regarding the daily weighing of mice food. However, we regret to inform you that we did not undertake this particular measurement methodology. We noticed that in regular cages, this type of measurement often fails when the mice interact with the high fat food, leading to destruction of the high-fat diet (HFD), with a significant portion ending up at the bottom of the cage mixed with bedding resulting in wrong measurement that indicate more the stress of the mice that the real food intake.

Regarding the absence of certain data points, it was not intentional. Regrettably, we do not have records for these mice due to technical complications encountered with the metabolic cage system. This lack of data was an unavoidable response to negative values arising from failures in the measurement system.

In an effort to address this concern, we have included data from another cohort of mice. We did not observe significant differences between the groups.

Figure 2. MKK3/6 deficiency in T cells does not affect food intake. MKK3/6^{CD4-KO} and CD4-Cre mice were fed a high-fat diet (HFD) for 9 weeks. Comparison of food intake between HFD-fed MKK3/6^{f/f} and MKK3/6^{CD4-KO} mice examined in metabolic cages (mean \pm SEM CD4-Cre n = 6 mice; MKK3/6^{CD4-KO} n = 9 mice).

3. Do Ucp1 and other thermogenic genes show similarly upregulation in KO mice housed at 30C?

We have analyzed the expression of UCP1, PGC1 α and FGF21 in brown adipose tissue and we have not found a significant increase in the expression of these thermogenic

genes in MKK3/6CD4-KO compared to control mice at thermoneutrality, although there is a tendency of increased PGC1a expression.

Figure 3. Thermogenic gene expression in brown adipose tissue isolated from HDF-fed mice in isothermal housing. MKK3/6^{CD4-KO} and CD4-Cre mice were fed a high-fat diet (HFD) for 8 weeks and housed at 30 °C during whole course of HFD. qRT-PCR analysis of thermogenic gene mRNA expression in BAT isolated from control or MKK3/6^{CD4-KO} mice. mRNA expression was normalized to the expression of β -actin mRNA and presented as fold increase compared to CD4-Cre (mean \pm SEM; CD4-Cre n = 9 mice; MKK3/6^{CD4-KO} n = 8 mice).

4. The authors attributed the obesity to defective BAT. However, when analyzing Tregs, only eWAT was used (Fig. 6, 7). eWAT does not contain UCP1+ beige adipocytes either. The role of MKK3/6 controlled Tregs in thermogenesis is unclear.

We appreciate the opportunity to address your concerns.

In response to your observation about Treg analysis specifically in BAT, we encountered technical challenges obtaining sufficient infiltrate in this tissue for accurate measurements. Unfortunately, this limitation hindered our ability to obtain reliable data for BAT. However, it is important to note that, in our study, we focused on the analysis of Tregs in eWAT. Contrary to the mentioned statement, eWAT does contain UCP1+ beige adipocytes, as supported by existing literature (PMID: 22796012).

5. A recent paper suggest Treg control adipose lipid uptake and iron metabolism besides BAT thermogenesis (PMID: 35874700). Could MKK3/6 also regulate lipid metabolism in WAT and control food intake through lipid signals?

We appreciate your question regarding the potential regulation of lipid metabolism in WAT and control of food intake through lipid signals by MKK3/6, as suggested by a recent publication (PMID: 35874700). While this is indeed an interesting avenue of exploration, we recognize that a comprehensive understanding of the role of Treg cells in lipid metabolism in AT would require further mechanistic studies which falls slightly outside the scope of our current study's objectives. We agree with the reviewer that we cannot rule out the contribution of adipose lipid uptake and iron metabolism in our model, and we will carefully address these limitations in our revised manuscript in EMBO reports.

6. Increased IL-35 was only shown in lymph nodes (Fig. 7C). It is necessary to demonstrate IL-35 is required for obesity protection in KO mice. What ethical reasons prevent the treatment with IL-35 or antibody? At least some intro experiments are needed to demonstrate IL-35 expression is controlled by MKK3/6 and IL-35 mediates their effects on BAT.

Thank you for your comment. We appreciate your evaluation of our work.

We fully acknowledge the importance of establishing a comprehensive understanding of IL-35's role in obesity protection. Regrettably, due to ethical considerations and limitations within our current protocol, continuous intravenous injection of IL-35 or antibodies is not feasible within the scope of our study. To change the protocol will require around 6 months because of the legal regulation.

In figure 7D and 7E of our manuscript, we showed in conditions *in vitro* that Treg cells with suppressed MKK3/6 pathway (either genetically or chemically by using inhibitor) showed reduced IL-35 expression. Also, IL-35 stimulation in mice increases BAT temperature (figure 8C of our manuscript) and UCP-1 expression in adipocytes *in vitro* (figure 8D of our manuscript). While we recognize that additional experiments could potentially provide further insights, we want to emphasize that our study, as presented, provides valuable data contributing to the understanding of the role of MKK3/6-controlled Tregs in the context of BAT thermogenesis and obesity protection.

We are committed to addressing these limitations in the revised manuscript.

Dear Dr. Sabio,

Thank you for the submission of your revised manuscript to our editorial offices. I have already forwarded the report from the 2 referees that I asked to re-evaluate your study, you will find again below. I also have received your point-by-point-response (further revision plan). After looking through your revision plan, I decided to invite a final revised manuscript that addresses the remaining referee points as indicated in your revision plan. Please also provide a detailed final point-by-point-response to these.

- Please remove the bullet points and the synopsis blurb from the manuscript text file. I have saved these separately and will forward them to our publisher.
- There is a legend for Figure 9, but no such figure was uploaded. Please check.
- The synopsis image provided has not the right format and if reduced in size has partly too small text labels. Please provide an image in jpeg or tiff format (with the exact width of 550 pixels and a height of not more than 400 pixels) with bigger fonts.
- Thank you for providing the source data (SD). Please review the SD checklist, as 4H has been noted twice, and it seems one should be 4F. Please upload the SD for the main figures as one ZIPed folder per figure and the SD for Appendix and EV figures grouped together in one ZIPed folder.
- Please make sure that all figure panels (main, EV and Appendix figures) are called out separately and sequentially. Presently, Fig. 3A is called out before 2B, the panels of Figure 5 are called out before 4B and a callout for Fig. 2F seems missing. Moreover, there is a callout for Fig. S10B that needs to be updated with a correct figure number and name. Please check.
- Please make sure that all the funding information is also entered into the online submission system and that it is complete and similar to the one in the acknowledgement section of the manuscript text file. Presently, the grants IMMUNOTHERCAN-CM S2010/BMD-2326 and B2017/BMD-3733 and the Instituto de Salud Carlos III (ISCIII) are only mentioned in the acknowledgements section.
- Co-corr. author Ivana Nikolic does not have an institutional email address. Please update the account in the online system with an institutional address and provide one on the title page
- The notification email for María Crespo (maria.crespo@cnic.es) bounced. Please provide a valid e-mail address.

Yours sincerely,

Referee #1:

Thank your for responding to the comments. It is evident that the energy metabolism studies have limitations and the mouse numbers are tool small to reach meaningful and significant conclusions here. This should be acknowledged.

Referee #2:

The revision unfortunately did not address all my previous concerns.

1. It's good to now include ANCOVA analysis of EE, however, Figure 3C still plotted EE normalized to $BW^{0.75}$. This is not acceptable.
2. Yes, there are a few rare begie adipocytes in eWAT, but its overall UCP1 mRNA/protein expression is hundreds time lower

than that in BAT and activated iWAT. eWAT's contribution to body temperature control is very limited if not zero. When characterizing EE phenotypes, core and BAT temperature was measured; however, Tregs in eWAT were analyzed. Therefore, there is no direct evidence that Treg-derived IL35 contributes to thermogenic and EE changes in this study.

Unless this manuscript remove their conclusions indicating that the p38-Treg-IL35-thermogenesis axis controls body weight, it is not acceptable for publication. Alternatively, they should specifically state that T cells in weight control and Treg-derived IL35 might be two independent mechanisms of p38.

Referee 1:

1. Thank you for responding to the comments. It is evident that the energy metabolism studies have limitations and the mouse numbers are too small to reach meaningful and significant conclusions here. This should be acknowledged.

Yes, we agree with the referee that it is a limitation of our current experimental setup. It is conceivable that the magnitude of the observed increase in EE is not substantial enough to be reliably detected within the confines of the metabolic cages, especially considering variations in body weight among the experimental groups of the energy experiments and we will acknowledge these limitations in the manuscript.

Referee 2:

1. It's good to now include ANCOVA analysis of EE, however, Figure 3C still plotted EE normalized to $BW^{0.75}$. This is not acceptable.

To gain a more comprehensive understanding of our results regarding energy expenditure (EE), we have analysed additional data sets with a larger number of mice. We have evaluated and plotted EE data without normalization within the metabolic cages. We found no significant increase in EE. Despite the clear evidence of elevated temperature associated with the phenotype of our animals, we were unable to detect notable changes in EE within the metabolic cages when the data were not normalized to body weight.

We acknowledge that this discrepancy may stem from the limitations of our current experimental setup. It is conceivable that the magnitude of the observed increase in EE is not substantial enough to be reliably detected within the confines of the metabolic cages, especially considering variations in body weight among the experimental groups (heavier animals typically exhibit higher EE, whereas KO mice, being smaller, exhibit comparable EE to WT mice that are heavier, indicating that they have more EE per gram). For that reason, we have included both graphs without or with normalizing by the lean mass.

We value your feedback and we will amend the text to accurately reflect our findings and the limitations of our experimental approach.

Figure 1. MKK3/6 deficiency in T cells increases energy expenditure. MKK3/6^{CD4-KO} and CD4-cre mice were fed a high-fat diet (HFD) for 8 weeks. Comparison of energy balance between HFD-fed MKK3/6^{CD4-KO} and CD4-cre mice examined in metabolic cages. Energy expenditure hour by hour and hour by hour normalize by the lean mass. ANCOVA analysis of EE as a function of body weight (mean \pm SEM CD4-cre n = 8 mice; MKK3/6^{CD4-KO} n = 8 mice).

2. Yes, there are a few rare beige adipocytes in eWAT, but its overall UCP1 mRNA/protein expression is hundreds time lower than that in BAT and activated iWAT. eWAT's contribution to body temperature control is very limited if not zero. When characterizing EE phenotypes, core and BAT temperature was measured; however, Tregs in eWAT were analyzed. Therefore, there is no direct evidence that Treg-derived IL35 contributes to thermogenic and EE changes in this study.

Yes, it is true that the contribution of epididymal white adipose tissue (eWAT) to body temperature regulation is limited. However, we chose to focus on this WAT depot due to technical ease in isolating T cells. Additionally, we have data indicating a higher contribution to thermogenesis from Treg infiltration in inguinal white adipose tissue (iWAT), where we observed an increased amount of Treg cells (figure 2 of this letter).

Furthermore, despite facing significant technical challenges due to the low number of infiltrated cells in brown adipose tissue (BAT), we were able to detect a slight increase in Treg cells in BAT from MKK3/6^{CD4-KO} mice (Figure 2 of this letter).

Figure 2. MKK3/6 deficiency in T cells promotes Treg cell accumulation in iWAT and BAT. MKK3/6^{CD4-KO} and CD4-Cre mice were fed a high-fat diet (HFD) for 8 weeks. (A) FACS quantification and representative dot plots of Treg cells (CD4⁺CD25⁺Foxp3⁺) in iWAT and BAT.

3. Unless this manuscript removes their conclusions indicating that the p38-Treg-IL35-thermogenesis axis controls body weight, it is not acceptable for publication. Alternatively, they should specifically state that T cells in weight control and Treg-derived IL35 might be two independent mechanisms of p38.

We acknowledge your concerns regarding the conclusions drawn in our manuscript regarding the p38-Treg-IL35-thermogenesis axis and its control over body weight.

Our main conclusion suggests that the p38-Treg-IL35 axis may be involved in body weight control. However, it is important to note that we cannot dismiss the possibility that T cells in weight control and Treg-derived IL35 are two independent mechanisms of p38, as mentioned by the reviewer. We will mention it in this way in the revised manuscript.

Dear Dr. Sabio,

Thank you for the submission of your further revised manuscript to our editorial offices. I have now received the report from the referee that I asked to re-evaluate your study, you will find below. As you will see, the referee now supports the publication of your study in EMBO reports. S/he has some remaining points and suggestions to improve the study, I ask you to address in a final revised manuscript. As indicated by the referee, please show the normalized and non-normalized data side-by-side and explain this accordingly in the text. Please also provide a final p-b-p-response to the referee points.

Moreover, I have these final editorial requests:

- Please have your final manuscript carefully proofread by a native speaker. The text contains several grammatical errors and needs improvement.
- The synopsis image has not the right format (jpeg or tiff format with the exact width of 550 pixels and a height of not more than 400 pixels). Downsizing the present results in an image with too small fonts. Please provide an image in the right format and with bigger fonts.
- Could scale bars or other size markers be added to the images in panels 2B, 3B and 4B?
- The source data for Fig. 1 just contains again the figure file. Please check and provide the correct source data.

Best,

Referee #2:

I am mostly fine with the revision. However, I would prefer that the authors show the normalized and non-normalized data side-by-side and let the readers interpret. The most rigorous way to determine if EE is causing a body weight phenotype is to measure metabolic cage before the body weight diverges between groups. Unfortunately, this is not done in most publications.

The bottom line is that the manuscript shows an interesting phenotype when p38 activation is impaired in T cells. The authors haven't convincingly shown increased EE (thermogenesis nor that IL-35 is a direct cause of that. A compromise is to explicitly acknowledge these limitations in the discussion and not to directly imply that Treg IL-35 (shown in visceral WAT increases thermogenesis (in BAT, thus causing leanness.

All editorial and formatting issues were resolved by the authors.

Dr. Guadalupe Sabio
CNIO
C/ Melchor Fernández Almagro 3
Madrid 28029
Spain

Dear Dr. Sabio,

I am very pleased to accept your manuscript for publication in the next available issue of EMBO reports. Thank you for your contribution to our journal.

Yours sincerely,
